# Planning Anything with Rigor: General-Purpose Zero-Shot Planning with LLM-based Formalized Programming

**Yilun Hao**
MIT
yilunhao@mit.edu

**Yang Zhang**
MIT-IBM Watson AI Lab
Yang.Zhang2@ibm.com

**Chuchu Fan**
MIT
chuchu@mit.edu

## Abstract

While large language models (LLMs) have recently demonstrated strong potential in solving planning problems, there is a trade-off between flexibility and complexity. LLMs, as zero-shot planners themselves, are still not capable of directly generating valid plans for complex planning problems such as multi-constraint or long-horizon tasks. On the other hand, many frameworks aiming to solve complex planning problems often rely on task-specific preparatory efforts, such as task-specific in-context examples and pre-defined critics/verifiers, which limits their cross-task generalization capability. In this paper, we tackle these challenges by observing that the core of many planning problems lies in optimization problems: searching for the optimal solution (best plan) with goals subject to constraints (preconditions and effects of decisions). With LLMs' commonsense, reasoning, and programming capabilities, this opens up the possibilities of a universal LLM-based approach to planning problems. Inspired by this observation, we propose LLMFP, a general-purpose framework that leverages LLMs to capture key information from planning problems and formally formulate and solve them as optimization problems from scratch, with no task-specific examples needed. We apply LLMFP to 9 planning problems, ranging from multi-constraint decision making to multi-step planning problems, and demonstrate that LLMFP achieves on average 83.7% and 86.8% optimal rate across 9 tasks for GPT-4o and Claude 3.5 Sonnet, significantly outperforming the best baseline (direct planning with OpenAI o1-preview) with 37.6% and 40.7% improvements. We also validate components of LLMFP with ablation experiments and analyzed the underlying success and failure reasons. Project page: https://sites.google.com/view/llmfp.

## 1 Introduction

Making complex plans subject to multiple constraints is a time- and labor-intensive process, but is critical in many aspects of our lives such as work arrangement, business management, logistics, and robotics. In the past, people used domain-specific tools and languages to make specific plans in their areas, which often required a steep learning curve and were hard to adapt to other domains. When large language models (LLMs) emerge with their versatile capabilities such as language understanding, reasoning, and tool-using, using LLMs for planning has gained significant traction.

For such planning systems to be deployed in complex, real-world applications, two desirable properties need to be satisfied: 1). *Zero-shot flexibility*: Unlike many experimental settings where planning tasks usually come with labeled datasets, it is very challenging to request such datasets from users in many realistic settings. Ideally, a flexible planning system should be able to conduct planning with only a task description provided by users, and nothing else. 2). *High performance on complex tasks*: Realistic planning problems usually require multi-step, long-horizon solutions, with many explicit and implicit constraints.

However, there is a trade-off between flexibility and task complexity, and thus existing LLM-based planning systems are typically unable to achieve both simultaneously. On one hand, planning systems capable of performing zero-shot planning, utilizing the abundant knowledge and generalization

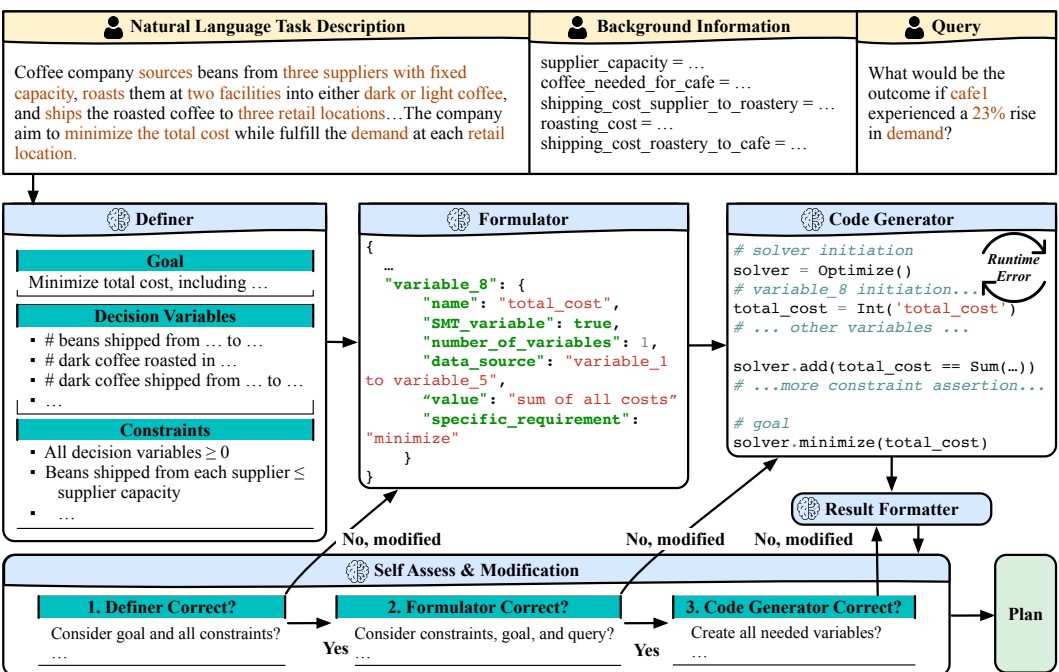

Figure 1: An overview of LLMFP and how it is applied to a coffee supply chain example. All sections in yellow are inputs, and all sections in blue are steps accomplished by LLMs. With task description, background information, and query as inputs, LLMFP defines the goal, decision variables, and constraints of this optimization problem, identifies all necessary variables and summarizes their key information into a JSON representation, generates codes to solve the optimization problem, executes the codes and formats the execution results, and performs self-assessment for each step.

capabilities in LLMs, in many successful applications can only solve single-objective tasks such as household chores, with step-by-step interactive planning and grounding (Huang et al., 2022a; Ahn et al., 2022; Huang et al., 2022b). For complex, multi-constraint, and long-horizon tasks that involve iterative trials and errors even for humans, LLMs still do not have the capabilities to generate valid plans by themselves (Kambhampati et al., 2024). On the other hand, recent research efforts to empower LLM-based planners to solve complex tasks are often based on well-designed task-specific in-context examples and extensive task-specific pre-defined efforts (Liu et al., 2023; Xie et al., 2023; Li et al., 2023; Song et al., 2023; Gundawar et al., 2024), impacting their zero-shot flexibility. In short, few existing planning systems can flexibly resolve generic complex tasks with only task descriptions in natural language. Hence we ask: Can we build a universal LLM-based planning system that can solve complex planning problems without task-specific efforts?

In this paper, we observe that, although planning problems come with drastic variations, many of them can be recast as constrained optimization problems — The optimization problems aim to find the optimal solution, which is equivalent to locating the best plan that satisfies the goal for planning problems; the decision's precondition and effect are equivalent to constraints of optimization problems. Furthermore, although solving complex planning tasks is generally beyond the capabilities of LLMs, converting any planning tasks into optimization problems is a much more tractable problem, and can be within the zero-shot capabilities of LLMs.

Motivated by this, we propose **LLM**-Based **F**ormalized **P**rogramming (LLMFP, illustrated in Fig. 1), a general-purpose zero-shot planning framework that leverages LLM's strong common reasoning, and programming capabilities to encode planning problems into optimization problems *without* any task-specific examples or designs, combined with a formal planner to solve the optimization problem. LLMFP takes in natural language domain description, natural language query under this domain, and available background information or APIs as inputs, and solves the planning problem in five steps. First, LLMFP prompts LLMs to reason and propose the goal, decision variables, and key constraints necessary for the task. Second, based on the response, LLMFP asks

LLMs to formulate a representation that includes all variables needed to construct and their detailed information and requirements. Third, with the representation, LLMs write codes to formally encode the problem into an optimization problem. Fourth, LLMFP executes the generated codes, converts the execution results into plans. Finally, LLMFP performs overall self-assessment and automatic modification to fix the broken parts of the previous steps. Currently, LLMFP uses the satisfiability modulo theory (SMT) to encode the optimization problems but can be adapted to any planners or solvers by updating the requirements and representation format in the prompts.

We evaluate our framework with 9 diverse planning problems, ranging from single-step supply chain problem to multi-step robot block stacking and moving. Experiment results demonstrate that LLMFP achieves strong performance across all tasks with an average of 83.7% and 86.8% optimal rates for GPT-4o and Claude 3.5 Sonnet, which greatly outperforms the baselines, including direct plan generation with OpenAI o1-preview. We conduct ablation experiments to validate the key components of LLMFP and investigate the underlying reasons why it is more effective than baselines. In addition, although our framework does not require task-specific examples, we show the ease of adding task-specific examples to one stage of LLMFP, and how it could help to clarify unclear queries and therefore can further improve the performance within the same domain.

In summary, our key contributions are:

- We offer a novel perspective on using LLMs to solve planning problems by rigorously constructing optimization problems from scratch, alike how human experts use optimization tools for planning.
- We propose LLMFP, a general-purpose planning framework with zero-shot generalization capability. To our knowledge, LLMFP is the first to enable LLMs to build and solve diverse types of planning problems as optimization problems with no task-specific examples or external critics.
- LLMFP notably achieves 83.7% and 86.8% optimal rates for GPT-4o and Claude 3.5 Sonnet, outperforming the best baseline (direct planning with OpenAI o1-preview) by 37.6% and 40.7%. We examine the effectiveness of our framework and analyze the success and failure reasons.

## 2 RELATED WORKS

### 2.1 LLMS FOR PLANNING

The remarkable capabilities of LLMs in reasoning (Wei et al., 2022; Kojima et al., 2022; Yao et al., 2022; 2024; Raman et al., 2024) and tool-use (Qin et al., 2023; Schick et al., 2024) brings up interests of utilizing LLMs for planning problems. Based on LLMs's zero-shot generalization capability, many works are proposed to perform zero-shot planning (Huang et al., 2022a; Ahn et al., 2022; Huang et al., 2022b). However, their planning scenarios are limited to single-objective tasks such as household cleaning and often require step-by-step interactive planning with grounding. To improve LLMs planning capabilities for complex problems, chain-of-thought (CoT) prompting LLMs to perform step-by-step reasoning (Wei et al., 2022). Recent works also propose to aid the LLM planning with external tools (Liu et al., 2023; Guan et al., 2023; Chen et al., 2023; Li et al., 2023; Gundawar et al., 2024; Hao et al., 2024; Chen et al., 2024). For example, (Liu et al., 2023; Xie et al., 2023; Gundawar et al., 2024) leverages LLMs to translates problems into fixed formats and solve with external planners, (Li et al., 2023) prompts LLM to add short codes to existing codes to account for follow-up what-if questions, and (Gundawar et al., 2024; Chen et al., 2024) empowers LLMs to iteratively refine plans or prompts based on feedback from external task-specific critics/verifiers/humans. However, to achieve strong performance, these methods often need extensive task-specific pre-defined efforts. For example, CoT depends on task-specific examples to achieve strong performance, planning domain definition language (PDDL) domain files (Aeronautiques et al., 1998; Haslum et al., 2019) are required for (Liu et al., 2023), mixed-integer linear program (MILP) codes of current domains are necessary for (Li et al., 2023), and external constraint critics are needed for (Gundawar et al., 2024). These requirements limit their generalization capabilities to new domains.

### 2.2 LLM + SOLVER

As existing LLMs do not have the capability to perform long-horizon reasoning for complex tasks (Achiam et al., 2023; Valmeekam et al., 2022; 2023; Kambhampati et al., 2024), many works propose to take advantages of both LLMs and external solvers by combining them for reasoning

or planning. (Wu et al., 2022; He-Yueya et al., 2023; Pan et al., 2023; Ye et al., 2024) combines LLM with symbolic solvers to solve logical reasoning problems. While most logical reasoning problems are single-step satisfaction problems with clear constraint descriptions, LLMFP aims to solve complex planning problems, which could include implicit constraints not described in the task description and could be sequentially long-horizon tasks with defined actions. In addition, LLMFP proposes a general approach, which does not require task-specific examples or task-specific efforts. (Li et al., 2023) teaches LLMs to add constraints to existing MILP codes. (Li et al., 2024) asks the developer to express planning problems into automaton and plan based on it. (Manas et al., 2024) uses LLMs to translate problem into linear temporal logic and solves with optimization solver. (Liu et al., 2023; Guan et al., 2023; Zhou et al., 2024; Xie et al., 2023) leverages PDDL planner to aid the planning processes. Except for the natural language task description, they require human efforts to design solver-related specifications and task-specific examples, which is not needed for LLMFP.

## 3 LLMFP

LLMFP aims to resolve generic planning problems. For example, consider a *coffee supply chain problem*, where a coffee company sources beans from three suppliers with fixed capacity, roasts them at two facilities into dark or light coffee and ships the roasted coffee to three retail locations to fulfill their demands. Then a planning problem involves accomplishing the task at the cheapest cost.

To achieve this, LLMFP takes the following inputs from users, as shown in Figure 1 (top panels).

- **Natural Language Task Description $d$.** A natural language description that details the problem settings and the planning objective, such as the above description of the coffee problem.
- **Background Information & API $i$.** A list of background information about the tasks as well as information on APIs that the planner can use. An example of the background information for the coffee task is the variables containing specific numbers of supplier capacities, cafe demands, and costs for shipping and roasting.
- **User Query $q$.** The question that either describes the detailed initial and/or goal states or adds/-modifies existing requirements of the tasks. In the coffee planning task, one example query is *'What would be the outcome if cafe1 experienced a 23% rise in demand'*.

Example inputs for all 9 tasks can be found in Appendix A.8. Note that LLMFP does not require any task-specific examples from the users. Considering the diversity of user requests, LLMFP needs to accommodate a large variety of domains, planning problem setups, user queries, constraints, and complexity levels, which poses a great challenge.

### 3.1 OVERVIEW

Devising solutions for the vast diversity of different complex planning problems seems prohibitive even for humans, let alone LLMs. However, despite the diversity of the planning problems, a planning problem can generally be cast as a constrained optimization problem. Formally, a constrained optimization, $\mathcal{P} = \{\boldsymbol{x}, f(\cdot), \boldsymbol{g}(\cdot), \boldsymbol{h}(\cdot)\}$, is defined as $\min_{\boldsymbol{x}} f(x)$, s.t., $\boldsymbol{g}(\boldsymbol{x}) \leq 0, \boldsymbol{h}(\boldsymbol{x}) = 0$. $\boldsymbol{x}$ represents the decision variables; $f(\cdot)$ represents the objective function; $\boldsymbol{g}(\cdot)$ represents the (multiple) inequality constraints; $\boldsymbol{h}(\cdot)$ represents the (multiple) equality constraints. In the coffee supply problem, $\boldsymbol{x}$ includes the amount of coffee beans shipped from each supplier to each roastery, and from each roastery to each cafe, $f$ describes the total cost of shipping, and $\boldsymbol{g}$ and $\boldsymbol{h}$ include the clearing conditions for each facility. A detailed description of the variables and constraints for the coffee supply problem can be found in Appendix A.9.2.

Once a planning problem is formulated as the constrained optimization problem, it can be rigorously solved by solvers such as the SMT solver. Therefore, we propose an pipeline that solves the planning problem by converting them into constrained optimization problems and then solving them using the SMT solver. Our pipeline consists of the following steps, as shown in Fig. 1. ❶ **DEFINER:** LLMFP first prompts an LLM to define the constrained optimization problem from the user inputs, $\mathcal{P} = \mathcal{D}(d, i)$ (Sec. 3.2). ❷ **FORMULATOR:** LLMFP asks LLM to think about the necessary variables and steps to build when programming, and formulate a representation to summarize all key information of these variables (Sec. 3.3). ❸ **CODE GENERATOR:** Given this representation, LLMs generate codes that initialize all necessary variables, assert constraints, and add goals (Sec. 3.4).

❹ **RESULT FORMATTER:** After LLMFP executes the generated codes, it prompts LLMs to convert the execution result into a fixed format and provide a short assessment of the execution results (Sec. 3.5). ❺ **SELF ASSESSMENT AND MODIFICATION:** LLMFP assesses each step based on the execution result, and modifies the first incorrect step (Sec. 3.6). The generated plan is delivered when it passes self-assessments of all steps. Please refer to Appendix A.9 and A.10 for example outputs and prompts of all steps in LLMFP.

## 3.2 DEFINER

The first step of building an optimization problem is to identify the goal, decision variables, and constraints of the problem from the user-supplied task description $d$ and background information $i$, *i.e.*, $\mathcal{P} = \mathcal{D}(d, i)$. The definer function $\mathcal{D}$ is accomplished by prompting the LLM to express in a natural language format (See Figure 1 for an example), where the prompt skeleton includes ❶ a description of what goal, decision variables, and constraints mean and ❷ an instruction to output the aforementioned information, which is invariant across tasks. The detailed prompt is listed in Appendix A.10.2.

While generating the goals and decision variables are straightforward, generating the constraints is challenging, because certain constraints are not explicitly stated and can only be inferred by commonsense reasoning. We refer to these as the *implicit constraints*.

For example, in the coffee supply chain task example, the implicit constraints include *'the roasted coffee in each roastery does not exceed the beans it receives'*, *'the shipped coffee from each roastery does not exceed the coffee it roasts'*, and importantly but easily overlooked, *'all numbers of shipped and roasted beans and coffee need to be non-negative integer'*.

To facilitate uncovering the implicit constraints, we include in the prompts (under the description of constraints) a three-step instruction to derive the constraints: ❶ Identify all decision variables in this task, ❷ for each pair of decision variables, consider relations (explicit, implicit, underlying assumption, unmentioned commonsense) between them to make sure all variables are connected, and ❸ provide a constraint reasoning first before answering. This effectively helps LLMs to better identify implicit constraints for multi-constraint planning problems. Since for multi-step planning problems the task description needs to explicitly define the preconditions and effects of each action, there will be no implicit constraint so this step is omitted.

## 3.3 FORMULATOR

Before turning the optimization problem $\mathcal{P}$ into an executable code to run the SMT solver, the FORMULATOR is called to supplement $\mathcal{P}$ with additional information regarding each variable in $\boldsymbol{x}$ that is necessary to ensure the correctness of the generated code. Formally, the FORMULATOR is defined as $\mathcal{R} = \mathcal{F}(\mathcal{P}, d, i, q)$, where the output $\mathcal{R}$ is a JSON representation that describes $N$ fields of information for each variable in $\boldsymbol{x}$, *i.e.*, $\mathcal{R} = \bigcup_i \{\boldsymbol{x}_i : \texttt{field}_1(\boldsymbol{x}_i)\}, \ldots, \texttt{field}_N(\boldsymbol{x}_i)\}$. Examples of the JSON representation are shown in Fig. 2.

Each field describes the information that governs the declaration and instantiation of each variable in the code, such as whether the variable is continuous, binary, or integer, what data structures they need to be arranged into, *etc.* The definition of fields is different for single-step and multi-step problems. In what follows, we will describe the fields in each problem type.

**Single-Step Multi-Constraint Problem** As shown in Fig. 1 and Fig. 2, for each variable, $\mathcal{R}$ includes 6 fields to summarize the information related to this variable. The `name` field indicates the variable name. Since we are using SMT as the optimization solver, the `SMT_variable` field indicates whether the variable is an SMT or a normal variable. SMT variables are different from normal variables in that they don't hold specific values upfront, rather, they are symbolic variables to represent unknown values. The `number_of_variables` field represents the length of the variable. The `data_source` field denotes the dependencies of the variable. The `value` field further specifies the value of the variable. It could either be a real number or list, dictionary descriptions, or any description of operations to do with the data source. The `specific_requirement` field is where we point to the constraints or goals related to this variable. For the coffee supply chain task, the total cost has a length of 1, the `data_source` is all shipping costs and roasting costs, the `value` is a string *"sum of all costs"*, and the specific requirement is *"minimize"*. With this intermediate

| Example Fomulator output for multi-constraint task Coffee | Example Fomulator output for multi-step task Blocksworld |
|---|---|
| ```{ ``` | ```{ ``` |

```
{
  "variable_1": {
    "name": "beans_shipped",
    "SMT_variable": true,
    "number_of_variables": 6,
    "data_source": "capacity_in_supplier,
shipping_cost_from_supplier_to_roastery",
    "value": "amount of coffee beans shipped from
each supplier to each roastery",
    "specific_requirement": "must not exceed
supplier's capacity"
  },
  "variable_2": {
    "name": "light_roasted",
    "SMT_variable": true,
    "number_of_variables": 2,
    "data_source": "roasting_cost_light",
    "value": "amount of light coffee roasted at each
roastery",
    "specific_requirement": "total amount of coffee
beans received by each roastery must equal total
amount of coffee roasted"
  },
  …
  "variable_6": {
    "name": "total_cost",
    "SMT_variable": true,
    "number_of_variables": 1,
    "data_source": "variable_1, variable_2,
variable_3, variable_4, variable_5",
    "value": "sum of shipping, roasting, and shipping
roasted coffee costs",
    "specific_requirement": "minimize"
  },
}
```

```
{
  "objects": {
    "variable_1": {
      "name": "objects",
      "SMT_variable": false,
      "number_of_variables": 1,
      "data_source": "query",
      "value": "all objects in the problem…",
      "specific_requirement": null
    }
  },
  "predicates": {
    "variable_2": {
      "name": "on",
      "SMT_variable": false,
      "number_of_variables": 1,
      "data_source": "query, variable_1",
      "value": "a dictionary of boolean variables
representing whether a block is on another block at a
timestep…",
      "specific_requirement": initialize step 0…"
    },
    …
  },
  "actions": {
    "variable_7": {
      "name": "pickup",
      "SMT_variable": false,
      "number_of_variables": 1,
      "data_source": "variable_1",
      "value": "a dictionary of boolean variables
representing whether pickup is performed on a block…",
      "specific_requirement": null
    },
    …
```

Figure 2: Example FORMULATOR output for multi-constraint Coffee and multi-step Blocksworld.

step between DEFINER and CODE GENERATOR, LLMFP is capable of obtaining a more detailed, well-formulated, and overall coding plan. We teach LLMs to formulate variables into this representation by including two simple examples in the prompt. These two examples are not task-specific examples of any of our testing tasks and we do not modify these two examples across tasks.

**Multi-Step Planning Problem** Since multiple steps are involved in multi-step planning problems, the FORMULATOR not only needs to deal with relationships between variables, but also needs to update the states of variables across different timesteps. All the variables are divided into five sections, representing the five stages for defining the variables: objects, predicates, actions, update, goal. The variables within each section are appended with the same six fields as introduced above. The objects stage declares all objects in the scenario. The predicates stage defines the predicates, which represent the properties of objects and the relationships between them. The actions stage initializes variables to represent all actions. The update stage adds assertions to existing action variables to account for the preconditions and effects of actions. The goal stage adds constraint to existing predicate variables to encode the goal. As shown in Fig. 2, the example on the right includes different stages and variables within each stage. The variable in predicates stage initializes a dictionary to represent whether a block is on another block at a certain timestep. The variable in actionstage initializes a dictionary to represent if action pickup is performed. For multi-step planning tasks, we replace the examples with a multi-step task example, and similarly, it is not a task-specific example and we do not modify it across tasks.

The formulator function $\mathcal{F}$ is enabled by prompting an LLM, where the prompt includes ❶ A brief instruction for the FORMULATOR, ❷ Example input-output pairs of FORMULATOR as demonstrations, and ❸ The user-provided task information and DEFINER's output. The detailed prompt is listed in Appendix A.10.2. Note that although example input-output pairs are used, they are task-agnostic examples fixed for all the planning tasks. No task-specific examples are needed.

### 3.4 CODE GENERATOR

With $\mathcal{R}$, now we have all the information needed to build an optimization problem with codes. In the CODE GENERATOR's prompt, we explain the meanings of different stages and fields in $\mathcal{R}$

and ask LLMs to follow Python and Z3 SMT syntax (De Moura & Bjørner, 2008). By including user inputs and results from DEFINER and FORMULATOR, with no examples, LLM could reliably generate reasonable, executable, and correct Python codes. Then, LLMFP executes the codes and returns to re-generation if there are runtime errors. We set maximum re-generation times to be 5.

## 3.5 RESULT FORMATTER

Since the variable names are decided by LLMs and have chances to be very different across queries, after code generation, we use a RESULT FORMATTER to ask LLM to convert the execution result to a fixed output format. For example, the output for the coffee task would be a JSON includes: ❶ the number of coffee beans shipped from each supplier to each roastery, ❷ the number of light and dark coffee roasted in each roastery, and ❸ the number of light and dark coffee shipped from each roastery to each cafe. After filling in this result, we prompt the LLM to provide a brief evaluation of the result based on whether the result achieves the goal, satisfies constraints, and makes sense in common sense. Taking commonsense into consideration is important because sometimes if a necessary constraint is missing from the DEFINER step, it could result in unreasonable execution result that is unrealistic in commonsense. For example, for the coffee task, if the DEFINER does not include the non-negative constraint, to minimize the cost, the solver could propose negative units of shipped coffee. Detecting these unrealistic plans is helpful for SELF ASSESS & MODIFICATION.

## 3.6 SELF ASSESS & MODIFICATION

With the execution result and evaluation, LLMFP perform self-assessment to reason about the correctness and provide ratings for the DEFINER, FORMULATOR, and CODE GENERATOR. If the assessment marks all three steps to be correct, this plan will be delivered as the final output. Otherwise, the assessment will reason about how to modify this step, and provide a modification by itself. This modification will replace the output of the incorrect step and LLMFP will loop back to continue the next steps from there again. That is, if the SELF ASSESSOR thinks the FORMULATOR output is incorrect, it will generate a JSON representation $\mathcal{R}'$ by itself, and the framework will use this $\mathcal{R}'$ to enter CODE GENERATOR again. We set the maximum number of loops to be 5.

## 3.7 CHOICE OF SOLVER

As a framework that formulates and solves planning problems as optimization problems, LLMFP can be adapted to use any planner or solver by modifying the requirements in prompt to follow the syntax of new solvers. In this work, we compare with popular PDDL and MILP solvers and choose SMT solver with following reasons: SMT allows explicit goal and constraint assertion from scratch, which can solve both single-step multi-constraint problems and multi-step problems, while PDDL solvers require rigidly formatted PDDL domain and problem files, which limits its applicability for non-PDDL problems. SMT is complete and sound, guaranteeing optimal plans, while PDDL planners lack completeness guarantees. Additionally, for all optimization solvers like SMT and MILP, building optimization problems involves the same steps: defining the goal, constraints, and decision variables, and writing codes to encode relationships between decision variables. Thus, utilizing any optimization planner has a similar process. We show how easily our framework could adapt to use MILP by including prompt differences and output examples in Appendix A.10.3. We selected SMT over MILP because the SMT Z3 solver is publicly available and more accessible to all users than the Gurobi MILP solver, which requires licenses and limits the number of devices per license.

# 4 EXPERIMENTAL RESULTS

## 4.1 DOMAINS

We test on 9 planning problems, which includes 5 multi-constraint decision making tasks,**Coffee**, **Workforce**, **Facility**, **Task Allocation**, and **Warehouse**, and 4 multi-step tasks, **Blocksworld**, **Mystery Blocksworld**, **Movie**, and **Gripper** (Li et al., 2023; Valmeekam et al., 2024; Stein & Koller, 2023). Task descriptions and complexity analysis are included in Appendix A.1. The queries are either what-if questions that change/add constraints to the existing scenarios or different task initial and goal conditions. Task inputs including example queries are given in Appendix A.8.

Table 1: Optimal rate (%) comparison of LLMFP with baselines on 5 multi-constraint problems.

| Method | Coffee | Workforce | Facility | Task Allocation | Warehouse | Average |
|---|---|---|---|---|---|---|
| Direct$_{\text{GPT-4O}}$ | 0.8 | 2.6 | 0.0 | 0.0 | 0.0 | 0.7 |
| Direct$_{\text{O1-PREVIEW}}$ | 25.9 | 47.6 | 4.8 | 4.0 | 66.0 | 29.7 |
| CoT$_{\text{GPT-4O}}$ | 0.0 | 5.6 | 0.0 | 0.0 | 16.0 | 4.3 |
| Code$_{\text{GPT-4O}}$ | 17.7 | 75.8 | 53.9 | 0.0 | 8.0 | 31.1 |
| Code_SMT$_{\text{GPT-4O}}$ | 0.0 | 10.8 | 0.6 | 0.0 | 2.0 | 2.7 |
| LLMFP $_{\text{GPT-4O}}$ | **64.7** | **92.2** | **70.7** | **96.0** | **72.0** | **79.1** |
| Direct$_{\text{CLAUDE 3.5 SONNET}}$ | 0.0 | 0.0 | 0.0 | 0.0 | 0.0 | 0.0 |
| CoT$_{\text{CLAUDE 3.5 SONNET}}$ | 7.1 | 0.0 | 0.0 | 0.0 | 14.0 | 4.2 |
| Code$_{\text{CLAUDE 3.5 SONNET}}$ | 59.8 | 71.9 | 47.3 | 0.0 | 42.0 | 44.2 |
| Code_SMT$_{\text{CLAUDE 3.5 SONNET}}$ | 75.6 | 36.8 | **49.7** | 86.0 | 64.0 | 62.4 |
| LLMFP $_{\text{CLAUDE 3.5 SONNET}}$ | **80.5** | **88.7** | 48.2 | **96.0** | **90.0** | **80.7** |

Table 2: Optimal rate (%) comparison of LLMFP with baselines on 4 multi-step problems.

| Method | Blocksworld | Mystery Blocksworld | Movie | Gripper | Average |
|---|---|---|---|---|---|
| Direct$_{\text{GPT-4O}}$ | 41.5 | 0.8 | 85.7 | 0.0 | 32.0 |
| Direct$_{\text{O1-PREVIEW}}$ | 88.4 | 31.9 | **100.0** | 52.0 | 68.1 |
| CoT$_{\text{GPT-4O}}$ | 39.9 | 2.7 | 81.0 | 0.0 | 30.9 |
| Code$_{\text{GPT-4O}}$ | 0.0 | 0.3 | 0.0 | 0.0 | 0.1 |
| Code_SMT$_{\text{GPT-4O}}$ | 0.0 | 0.0 | 0.0 | 4.0 | 1.0 |
| LLMFP $_{\text{GPT-4O}}$ | **96.2** | **77.7** | **100.0** | **76.0** | **87.5** |
| Direct$_{\text{CLAUDE 3.5 SONNET}}$ | 43.2 | 0.5 | **100.0** | 12.0 | 38.9 |
| CoT$_{\text{CLAUDE 3.5 SONNET}}$ | 52.8 | 2.8 | **100.0** | 28.0 | 45.9 |
| Code$_{\text{CLAUDE 3.5 SONNET}}$ | 0.0 | 0.0 | 0.0 | 0.0 | 0.0 |
| Code_SMT$_{\text{CLAUDE 3.5 SONNET}}$ | 0.0 | 0.0 | 0.0 | 0.0 | 0.0 |
| LLMFP $_{\text{CLAUDE 3.5 SONNET}}$ | **93.0** | **98.0** | **100.0** | **76.0** | **91.8** |

## 4.2 LLMFP PERFORMANCE

We evaluate LLMFP on 9 tasks with GPT-4o (gpt) and Claude 3.5 Sonnet (cla) with temperature 0. Each task comes with natural language task descriptions, queries, background information, APIs. LLMFP takes these inputs and outputs plans with no task-specific examples. We use optimal rate as the evaluation metric, measuring whether plans are optimal for given the task and query. We also include success rate as another metric and results in A.3.

**Baselines** We compare LLMFP against 1) Direct: LLM direct plan generation, 2) CoT: chain-of-thought prompting (Wei et al., 2022) by asking LLMs to reason before generating the final answer, 3) Code: prompts LLM to generate Python codes to solve the problem, allowing the use of any package or solver, and 4) Code_SMT: prompts LLM to generate Python codes using Z3 SMT solver, the same tool we use in LLMFP. For all baselines, we use both GPT-4o and Claude 3.5 Sonnet and also include a direct plan generation baseline with OpenAI o1-preview (o1p). All baselines are zero-shot with no task-specific examples. All baselines have the same input information as LLMFP, including task description, task background information or info collection API, and query. We also provide all baselines with formatters to convert their generated plans to fixed formats for better evaluation. Please refer to Sec. A.10 for prompts of baselines.

**Results and Analysis** We include the optimal rate comparison of LLMFP and baselines on 5 multi-constraint problems and 4 multi-step problems in Table 1 and 2. There are four key takeaways:

First, LLMFP achieves strong performance across all 9 tasks, significantly outperforming all baselines. For GPT-4o, LLMFP achieves an average of 83.7% optimal rate across 9 tasks (79.1% for 5 multi-constraint problems and 87.5% for 4 multi-step problems). For Claude 3.5 Sonnet, LLMFP achieve an 86.8% optimal rate across 9 tasks (80.7% for 5 multi-constraint problems and 91.8% for 4 multi-step problems). For 5 multi-constraint problems, LLMFP $_{\text{GPT-4O}}$ and LLMFP $_{\text{CLAUDE 3.5 SONNET}}$

outperform best baselines $Code_{GPT-4O}$ and $Code\_SMT_{CLAUDE\ 3.5\ SONNET}$ by a large margin of 48.0% and 18.3%. For 4 multi-step problems, $LLMFP_{GPT-4O}$ and $LLMFP_{CLAUDE\ 3.5\ SONNET}$ outperform $Direct_{O1-PREVIEW}$ and $CoT_{CLAUDE\ 3.5\ SONNET}$ by an average of 19.4% and 45.9%. This highlights both the effectiveness and the generalization capability of LLMFP.

Second, among baselines, Code works better for multi-constraint problems, while Direct and CoT work better for multi-step problems. This validates that the skills required for solving different tasks are different. For multi-constraint problems, as heavy calculations are required to test every possible solution, it is hard for LLMs to directly plan, even with the strongest o1-preview model. For multi-step problems, since Code tries to use a PDDL planner, which requires LLM to generate fixed-format PDDL domain and problem files, it almost always fails to generate and call them correctly. While it is easier for LLMs to directly devise plans as the preconditions and effects of each action are easier to reason about than calculations. This further proves that LLMFP can tackle problems that are fundamentally different because it uses a universal and formal approach for all tasks.

Third, for Direct and CoT, Mystery Blocksworld's performance degrades largely compared to Blocksworld, though they are fundamentally same problems. Changing predicate and action names to illogical names makes LLMs hard to understand the problem and generate reasonable plans. However, LLMFP still can obtain an overall strong optimal rate of 77.7% and 98.0% on Mystery Blocksworld for GPT-4o and Claude 3.5 Sonnet. This shows LLMFP is robust to obfuscated problems, as it can encode the problem as long as the problem is clearly defined regardless of the names.

Fourth, for multi-constraint problems, $Code\_SMT_{CLAUDE\ 3.5\ SONNET}$ improves 18.2% compared to $Code_{CLAUDE\ 3.5\ SONNET}$, though $Code\_SMT_{GPT-4O}$ performs poorly. This showcases the strong coding capability of Claude 3.5 Sonnet, especially the capability to understand and utilize the SMT solver. At the same time, it showcases the instability for different LLMs to reach strong performance, motivating the need of frameworks like LLMFP to overcome existing limitations of LLMs.

To summarize, LLMFP is capable of solving all 9 tasks with strong performance and is robust to fundamentally different and obfuscated problems. We show the performance of LLMFP across iterations, time and cost statistics, and failure analysis in Appendix A.2, A.4, and A.6.

## 4.3 EFFECTIVENESS OF LLMFP COMPONENTS

We then validate each component of LLMFP with ablation experiments on 9 tasks. We examine the effectiveness of DEFINER, FORMULATOR, and SELF ASSESS & MODIFICATION by removing these components from our framework one at a time and comparing with LLMFP. We do not remove CODE GENERATOR and RESULT FORMATTER because they are the necessary components of LLMFP to deliver outputs. We use GPT-4o as the LLMs and optimal rate as the evaluation metric.

**Results and Analysis** We include the optimal rate performance comparison of LLMFP and baselines on 9 problems in Table 3. From Table 3 there are two key takeaways:

First, removing any of the 3 components from LLMFP negatively affects the performance. For multi-constraint problems, removing DEFINER, FORMULATOR, and SELF ASSESS & MODIFICATION lowers the optimal rate by 15.4%, 22.2%, and 21.9%. For multi-step problems, removing FORMULATOR and SELF ASSESS & MODIFICATION reduces the optimal rate by 87.4% and 12.4%.

Second, for different problems, the most effective components are different. Coffee degrades the most for No DEFINER; Warehouse and all multi-step problems drop the most for No FORMULATOR; and Workforce decreases the most for No SELF ASSESS & MODIFICATION. This again validates the diversity of the 9 problems and how they require different efforts to be successfully solved. Thus, LLMFP is an overall framework that could aid the process of planning from all aspects.

To summarize, all three components in LLMFP are effective and could account for diverse problems by providing comprehensive aids to solve planning problems.

## 4.4 LLMFP WITH TASK-SPECIFIC EXAMPLE

Although LLMFP is capable of achieving strong performance on a wide range of problems with no task-specific example, we test LLMFP by only replacing the two examples in FORMULATOR to one task-specific example on Coffee task to see how much the task-specific example could further

Table 3: Optimal rate (%) comparison when removing some key components of LLMFP on all 9 tasks. LLMs used are GPT-4o.

| Domain | No Definer | No Formulator | No Self Assess & Modification | LLMFP |
|---|---|---|---|---|
| Coffee | 8.6 | 56.4 | 55.3 | **64.7** |
| Workforce | 84.4 | 80.5 | 27.3 | **92.2** |
| Facility | 61.6 | 53.7 | 53.7 | **70.7** |
| Task Allocation | 74.0 | 92.0 | 96.0 | **96.0** |
| Warehouse | **90** | 2.0 | 54.0 | 72.0 |
| Average | 63.7 | 56.9 | 57.2 | **79.1** |
| Blocksworld | N/A | 0.2 | 95.3 | **96.2** |
| Mystery Blocksworld | N/A | 0.0 | 74.4 | **77.7** |
| Movie | N/A | 0.0 | 66.7 | **100.0** |
| Gripper | N/A | 0.0 | 64.0 | **76.0** |
| Average | N/A | 0.1 | 75.1 | **87.5** |

Table 4: Optimal rate (%) comparison of LLMFP and LLMFP with one task-specific example in Formulator on **Coffee** task. Sets represent different types of what-if questions. LLMs are GPT-4o.

| Method | Set 1 | Set 2 | Set 3 | Set 4 | Set 5 | Set 6 | Set 7 | Average |
|---|---|---|---|---|---|---|---|---|
| LLMFP | 58.3 | 70.9 | 11.8 | 42.4 | 80.0 | 83.3 | 81.5 | 61.2 |
| LLMFP $_{\text{TASK-SPECIFIC}}$ | 78.3 | 72.7 | 70.6 | 84.8 | 91.4 | 100.0 | 100.0 | 85.4 |

improve LLMFP. Queries of Coffee tasks are what-if questions and are categorized into 7 sets. Each set is a type of question. For example, every query in Set 1 asks about what if the demand in some cafes increases some amount. For each set, we include one task-specific example in FORMULATOR prompt, test LLMFP $_{\text{TASK-SPECIFIC}}$ over this set, and compare with LLMFP over this set. We use GPT-4o for LLMs. From the results in Table 4, we observe that on average, LLMFP $_{\text{TASK-SPECIFIC}}$ improves the performance of LLMFP by 24.2%. The performance of Set 3 increases the most. This set asks queries like *"What led to the decision to use supplier3 for the roasting facility at roastery1?"*. It is both plausible to test *"using supplier3"* or to test *"not using supplier3"* to answer the question. However, the ground truth answer for this type of question is to *"not using supplier 3"*. They are confusing queries even for humans to understand, let alone LLMs. Thus, for these queries, adding task-specific examples significantly improves the performance. To summarize, LLMFP can achieve strong performance with no task-specific example, but easily adding task-specific examples only to FORMULATOR improves the performance, especially when the task or query is not clearly presented.

## 5 CONCLUSION

To account for the challenging trade-off between flexibility and task complexity for LLM planning, we observe that the core of many planning problems lies in optimization problems and propose a universal approach for LLMs to solve planning problems. We propose LLMFP, a general-purpose LLM-based framework that captures key information from planning problems and formally formulates and solves them as optimization problems, with no task-specific examples needed. We test LLMFP on 9 diverse planning tasks with two LLMs, proving LLMFP can achieve strong performance over fundamentally different tasks and showing the effectiveness of components in LLMFP.

**Limitations** LLMFP needs clear and detailed task descriptions and queries. It is hard for LLMFP to define the problems' goals and constraints if the task description is ambiguous or missing some important information. In addition, since LLMFP solves encoded planning problems with optimization solvers, the capability of LLMFP depends on the strength of the solvers. For massive databases with numerous feasible plans, the speed for solvers to search for optimal plans is slow. Ways to mitigate this is to introduce heuristics to prioritize a portion of the choices or to switch from solving optimization problems to satisfaction problems for planning tasks that do not require optimality.

## 6 ACKNOWLEDGMENTS

This work was supported by ONR under Award N00014-22-1-2478 and MIT-IBM Watson AI Lab. However, this article solely reflects the opinions and conclusions of its authors.

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

PLANNING ANYTHING WITH RIGOR: GENERAL-PURPOSE ZERO-SHOT
PLANNING WITH LLM-BASED FORMALIZED PROGRAMMING

# A APPENDIX

## A.1 DOMAINS AND COMPLEXITY ANALYSIS

We test on 9 planning problems, including 5 multi-constraint decision making tasks and 4 multi-step tasks (Li et al., 2023; Valmeekam et al., 2024; Stein & Koller, 2023):

- **Coffee** Coffee company sources beans from three suppliers with fixed capacity, roasts them at two facilities into dark or light coffee, and ships the roasted coffee to three retail locations. The company aims to minimize the total shipping and roasting cost while fulfilling the demand at each retail location. There are 266 different queries of 7 types in the dataset.

- **Workforce** Assign workers to shifts; each worker may or may not be available on a particular day. The goal is to minimize the total payments to workers while fulfilling the shift requirements for two weeks. There are 231 different queries of 5 types in the dataset.

- **Facility** A company currently ships its product from 5 plants to 4 warehouses. It is considering closing some plants to reduce costs. The goal is to decide which plant(s) to close to minimize transportation and fixed costs. There are 165 different queries of 4 types in the dataset.

- **Task Allocation** Given tasks and three robots skilled in different tasks, the goal is to assign tasks to robots to minimize finish time. The finish time counts when the last robot stops working. There are 50 different queries describing the number of different tasks. This task and its data and queries are created by us.

- **Warehouse** The robots need to finish tasks by visiting stations that are capable of accomplishing corresponding tasks. The goal is to find the list of stations while minimizing the total distance traveled. There are 50 different queries that include the random-length list of tasks to finish. This task and its data and queries are created by us.

- **Blocksworld** The robot has four actions: pickup, putdown, stack, and unstack. The goal is to stack the blocks in the scene from their initial setup to a specific order with minimum steps. There are 602 different queries that describe blocks' initial conditions and goal states.

- **Mystery Blocksworld** An obfuscated version of Blocksworld. The action and predicate names are replaced with names that logically make no sense. There are 602 different queries that describe objects' initial conditions and goal states.

- **Movie** The goal is to get the required snacks, watch the movie, and recover the movie and counter to the original state with minimum steps. There are 21 different queries that describe objects' initial conditions and goal states.

- **Gripper** There are robots and balls in different rooms. Each robot, with two grippers, can pick, drop, and move balls between rooms. The goal is to place balls in specific rooms with minimum steps. There are 25 different queries describing objects' initial conditions and goal states.

The queries for Coffee, Workforce, and Facility are what-if questions that change or add constraints to the existing scenarios. The queries of the rest tasks are different task initial and goal conditions. Task inputs including example queries are given in Appendix A.8.

**Mathematical Representation** We use the benchmark from (Li et al., 2023) for the first 3 problems (Coffee, Workforce, and Facility), in which these 3 problems are built as Mixed-integer linear programming (MILP) problems. As an example, here is the problem definition of Coffee as a MILP problem (Defined in Page 6-7 from (Li et al., 2023)):

$x_{s,r}$ is the number of units purchased from supplier $s$ for roasting facility $r$, and $y_{r,\ell}^L$ and $y_{r,\ell}^D$ is the amount of light and dark roast sent to retail location $\ell$ from roasting facility $r$. $C_s$ is the capacity for each supplier $s$, and $D_\ell^L$ and $D_\ell^D$ are demand for light and dark roast for each retail location $\ell$. There is a cost $c_{s,r}$ for each unit purchased from supplier $s$ for roasting facility $r$, a shipping cost of $g_{r,\ell}$ for each unit sent to retail location $\ell$ from roasting facility $r$, and a roasting cost $h_r^L$ and $h_r^D$ per

unit of light roast and dark roast respectively in facility $r$.

$$\text{minimize} \quad \left( \sum_{s,r} x_{s,r} \cdot c_{s,r} + \sum_{r,\ell} y_{r,\ell}^L \cdot h_{r,\ell}^L + \sum_{r,\ell} y_{r,\ell}^D \cdot h_r^D + \sum_{r,\ell} \left( y_{r,\ell}^L + y_{r,\ell}^D \right) \cdot g_{r,\ell} \right)$$

subject to

$$\sum_r x_{s,r} \leq C_s, \quad \forall s \quad \text{(Supplier capacity constraint)}$$

$$\sum_s x_{s,r} = \sum_\ell \left( y_{r,\ell}^L + y_{r,\ell}^D \right), \quad \forall r \quad \text{(Conservation of flow constraint)}$$

$$\sum_r y_{r,\ell}^L \geq D_\ell^L, \quad \forall \ell \quad \text{(Light coffee demand constraint)}$$

$$\sum_r y_{r,\ell}^D \geq D_\ell^D, \quad \forall \ell \quad \text{(Dark coffee demand constraint)}$$

$$x_{s,r}, y_{r,\ell}^L, y_{r,\ell}^D \in \mathbb{Z}^+, \quad \forall s, r, \ell \quad \text{(Integrality constraint)}$$

**Complexity Analysis**

The Coffee problem can be framed as a max-flow problem, which can be solved in polynomial time. Specifically, some algorithms can solve the max-flow problem with $O(VE)$ or $O(V^2 E)$

The Workforce problem, with no additional constraint, can also be framed as a max-flow problem. However, different types of constraints are added by users to form different instances. Some types of queries can increase the complexity. For example, "What if Gu and Bob cannot work on the same day?". Adding constraints to introduce conflicting workers turns the problem to be as hard as a maximum independent set problem(also NP-Hard), where we add an edge between conflicted workers, and finding the maximum indentpendent set.

The facility problem is a NP-Hard problem Capacited Facility Location Problem(CFLP). The formal definition of CFLP is as below:

$$\min \sum_{i=1}^n \sum_{j=1}^m c_{ij} d_j y_{ij} + \sum_{i=1}^n f_i x_i$$

$$\text{s.t.} \quad \sum_{i=1}^n y_{ij} = 1 \quad \text{for all } j = 1, \ldots, m$$

$$\sum_{j=1}^m d_j y_{ij} \leq u_i x_i \quad \text{for all } i = 1, \ldots, n$$

$$y_{ij} \geq 0 \quad \text{for all } i = 1, \ldots, n \text{ and } j = 1, \ldots, m$$

$$x_i \in \{0, 1\} \quad \text{for all } i = 1, \ldots, n$$

where $x_i = 1$ if facility $i$ is open, and $x_i = 0$ otherwise. $y_{ij}$ for $i = 1, \ldots, n$ and $j = 1, \ldots, m$, which represents the fraction of the demand $d_j$ filled by facility $i$.

For the Task Allocation problem, since it is equivalent to a multi-agent traveling salesman problem(agent=robots, tasks=cities), it reduces from the classic traveling salesman problem (TSP) and thus is also NP-hard.

For the Warehouse problem, as TSP is a special case when one station can be used to finish one specific task and there are no extra stations, the Warehouse problem is at least as complex as TSP, thus is also NP-hard.

For multi-step problems, Blocksworld is proved to be a NP-hard problem (Gupta & Nau, 1992), so same for Mystery Blocksworld as it is the same problem with obfuscated names. Although there is no existing proof, Movie has 13 predicates and 9 possible actions, and Gripper has 4 types of objects (rooms, objects, robots, grippers), 4 predicates, and 3 possible actions. These show that they are not simple straightforward tasks.

**LLMFP Performance on Sokoban** To further test capability of LLMFP on even more challenging tasks, we tested LLMFP on the Sokoban environment, a NP-Hard problem with large maps thus needs more variables. We created an evaluation set containing 15 queries describing the game setup and goals with different map sizes and number of boxes. We have five queries with 5x5 maps and 1 box, five queries with 6x6 maps and 1 box, and five queries with 5x5 maps and 2 boxes. The evaluation results are presented in the following table.

Table 5: Optimal rates (%) comparison of LLMFP with baselines on Sokoban task.

| Direct$_{\text{GPT-4o}}$ | Direct$_{\text{o1-PREVIEW}}$ | CoT$_{\text{GPT-4o}}$ | Code$_{\text{GPT-4o}}$ | Code_SMT$_{\text{GPT-4o}}$ | LLMFP $_{\text{GPT-4o}}$ |
|---|---|---|---|---|---|
| 0.0 | 26.7 | 0.0 | 0.0 | 0.0 | 80.0 |

As can be observed, LLMFP achieves a success rate of 80%, outperforming the baselines. The new results, along with other problems, showcase the potential of LLMFP to solve complex problems.

The major failure mode is: when the generated codes initialize the adjacent predicate, it only initializes adjacent positions to be True but fails to initialize unmentioned positions to be False (since the query only mentions position_x and position_y are adjacent), so the solver would set the non-adjacent positions to adjacent to get solution with fewer steps. In addition, although LLMFP is demonstrated to be capable of correctly encoding and solving the Sokoban problem, it is true that there are many more variables in the Sokoban problem than in other tasks because the problem is represented with a map with a large number of different positions. This slows down the speed of the SMT solver. To mitigate this problem, some potential solutions include 1) introducing methods to estimate the lower and upper bounds of step numbers needed and start from there, 2) developing heuristics to prioritize some possible options first, and 3) developing methods that put attention on a part of the map and ignore the unnecessary positions in the map. We would love to extend our work to explore these directions to make our framework more efficient.

## A.2 LLMFP Performance over Iterations

Fig. 3 shows the performance of LLMFP over 5 iterations. The key observation is: number of iterations of Self Assess & Midification stage enables LLMFP to further improve the optimal rates, although we can observe that LLMFP does not need extensive iterations to achieve an overall satisfying performance.

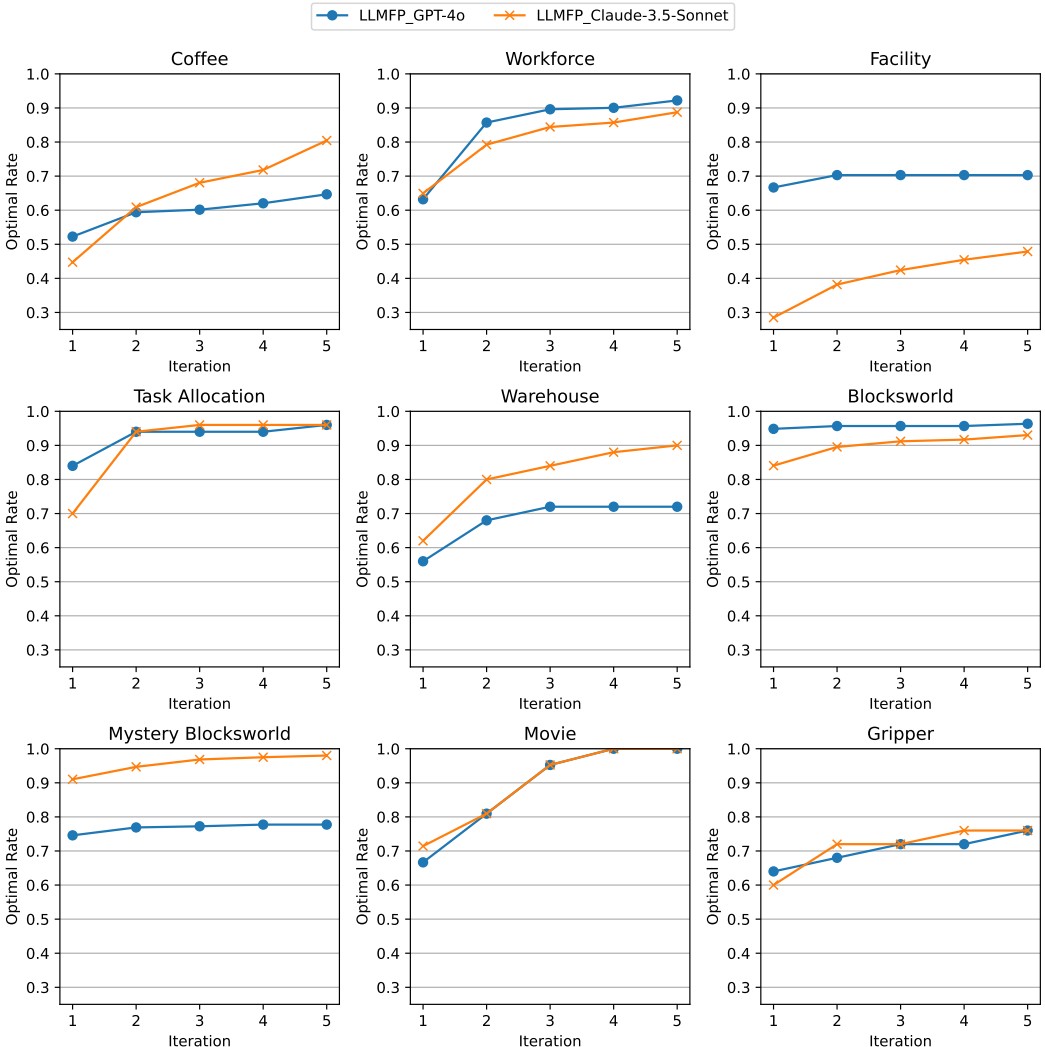

Figure 3: Optimal rates of models across LLMFP Iterations

A.3  ADDITIONAL METRIC PERFORMANCE: SUCCESS RATE

In addition to the optimal rate, we also include another metric, success rate to evaluate baselines and LLMFP. We include the result in Table 6 and Table 7.

Note that although for multi-constraint problems the optimization goal is described in the task description, we exclude the optimization goal when calculating the success rate and only evaluate whether the plan fulfills the task setup and the query. This would largely decrease the difficulty of multi-constraint problems. For example, even assigning all tasks to one robot is considered a success for the task allocation task. Thus, the success rates of all baselines for multi-constraint problems are significantly higher than the optimal rates. However, although the success rates of LLMFP almost remains the same as optimal rates since the SMT solver guarantees to output the optimal result with correct encoding, the performance of LLMFP still outperforms other baselines, with an average of 86.4%, 18.1% higher than the best baseline.

While for the multi-step problems, considering all initial conditions, predicate and action definitions, and goals are the same, developing a reasonable and correct plan is not significantly easier than developing an optimal plan with the least number of steps. Thus, the success rates of baselines are improved, but not significantly, compared to the optimal rates.

Table 6: Success rate (%) comparison of LLMFP with baselines on 5 multi-constraint problems.

| Method | Coffee | Workforce | Facility | Task Allocation | Warehouse | Average |
|---|---|---|---|---|---|---|
| Direct$_{\text{GPT-4O}}$ | 5.6 | 54.5 | 31.7 | **100.0** | 42.0 | 46.8 |
| Direct$_{\text{O1-PREVIEW}}$ | 26.3 | **92.6** | 41.5 | 94.0 | 86.0 | 68.1 |
| CoT$_{\text{GPT-4O}}$ | 17.7 | 72.3 | 31.7 | **100.0** | 82.0 | 60.7 |
| Code$_{\text{GPT-4O}}$ | 18.8 | 76.2 | 64.6 | 92.0 | 90.0 | 68.3 |
| Code_SMT$_{\text{GPT-4O}}$ | 0.0 | 10.8 | 1.2 | 0.0 | 34.0 | 9.2 |
| LLMFP $_{\text{GPT-4O}}$ | **64.7** | 92.2 | **79.3** | **100.0** | **96.0** | **86.4** |
| Direct$_{\text{CLAUDE 3.5 SONNET}}$ | 5.3 | **91.3** | 36.0 | **100.0** | 76.0 | 61.7 |
| CoT$_{\text{CLAUDE 3.5 SONNET}}$ | 10.9 | 60.6 | 1.2 | **100.0** | 96.0 | 53.7 |
| Code$_{\text{CLAUDE 3.5 SONNET}}$ | 61.3 | 89.2 | 59.1 | **100.0** | 60.0 | 73.9 |
| Code$_{\text{CLAUDE 3.5 SONNET}}$ | 77.1 | 39.0 | 59.1 | 90.0 | 74.0 | 67.8 |
| LLMFP $_{\text{CLAUDE 3.5 SONNET}}$ | **80.5** | 88.7 | **61.6** | **100.0** | **92.0** | **84.6** |

Table 7: Success rate (%) comparison of LLMFP with baselines on 4 multi-step problems.

| Method | Blocksworld | Mystery Blocksworld | Movie | Gripper | Average |
|---|---|---|---|---|---|
| Direct$_{\text{GPT-4O}}$ | 56.1 | 1.0 | 90.5 | 16.0 | 40.9 |
| Direct$_{\text{O1-PREVIEW}}$ | 90.9 | 37.9 | **100.0** | **76.0** | 76.2 |
| CoT$_{\text{GPT-4O}}$ | 62.0 | 3.0 | 95.2 | 10.0 | 42.5 |
| Code$_{\text{GPT-4O}}$ | 0.0 | 0.3 | 0.0 | 0.0 | 0.1 |
| Code_SMT$_{\text{GPT-4O}}$ | 0.2 | 0.0 | 0.0 | 4.0 | 1.0 |
| LLMFP $_{\text{GPT-4O}}$ | **96.2** | **77.7** | **100.0** | 76.0 | **87.5** |
| Direct$_{\text{CLAUDE 3.5 SONNET}}$ | 54.5 | 0.5 | **100.0** | 56.0 | 52.7 |
| CoT$_{\text{CLAUDE 3.5 SONNET}}$ | 76.1 | 3.2 | **100.0** | 72.0 | 62.8 |
| Code$_{\text{CLAUDE 3.5 SONNET}}$ | 0.0 | 0.0 | 0.0 | 0.0 | 0.0 |
| Code_SMT$_{\text{CLAUDE 3.5 SONNET}}$ | 0.0 | 0.0 | 4.0 | 0.0 | 1.0 |
| LLMFP $_{\text{CLAUDE 3.5 SONNET}}$ | **93.4** | **98.0** | **100.0** | **76.0** | **91.8** |

A.4   TIME AND COST STATISTICS AND ANALYSIS

Table 8 and Tabel 9 show the wall time comparison of all methods for GPT-4o on 9 tasks. From the results, we could observe that the time taken for LLMFP, although longer than most of the baselines, is within reasonable ranges. Especially, for multi-constraint problems, it is shorter than Direct with o1-preview because of the inherent difficulty for LLMs to solve these combinatorial optimization problems.

Table 10 shows the detailed time statistics of all components of LLMFP for GPT-4o on 9 tasks. We could observe that for both LLM querying time and solver running time, all stages of LLMFP requires reasonable runtime. The longest runtime is prompting FORMULATOR it is time-consuming to reason about all needed variables and information to form representation formulation.

Table 8: Average wall time (s) per query comparison for 5 multi-constraint problems with GPT-4o.

| Method | Coffee | Workforce | Facility | Task Allocation | Warehouse | Average |
|---|---|---|---|---|---|---|
| Direct$_{\text{GPT-4o}}$ | 8.8 | 2.2 | 2.1 | 1.8 | 0.9 | 3.2 |
| Direct$_{\text{o1-PREVIEW}}$ | 104.2 | 63.9 | 77.7 | 70.5 | 63.7 | 76.0 |
| CoT$_{\text{GPT-4o}}$ | 16.9 | 12.0 | 6.0 | 9.6 | 7.4 | 10.4 |
| Code$_{\text{GPT-4o}}$ | 30.6 | 10.0 | 8.2 | 5.7 | 7.1 | 12.3 |
| Code_SMT$_{\text{GPT-4o}}$ | 30.0 | 15.3 | 10.3 | 15.0 | 8.3 | 15.8 |
| LLMFP $_{\text{GPT-4o}}$ | 87.1 | 55.1 | 29.9 | 62.3 | 28.9 | 52.7 |

Table 9: Average wall time (s) per query comparison for 4 multi-step problems with GPT-4o.

| Method | Blocksworld | Mystery Blocksworld | Movie | Gripper | Average |
|---|---|---|---|---|---|
| Direct$_{\text{GPT-4o}}$ | 0.7 | 0.7 | 0.5 | 8.8 | 2.7 |
| Direct$_{\text{o1-PREVIEW}}$ | 26.3 | 87.9 | 25.7 | 23.8 | 40.9 |
| CoT$_{\text{GPT-4o}}$ | 2.1 | 4.0 | 1.0 | 10.2 | 4.3 |
| Code$_{\text{GPT-4o}}$ | 19.7 | 8.9 | 7.3 | 8.2 | 11.0 |
| Code_SMT$_{\text{GPT-4o}}$ | 9.1 | 8.5 | 10.6 | 12.9 | 10.3 |
| LLMFP $_{\text{GPT-4o}}$ | 43.3 | 48.3 | 58.6 | 141.6 | 73.0 |

Table 10: Average time (s) spent per query for all components of LLMFP $_{\text{GPT-4o}}$ on all 9 tasks.

| Domain | Definer | Formulator | Solver | Formatter | Code Gen. | Self Assess & Mod. |
|---|---|---|---|---|---|---|
| Coffee | 5.6 | 10.8 | 17.1 | 0.1 | 14.2 | 11.3 |
| Workforce | 3.4 | 5.1 | 8.3 | 11.0 | 3.1 | 7.8 |
| Facility | 3.8 | 7.5 | 6.4 | 0.7 | 4.3 | 4.0 |
| Task Allocation | 8.6 | 23.8 | 5.2 | 0.2 | 6.5 | 5.9 |
| Warehouse | 3.9 | 3.1 | 6.2 | 0.2 | 4.1 | 3.3 |
| Blocksworld | N/A | 21.0 | 14.6 | 0.6 | 1.9 | 3.4 |
| Mys. Blocksworld | N/A | 24.3 | 14.6 | 0.6 | 2.3 | 4.1 |
| Movie | N/A | 21.2 | 9.9 | 0.3 | 2.1 | 10.6 |
| Gripper | N/A | 18.3 | 16.0 | 6.9 | 11.2 | 7.0 |

Table 11 shows the average cost comparison of all methods on the coffee task, and Table 12, and 13 shows the cost statistics of LLMFP over all 9 tasks. We observe that although LLMFP is more costly than most of the baselines, it is cheaper than Direct with o1-preview with better performance. In addition, the average cost per query for all 9 tasks is around 0.1 dollar, indicating LLMFP is not very costly.

Table 11: Average cost (\$) per query comparison of LLMFP $_{\text{GPT-4O}}$ on the Coffee task.

| Direct$_{\text{GPT-4O}}$ | Direct$_{\text{O1-PREVIEW}}$ | CoT$_{\text{GPT-4O}}$ | Code$_{\text{GPT-4O}}$ | Code_SMT$_{\text{GPT-4O}}$ | LLMFP $_{\text{GPT-4O}}$ |
|---|---|---|---|---|---|
| 0.008 | 0.536 | 0.013 | 0.023 | 0.024 | 0.139 |

Table 12: Average cost (\$) per query of LLMFP $_{\text{GPT-4O}}$ on 5 multi-constraint problems.

| Coffee | Workforce | Facility | Task Allocation | Warehouse |
|---|---|---|---|---|
| 0.139 | 0.140 | 0.083 | 0.081 | 0.085 |

Table 13: Average cost (\$) per query of LLMFP $_{\text{GPT-4O}}$ on 4 multi-step problems.

| Blocksworld | Mystery Blocksworld | Movie | Gripper |
|---|---|---|---|
| 0.122 | 0.105 | 0.131 | 0.128 |

## A.5    Baselines Failure Case Analysis

Here we describe the major failure cases for the baselines. Please refer to Appendix A.9 for example outputs.

### A.5.1    Direct, Direct$_{\text{o1-preview}}$, and CoT

For multi-constraint tasks, since they involve various constraints, intensive calculations, and numerous possible solutions, LLMs still do not have the capability to directly solve the optimal solution considering all constraints. They either fail to understand or consider some important constraints or fail to optimize the goal. Although utilizing stronger o1-preview or taking advantages of prompting techniques like CoT could result in less mistakes, the major underlying failure reasons are similar.

For multi-step tasks, the major failure cases are the failure to deliver reasonable plans considering preconditions and effects of all actions accurately.

### A.5.2    Code

For multi-constraint tasks, the major failure cases are 1) failing to consider all necessary constraints, 2) failing to consider or understand the query, and 3) overwriting the given API.

For multi-step tasks, the major failure cases are 1) failing to correctly represent the problem, including the problem setup, predicates, and actions, 2) failing to write codes with correct logic or syntax.

### A.5.3    Code_SMT

For multi-constraint tasks, the major failure cases are same as Code.

For multi-step tasks, the major failure cases are 1) Poor SMT Utilization: including failing to distinguish And and Implies, to correctly use SMT Array or Function, or to write correct SMT syntax (or Python syntax sometimes), and 2) Poor Problem Understanding: failing to initialize the initial value of unmentioned predicates (eg. when the query says blocks a, b, d are clear, codes also need to initialize c to be not clear), to assert one action per step, or to update unchanged variables for next step

### A.5.4    Theoretical Insights

Overall, LLMs are good at understanding the syntax and semantics of planning problems as optimization problems but are not good at solving optimization problems directly. Specifically, next-token prediction is fundamentally different from deterministic algorithms for optimization. There's a growing belief that next token prediction cannot truly model human thought and cannot support human-like capabilities of understanding a problem, imagining, curating, and backtracking plans before executing [1-3]. Specifically, the claim that next token predictions are "ill-suited for planning tasks" is supported by works [4-7], which tested the planning capabilities of LLMs on various planning tasks. These works empirically show that in addition to identifying patterns in language and predicting the next word in a sequence, LLMs still can not truly understand a problem and thus do not have the capability to perform intense calculations to optimize for any objectives. Thus, this is a major reason why baselines are not capable of solving the complex planning problems in our paper. However, since LLMFP teaches LLMs to build the optimization problem step by step and calls the external solver to solve for a plan, this bypasses the need to devise a plan by LLMs themselves.

To support this claim that LLMs cannot understand and solve an optimization problem, we conduct an experiment on the Coffee task that, instead of using natural language task descriptions as inputs, we directly map this Coffee task to an optimization problem and use the formal mathematical definition of this problem as the inputs to LLMs. Thus, LLMs do not need to understand the problem and find the underlying constraints, as a formal definition is given and could be directly solved.

We tested Direct with the most powerful LLM OpenAI o1-preview model on all queries of Coffee, which only achieves an optimal rate of 34.2%. Compared to its 25.9% optimal rate with natural language task description, this is not a significant improvement, given all goals and constraints are clearly formally specified in the new setting. This is consistent with the conclusion that LLMs still

cannot solve optimization problems by themselves, even given a formal representation. LLMFP enables LLMs to formalize planning problems as optimization problems. Since SMT solvers can guarantee to return correct answers given correct input, the high optimal and success rate of LLMFP indicates that LLMFP allows LLM to parse the correct syntax and semantics information of a planning problem from its natural language description to a formal mathematical description. Such translation is also non-trivial when no task-specific examples are provided. As shown in our newly added baseline approach Code_SMT as shown in Table 1 and 2, when we directly ask LLMs to translate and encode the natural language task description in an SMT format, the optimal rate is low, with an average of 2.7% and 62.4% for multi-constraint tasks, and 1.0% and 0.0% for multi-step tasks across two LLMs GPT-4o and Claude 3.5 Sonnet.

A.6 LLMFP FAILURE CASE ANALYSIS

Here we analyze the major failure cases for all 9 tasks.

A.6.1 COFFEE

There are two major failure cases for Coffee tasks:

First, some queries are not clearly presented, indicating ambiguous information. Queries of Coffee tasks are what-if questions and are categorized into 7 sets. Each set is a type of question. We notice that the type *"supply-roastery"* asks queries like *"What led to the decision to use supplier3 for the roasting facility at roastery1?"*. To answer this question, it is both plausible to test *"using supplier3"* or to test *"not using supplier3"* to see the performance. However, the ground truth answer for this type of questions is to *"not using supplier 3"*. As confusing queries even for human, they are hard for LLMs to understand. Thus, for these queries, LLMFP sometimes generate codes with opposite meanings as what is expected.

Second, sometimes LLMFP DEFINER fails to consider all implicit constraints. The most easily neglectable implicit constraints are 1) the beans roasted in each roastery do not exceed the beans it receives, and 2) the beans ship from each roastery do not exceed the coffee it roasts. When any of the two constraints are missing, to minimize the cost, the model will automatically set the shipped beans or roasted coffee to be 0, assuming the company delivers coffee without sourcing beans or roasting coffee.

A.6.2 WORKFORCE

There are two major failure cases for Coffee tasks:

First, sometimes LLMFP fails to understand the queries. Some of the queries asks questions like *'Can Gu transition from Sun14 to Sun7 for work purposes?'*. The meaning is to **force** Gu to work on Sun7 and take rest on Sun14. However, sometimes LLMFP builds variables to test both taking and not taking this transition, and returns solutions with less costs.

Second, sometimes when the solution space is large, it is hard to find the optimal solution within maximum runtime set for solver. We set the maximum solver runtime to be 15 minutes, which is exceeded when solving some hard queries.

A.6.3 FACILITY

Similarly as the first failure case of Coffee, some queries are not clearly presented. The queries are like *"What justifies the opening of plant 0?"*, which is confusing even for humans. Both opening plant 0 and closing plant 0 to report the costs make sense to answer this query. However, the ground truth meaning of this query is to close plant 0.

A.6.4 TASK ALLOCATION

LLMFP only fails one query in Task Allocation. The reason is the FORMULATOR generates wrong values for robot finish time.

A.6.5 WAREHOUSE

The major failure case for Warehouse is CODE GENERATOR overwrites the provided API `get_distance` and provide 1 as the output during Code Generation. Thus, the distance between each station is mistakenly set to be 1.

A.6.6 BLOCKSWORLD

One major failure case for Blocksworld is CODE GENERATOR fails to initialize the states of predicates correctly and thoroughly. Since the query will only meantion the predicates that are true, for example, block 1 is on block 2, but when initializing, LLMFP needs to initialize both mentioned states but also unmentioned states that are false. For example, block 2 is not on block 1. However, CODE GENERATOR sometimes fails to consider all unmentioned states.

### A.6.7 MYSTERY BLOCKSWORLD

Similarly as Blocksworld, Mystery Blocksworld has same failure case. For Mystery Blocksworld, since the predicate and action names are not meaningful, more this kind of errors are made by GPT-4o. However, Claude seems to have better reasoning capability to support it from making more these errors.

### A.6.8 MOVIE

There is no failure case for Movie.

### A.6.9 GRIPPER

The major failure case for Gripper is when the solver fails to find the solution because there are some code generation errors, the SELF ASSESS & MODIFICATION sometimes would think it is because the timestep is not enough, thus adding another loop within the original loop. However, this would result in the program to execute forever.

A.7 BASELINES WITH EXPLICIT OPTIMAL REQUIREMENTS

For all methods including LLMFP and baselines, we implicitly mention the goal of each multi-constraint task in the task description. For example, for the Coffee task, the task description *"...The company's objective is to **minimize** the total cost, including shipping beans, roasting, and shipping roasted coffee, while ensuring that all coffee produced meets or exceeds the demand at each retail location"* implicitly shows the goal is to find the plan that minimizes the total cost.

While for multi-step problems, the methods are not explicitly instructed to provide optimal solutions. Since SMT solver guarantees to find the solution if there exists one, it can rigorously show the solution does not exist for smaller timesteps and increase timestep, thus can always find the optimal solution if the formulation and generated codes are correct. This is an advantage of incorporating a complete and sound solver like SMT in our framework.

However, to better understand the capabilities of baselines, we modify the prompts to explicitly instruct them to find the optimal solution and re-evaluate them on the 4 multi-step problems. We add _Opt to the name of the baselines to represent the baselines with explicit optimal instructions.

Table 14 shows the optimal rate of baselines with explicit instruction on finding the optimal plan. Compared with Table 2, we could observe that some baselines achieve better performance (from average 0.1% to 16.4% for Code_Opt$_{GPT-4O}$, and from average 30.9% to 36.7% for CoT_Opt$_{GPT-4O}$), while some achieve slightly worse performance(average 68.1% to 67.0% for Direct_Opt$_{O1-PREVIEW}$). However, despite the changes due to the explicit instruction to find the optimal plan, LLMFP still could largely outperform all baselines.

Table 14: Optimal rate (%) comparison of LLMFP with baselines that explicitly instructed to generate optimal plans on 4 multi-step problems.

| Method | Blocksworld | Mystery Blocksworld | Movie | Gripper | Average |
|---|---|---|---|---|---|
| Direct_Opt$_{GPT-4O}$ | 35.2 | 0.8 | **100.0** | 0.0 | 34.0 |
| Direct_Opt$_{O1-PREVIEW}$ | 80.9 | 39.0 | **100.0** | 48.0 | 67.0 |
| CoT_Opt$_{GPT-4O}$ | 33.4 | 2.3 | 95.2 | 16.0 | 36.7 |
| Code_Opt$_{GPT-4O}$ | 0.0 | 3.8 | 61.9 | 0.0 | 16.4 |
| Code_SMT_Opt$_{GPT-4O}$ | 0.0 | 0.0 | 0.0 | 0.0 | 0.0 |
| LLMFP $_{GPT-4O}$ | **96.2** | **77.7** | **100.0** | **76.0** | **87.5** |
| Direct_Opt$_{CLAUDE 3.5 SONNET}$ | 40.9 | 1.5 | **100** | 20.0 | 40.6 |
| CoT_Opt$_{CLAUDE 3.5 SONNET}$ | 52.5 | 4.5 | **100** | 20.0 | 44.2 |
| Code_Opt$_{CLAUDE 3.5 SONNET}$ | 0.0 | 0.0 | 0.0 | 0.0 | 0.0 |
| Code_SMT_Opt$_{CLAUDE 3.5 SONNET}$ | 0.0 | 0.0 | 0.0 | 0.0 | 0.0 |
| LLMFP $_{CLAUDE 3.5 SONNET}$ | **93.0** | **98.0** | **100.0** | **76.0** | **91.8** |

A.8   INPUTS ON 9 TASKS

We include the inputs, which includes task description, background information or API, and example queries, for all 9 tasks in Fig. 4 - Fig. 12:

| **Coffee** |
| --- |
| **Task Description:** |
| A coffee production company sources beans from three suppliers, roasts them at one of two facilities into either dark or light coffee, and ships the roasted coffee to three retail locations. Each supplier has a limited capacity. Each roastery, with no existing inventory, can roast one unit coffee bean into one unit of dark or light coffee. The retail locations have specific demands for dark and light coffee, with no existing inventory. The company's objective is to minimize the total cost, including shipping beans, roasting, and shipping roasted coffee, while ensuring that all coffee produced meets or exceeds the demand at each retail location. |

**Background Information or API:**
capacity_in_supplier = {'supplier1': 150, 'supplier2': 50, 'supplier3': 100}
light_coffee_needed_for_cafe = {'cafe1': 20, 'cafe2': 30, 'cafe3': 40}
dark_coffee_needed_for_cafe = {'cafe1': 20, 'cafe2': 20, 'cafe3': 100}
shipping_cost_from_supplier_to_roastery = {
   ('supplier1', 'roastery1'): 5,
   ('supplier1', 'roastery2'): 4,
   ('supplier2', 'roastery1'): 6,
   ('supplier2', 'roastery2'): 3,
   ('supplier3', 'roastery1'): 2,
   ('supplier3', 'roastery2'): 7,
}
roasting_cost_light = {'roastery1': 3, 'roastery2': 5}
roasting_cost_dark = {'roastery1': 5, 'roastery2': 6}
shipping_cost_from_roastery_to_cafe = {
   ('roastery1', 'cafe1'): 5,
   ('roastery1', 'cafe2'): 3,
   ('roastery1', 'cafe3'): 6,
   ('roastery2', 'cafe1'): 4,
   ('roastery2', 'cafe2'): 5,
   ('roastery2', 'cafe3'): 2,
}
math package: function math.ceil() to round UP float to int and math.floor() to round DOWN float to int
Expect output format

**Example Query:**
Set 1: What is the potential impact of a 29% increase in demand at cafe cafe2?
Set 2: What if demand for light coffee at cafe cafe1 increased by 23%?
Set 3: Why are we using supplier supplier2 for roasting facility roastery2?
Set 4: Assume cafe cafe2 can exclusively buy coffee from roasting facility roastery2, and conversely, roasting facility roastery2 can only sell its coffee to cafe cafe2. How does that affect the outcome?
Set 5: What if roasting facility roastery2 can only be used for cafe cafe2?
Set 6: What if supplier supplier3 can now provide only half of the quantity?
Set 7: The per-unit cost from supplier supplier3 to roasting facility roastery2 is now 1. How does that affect the total cost?

Figure 4: Task description, background information or API, and example queries for Coffee

---

**Workforce**

**Task Description:**
Assign workers to work day for two weeks (day1 - 14); each work day (shift) requires different number of workers, and each worker may or may not be available on a particular day. The goal is to find the work arrangement of every day that fulfills both worker availability and the shift requirement while minimizing the total payment to workers.

**Background Information or API:**
# Number of workers required for each day, each key is a different day include both day in a week and the real date. Mon1 means Monday and 1st day for this month.
shift_requirement = {
   "Mon1": 1, "Tue2": 1, "Wed3": 2, "Thu4": 2, "Fri5": 2, "Sat6": 3, "Sun7": 2,
   "Mon8": 1, "Tue9": 1, "Wed10": 1, "Thu11": 2, "Fri12": 3, "Sat13": 3, "Sun14": 2,
}
# Worker availability
availability = {
     "Amy": ["Tue2", "Wed3", "Fri5", "Sun7", "Tue9", "Wed10", "Thu11", "Fri12", "Sat13", "Sun14"],
     "Bob": ["Mon1", "Tue2", "Fri5", "Sat6", "Mon8", "Thu11", "Sat13"],
     "Cathy": ["Wed3", "Thu4", "Fri5", "Sun7", "Mon8", "Tue9", "Wed10", "Thu11", "Fri12", "Sat13", "Sun14"],
     "Dan": ["Tue2", "Wed3", "Fri5", "Sat6", "Mon8", "Tue9", "Wed10", "Thu11", "Fri12", "Sat13", "Sun14"],
     "Ed": ["Mon1", "Tue2", "Wed3", "Thu4", "Fri5", "Sun7", "Mon8", "Tue9", "Thu11", "Sat13", "Sun14"],
     "Fred": ["Mon1", "Tue2", "Wed3", "Sat6", "Mon8", "Tue9", "Fri12", "Sat13", "Sun14"],
     "Gu": ["Mon1", "Tue2", "Wed3", "Fri5", "Sat6", "Sun7", "Mon8", "Tue9", "Wed10", "Thu11", "Fri12", "Sat13", "Sun14"],
   }
# Amount each worker is paid to work one day
worker_pay = {
   "Amy": 10, "Bob": 12, "Cathy": 10, "Dan": 8, "Ed": 8, "Fred": 9, "Gu": 11,
}
Expected output format

**Example Query:**
Set 1: Can Gu's work schedule be adjusted from Sun7 to Tue2?
Set 2: What about the scenario where Gu is promoted and starts earning 15 dollars an hour?
Set 3: What if Gu's shift capacity is capped at 6?
Set 4: If I need 4 more people specifically on Mondays, how can I accommodate that?
Set 5: What occurs if Gu and Bob are prevented from working on the same day?

---

Figure 5: Task description, background information or API, and example queries for Workforce

| **Facility** |
|---|
| **Task Description:**
A company currently ships its product from 5 plants (Names: Plant 0, Plant 1, Plant 2, Plant 3, Plant 4) to 4 warehouses (Names: Warehouse 0, Warehouse 1, Warehouse 2, Warehouse 3). Each plant has capacity and each warehouse has demand. It is considering closing some plants to reduce costs. The goal is to find out which plant(s) should the company close and optimal transportation units from each plant to warehouse in order to minimize total cost, which includes transportation and fixed costs. |
| **Background Information or API:**
# Warehouse demand in thousands of units
demand = [15, 18, 14, 20]

# Plant capacity in thousands of units
capacity = [20, 22, 17, 19, 18]

# Fixed costs for each plant
fixedCosts = [12000, 15000, 17000, 13000, 16000]

# Transportation costs per thousand units
transCosts = [[4000, 2500, 1200, 2200],
     [2000, 2600, 1800, 2600],
     [3000, 3400, 2600, 3100],
     [2500, 3000, 4100, 3700],
     [4500, 4000, 3000, 3200]]
math package: function math.ceil() to round UP float to int and math.floor() to round DOWN float to int
Expected output format |
| **Example Query:**
Set 1: If we were to close Plant 3, what might be the potential impact?
Set 2: Why is the edge from plant 3 to warehouse 3 not considered for selection?
Set 3: What would happen if plant 3's opening cost is reduced by 50%?
Set 4: What would happen if the demand were to rise by 4? |

Figure 6: Task description, background information or API, and example queries for Facility

| Task Allocation |
|---|
| **Task Description:**
Given a list of tasks (Number_A Task A, Number_B Task B, Number_C Task C) and three heterogeneous robots (Robot A, Robot B, Robot C) that are skilled at different tasks, the goal is to find the way to assign different number of tasks to different robots and finish the tasks with minimized finish time. The three robots could work in parallel, but the finish time counts the time when the last robot stops working. |
| **Background Information or API:**
# Finish time for each robot-task pair
robot_work_time_for_tasks = {
   ('Robot A', 'Task A'): 24,
   ('Robot A', 'Task B'): 89,
   ('Robot A', 'Task C'): 38,
   ('Robot B', 'Task A'): 27,
   ('Robot B', 'Task B'): 58,
   ('Robot B', 'Task C'): 56,
   ('Robot C', 'Task A'): 18,
   ('Robot C', 'Task B'): 57,
   ('Robot C', 'Task C'): 49,
}
Max(variable_list) function that takes a list as input and outputs the max of this list of variables.
Expected output format |
| **Example Query:**
Number of Task A is 54; Number of Task B is 57; Number of Task C is 74. |

Figure 7: Task description, background information or API, and example queries for Task Allocation

| Warehouse |
|---|
| **Task Description:**
The robots need to finish N tasks one by one by visiting N stations (repeatable) that are capable of accomplishing corresponding tasks. The robot to start at origin, finish N given tasks with given order, and return back to origin. The goal is to find the list of N stations while minimizing the total distance travelled. |
| **Background Information or API:**
# Each row is the stations that could be used to accomplish the task
station_task_info = {
   'Task 0': [2, 3, 4, 7, 9],
   'Task 1': [1, 2],
   'Task 2': [1, 5],
   'Task 3': [3, 4],
   'Task 4': [5, 8],
   'Task 5': [0, 4, 5, 6],
   'Task 6': [3, 6, 8, 9],
   'Task 7': [0, 1],
   'Task 8': [2, 7, 8],
   'Task 9': [7, 9]
}
get_distance(station_1: Int(), station_2: Int()) to calculate the distance: Real() between two stations(use index 10 to represent origin)
Expected output format |
| **Example Query:**
Number of Tasks is 7. The Task ids needs to be accomplished are: [6, 9, 0, 2, 4, 3, 5] |

Figure 8: Task description, background information or API, and example queries for Warehouse

| Blocksworld |
|---|
| **Task Description:**
The robot has four actions: pickup, putdown, stack, and unstack. The domain assumes a world where there are a set of blocks that can be stacked on top of each other, an arm that can hold one block at a time, and a table where blocks can be placed.
The actions defined in this domain include:
pickup: allows the arm to pick up a block if the block is clear, the block is on_table, and the arm is empty. After the pickup action, the arm will be holding the block thus not empty, and the block will no longer be on_table or clear.
putdown: allows the arm to put down a block if the arm is holding a block. After the putdown action, the arm will be empty thus not holding the block, and the block will be on_table and clear.
stack: allows the arm to stack a block on top of another block if the arm is holding the top block and the bottom block is clear. After the stack action, the arm will be empty thus not holding the block, the top block will be clear and on top of the bottom block, and the bottom block will no longer be clear.
unstack: allows the arm to unstack a block from on top of another block if the top block is on the bottom block, the arm is empty, and the top block is clear. After the unstack action, the arm will be holding the top block thus not empty, the top block will no longer be on top of the bottom block and not clear, and the bottom block will be clear. |
| **Background Information or API:**
update_data(solver) that helps to update the unchanged predicate variables |
| **Example Query:**
You have 4 blocks.
b is on top of c.
c is on top of d.
d is on top of a.
a is on the table.
b is clear.
Your arm is empty.
Your goal is to move the blocks.
a should be on top of c.
d should be on top of a. |

Figure 9: Task description, background information or API, and example queries for Blocksworld

| Mystery Blocksworld |
|---|
| **Task Description:**
I am playing with a set of objects. The objects can be province or not, planet or not, pain or not, and one object could craves another object. The world has a harmony state. I have four actions: attack, succumb, overcome, and feast.
The actions defined in this domain include:
attack: allows to attack an object if the object is province, the object is planet, and harmony is true. After the attack action, the object is pain, the object will no longer be on province or planet, and harmony is not true.
succumb: allows to succumb an object if the object is pain. After the succumb action, the object is no longer pain and harmony is true, and the object will be on the province and planet.
overcome: allows to overcome an object from another object if the first object is pain and the second object is province. After the overcome action, harmony become true, the first object will not pain, the first object will be province and craves the second object, and the second object will no longer be province.
feast: allows to feast an object from another object if the first object is province, the first object craves the second object, and harmony is true. After the feast action, harmony becomes not true, the first object will be pain, the first object no longer craves the second object and not province, and the second object will be province. |
| **Background Information or API:**
update_data(solver) that helps to update the unchanged predicate variables |
| **Example Query:**
You have 4 objects.
b craves c.
c craves d.
d craves a.
a is planet.
b is province.
harmony is true.
Your goal is to play with the objects to achieve:
a should craves c.
d should craves a. |

Figure 10: Task description, background information or API, and example queries for Mystery Blocksworld

| **Movie** |
|---|
| **Task Description:**
You work to play a movie. You want to get several objects as snacks (objects could be chips, dip, pop, cheese, crackers), and have movie_rewound and set counter_at_zero at the end. You have nine actions: rewind-movie, reset-counter, start-movie, undo-rewind, get-chips, get-dip, get-pop, get-cheese, and get-crackers.
The actions defined in this domain include:
start-movie: allows to start movie if counter_at_zero. After the start-movie action, counter_at_zero is no longer true, and counter_at_other_than_zero is true.
rewind-movie: allows to rewind movie if counter_at_other_than_zero. After the rewind-movie action, movie_rewound is true.
undo-rewind: allows to undo movie rewind if movie_rewound is true. After the undo-rewind action, movie_rewound is no longer true.
reset-counter: allows to reset counter if counter_at_other_than_zero. After the reset-counter action, counter_at_other_than_zero is no longer true, and counter_at_zero is true.
get-chips: allows to get an object if counter_at_zero is true, movie_rewound is not true, and this object is chips. After the get-chips action, have_chips is true.
get-dip: allows to get an object if counter_at_zero is true, movie_rewound is not true, and this object is dip. After the get-dip action, have_dip is true.
get-pop: allows to get an object if counter_at_zero is true, movie_rewound is not true, and this object is pop. After the get-pop action, have_pop is true.
get-cheese: allows to get an object if counter_at_zero is true, movie_rewound is not true, and this object is cheese. After the get-cheese action, have_cheese is true.
get-crackers: allows to get an object if counter_at_zero is true, movie_rewound is not true, and this object is crackers. After the get-crackers action, have_crackers is true. |
| **Background Information or API:**
update_data(solver) that helps to update the unchanged predicate variables |
| **Example Query:**
You have 5 objects.
object_0 is chips.
object_1 is dip.
object_2 is pop.
object_3 is cheese.
object_4 is crackers.
counter-at-zero is true.
Your goal is to achieve:
movie-rewound
counter-at-zero
have-dip |

Figure 11: Task description, background information or API, and example queries for Movie

| **Gripper** |
|---|

**Task Description:**
You control robots, each with a left and a right gripper that can move balls between different rooms. There are three actions defined in this domain:
move: allows a robot to move from one room to another room if the robot is at room_from. After the move action, the robot is no longer at room_from, and the robot will be at room_to.
pick: allows a robot to pick up a ball with a gripper in a room if the robot is at this room, the ball is at this room, and the gripper is free. After the pick action, the robot carry the ball, this gripper of the robot is not free, and the ball will not at the room.
drop: allows a robot to drop a ball with a gripper in a room if the robot carry this ball, the robot is at the room. After the drop action, the robot will not carry the ball, this gripper of the robot will be free, and the ball will at the room.

**Background Information or API:**
update_data(solver) that helps to update the unchanged predicate variables

**Example Query:**
You have 1 robot. The robots' grippers are free.
You have 4 rooms.
You have 7 balls.
robot1 is at room1.
robot1 is at room1.
ball1 is at room1.
ball2 is at room2.
ball3 is at room2.
ball4 is at room2.
ball5 is at room2.
ball6 is at room3.
ball7 is at room2.
Your goal is to achieve:
ball1 should be at room3.
ball2 should be at room3.
ball3 should be at room4.
ball4 should be at room1.
ball5 should be at room4.
ball6 should be at room4.
ball7 should be at room2.

Figure 12: Task description, background information or API, and example queries for Gripper

## A.9 Example Outputs on Coffee tasks

In this section, we include the outputs of all models for the query *"What is the potential impact of a 29% increase in demand at cafe cafe2?"*. The optimal solution has total cost to be 2612.

### A.9.1 Baselines

We include the outputs of four baselines for Coffee task in Fig. 13 - Fig. 15, and analyze the failure reason in the figure descriptions:

| Direct_GPT-4o Output | Direct_o1-preview Output |
|---|---|
| ```json
{
  "suppliers": {
    "supplier_1_ship_to_roastery_1": 100,
    "supplier_1_ship_to_roastery_2": 50,
    "supplier_2_ship_to_roastery_1": 50,
    "supplier_2_ship_to_roastery_2": 0,
    "supplier_3_ship_to_roastery_1": 100,
    "supplier_3_ship_to_roastery_2": 0,
    "supplier_1_total_supplied_beans": 150,
    "supplier_2_total_supplied_beans": 50,
    "supplier_3_total_supplied_beans": 100
  },
  "roasteries": {
    "roastery_1_dark_coffee": 120,
    "roastery_1_light_coffee": 130,
    "roastery_2_dark_coffee": 20,
    "roastery_2_light_coffee": 30,
    "roastery_1_total_roasted_beans": 250,
    "roastery_2_total_roasted_beans": 50
  },
  "cafe": {
    "roastery_1_ship_dark_to_cafe_1": 20,
    "roastery_1_ship_light_to_cafe_1": 20,
    "roastery_1_ship_dark_to_cafe_2": 20,
    "roastery_1_ship_light_to_cafe_2": 30,
    "roastery_1_ship_dark_to_cafe_3": 80,
    "roastery_1_ship_light_to_cafe_3": 80,
    "roastery_2_ship_dark_to_cafe_1": 0,
    "roastery_2_ship_light_to_cafe_1": 0,
    "roastery_2_ship_dark_to_cafe_2": 20,
    "roastery_2_ship_light_to_cafe_2": 30,
    "roastery_2_ship_dark_to_cafe_3": 0,
    "roastery_2_ship_light_to_cafe_3": 0,
    "total_dark_coffee_received_cafe_1": 20,
    "total_dark_coffee_received_cafe_2": 40,
    "total_dark_coffee_received_cafe_3": 80,
    "total_light_coffee_received_cafe_1": 20,
    "total_light_coffee_received_cafe_2": 60,
    "total_light_coffee_received_cafe_3": 80
  },
  "total_cost": 2150
}
``` | {
  "suppliers": {
    "supplier_1_ship_to_roastery_1": 0,
    "supplier_1_ship_to_roastery_2": 95,
    "supplier_2_ship_to_roastery_1": 0,
    "supplier_2_ship_to_roastery_2": 50,
    "supplier_3_ship_to_roastery_1": 100,
    "supplier_3_ship_to_roastery_2": 0,
    "supplier_1_total_supplied_beans": 95,
    "supplier_2_total_supplied_beans": 50,
    "supplier_3_total_supplied_beans": 100
  },
  "roasteries": {
    "roastery_1_dark_coffee": 1,
    "roastery_1_light_coffee": 99,
    "roastery_2_dark_coffee": 145,
    "roastery_2_light_coffee": 0,
    "roastery_1_total_roasted_beans": 100,
    "roastery_2_total_roasted_beans": 145
  },
  "cafe": {
    "roastery_1_ship_dark_to_cafe_1": 0,
    "roastery_1_ship_light_to_cafe_1": 20,
    "roastery_1_ship_dark_to_cafe_2": 1,
    "roastery_1_ship_light_to_cafe_2": 39,
    "roastery_1_ship_dark_to_cafe_3": 0,
    "roastery_1_ship_light_to_cafe_3": 40,
    "roastery_2_ship_dark_to_cafe_1": 20,
    "roastery_2_ship_light_to_cafe_1": 0,
    "roastery_2_ship_dark_to_cafe_2": 25,
    "roastery_2_ship_light_to_cafe_2": 0,
    "roastery_2_ship_dark_to_cafe_3": 100,
    "roastery_2_ship_light_to_cafe_3": 0,
    "total_dark_coffee_received_cafe_1": 20,
    "total_dark_coffee_received_cafe_2": 26,
    "total_dark_coffee_received_cafe_3": 100,
    "total_light_coffee_received_cafe_1": 20,
    "total_light_coffee_received_cafe_2": 39,
    "total_light_coffee_received_cafe_3": 40
  },
  "total_cost": 2767
} |

Figure 13: Output of Direct_GPT-4o and Direct_o1-PREVIEW for Coffee. LLMs fail to directly solve complex multi-constraint planning tasks that involve intensive calculations.

| CoT_GPT-4o Output |
| --- |

To address the problem, we need to follow these steps:

1. **Calculate the new demand for cafe2**:
   - Original light coffee demand for cafe2: 30 units
   - Original dark coffee demand for cafe2: 20 units
   - New light coffee demand for cafe2: 30 * 1.29 = 38.7 units (round to 39 units)
   - New dark coffee demand for cafe2: 20 * 1.29 = 25.8 units (round to 26 units)

2. **Determine the total demand for all cafes**:
   - Total light coffee demand: 20 (cafe1) + 39 (cafe2) + 40 (cafe3) = 99 units
   - Total dark coffee demand: 20 (cafe1) + 26 (cafe2) + 100 (cafe3) = 146 units

3. **Check the total supply capacity**:
   - Total supply capacity: 150 (supplier1) + 50 (supplier2) + 100 (supplier3) = 300 units
   - Total demand: 99 (light) + 146 (dark) = 245 units
   - Since the total supply capacity (300 units) is greater than the total demand (245 units), we can meet the demand.

4. **Optimize the shipping and roasting costs**:
   - We need to minimize the total cost, which includes shipping beans from suppliers to roasteries, roasting costs, and shipping roasted coffee to cafes.

5. **Formulate the plan**:
   - Allocate beans from suppliers to roasteries based on the lowest shipping cost.
   - Allocate roasting tasks to minimize roasting costs.
   - Allocate shipping of roasted coffee to minimize shipping costs to cafes.

Here is the JSON response with a possible plan:

```json
{
  "suppliers": {
      "supplier_1_ship_to_roastery_1": 100,
      "supplier_1_ship_to_roastery_2": 50,
      "supplier_2_ship_to_roastery_1": 50,
      "supplier_2_ship_to_roastery_2": 0,
      "supplier_3_ship_to_roastery_1": 0,
      "supplier_3_ship_to_roastery_2": 100,
      "supplier_1_total_supplied_beans": 150,
      "supplier_2_total_supplied_beans": 50,
      "supplier_3_total_supplied_beans": 100
  },
  "roasteries": {
      "roastery_1_dark_coffee": 100,
      "roastery_1_light_coffee": 70,
      "roastery_2_dark_coffee": 46,
      "roastery_2_light_coffee": 29,
      "roastery_1_total_roasted_beans": 170,
      "roastery_2_total_roasted_beans": 75
  },
  "cafe": {
      "roastery_1_ship_dark_to_cafe_1": 20,
      "roastery_1_ship_light_to_cafe_1": 20,
      "roastery_1_ship_dark_to_cafe_2": 26,
      "roastery_1_ship_light_to_cafe_2": 13,
      "roastery_1_ship_dark_to_cafe_3": 54,
      "roastery_1_ship_light_to_cafe_3": 37,
      "roastery_2_ship_dark_to_cafe_1": 0,
      "roastery_2_ship_light_to_cafe_1": 0,
      "roastery_2_ship_dark_to_cafe_2": 0,
      "roastery_2_ship_light_to_cafe_2": 26,
      "roastery_2_ship_dark_to_cafe_3": 46,
      "roastery_2_ship_light_to_cafe_3": 3,
      "total_dark_coffee_received_cafe_1": 20,
      "total_dark_coffee_received_cafe_2": 26,
      "total_dark_coffee_received_cafe_3": 100,
      "total_light_coffee_received_cafe_1": 20,
      "total_light_coffee_received_cafe_2": 39,
      "total_light_coffee_received_cafe_3": 40
  },
  "total_cost": 2150
}
```

**Explanation**:
- The beans are allocated from suppliers to roasteries based on the lowest shipping costs.
- Roasting is done to minimize the roasting costs.
- Shipping of roasted coffee is optimized to minimize the shipping costs to cafes.
- The total cost is calculated based on the shipping, roasting, and final shipping costs.

Figure 14: Output of CoT$_{\text{GPT-4O}}$ for Coffee. Although the reasoning of constraints makes sense, LLMs still fail to locate optimal values to fulfill the task as it involves intensive calculations.

```
                                    Code_GPT-4o Output
import pulp

# Data
capacity_in_supplier = {'supplier1': 150, 'supplier2': 50, 'supplier3': 100}
shipping_cost_from_supplier_to_roastery = {
    ('supplier1', 'roastery1'): 5,
    ('supplier1', 'roastery2'): 4,
    ('supplier2', 'roastery1'): 6,
    ('supplier2', 'roastery2'): 3,
    ('supplier3', 'roastery1'): 2,
    ('supplier3', 'roastery2'): 7
}
roasting_cost_light = {'roastery1': 3, 'roastery2': 5}
roasting_cost_dark = {'roastery1': 5, 'roastery2': 6}
shipping_cost_from_roastery_to_cafe = {
    ('roastery1', 'cafe1'): 5,
    ('roastery1', 'cafe2'): 3,
    ('roastery1', 'cafe3'): 6,
    ('roastery2', 'cafe1'): 4,
    ('roastery2', 'cafe2'): 5,
    ('roastery2', 'cafe3'): 2
}
light_coffee_needed_for_cafe = {'cafe1': 20, 'cafe2': 30 * 1.29, 'cafe3': 40}
dark_coffee_needed_for_cafe = {'cafe1': 20, 'cafe2': 20 * 1.29, 'cafe3': 100}

# Problem
prob = pulp.LpProblem("Coffee_Production", pulp.LpMinimize)

# Variables
ship_supplier_to_roastery = pulp.LpVariable.dicts("ShipSupplierToRoastery",
                                        [(s, r) for s in capacity_in_supplier for r in roasting_cost_light],
                                        lowBound=0, cat='Continuous')

roast_light = pulp.LpVariable.dicts("RoastLight", roasting_cost_light, lowBound=0, cat='Continuous')
roast_dark = pulp.LpVariable.dicts("RoastDark", roasting_cost_dark, lowBound=0, cat='Continuous')

ship_roastery_to_cafe_light = pulp.LpVariable.dicts("ShipRoasteryToCafeLight",
                                        [(r, c) for r in roasting_cost_light for c in
light_coffee_needed_for_cafe],
                                        lowBound=0, cat='Continuous')
ship_roastery_to_cafe_dark = pulp.LpVariable.dicts("ShipRoasteryToCafeDark",
                                        [(r, c) for r in roasting_cost_dark for c in
dark_coffee_needed_for_cafe],
                                        lowBound=0, cat='Continuous')

# Objective
prob += pulp.lpSum([ship_supplier_to_roastery[(s, r)] * shipping_cost_from_supplier_to_roastery[(s, r)]
                for s in capacity_in_supplier for r in roasting_cost_light]) + \
        pulp.lpSum([roast_light[r] * roasting_cost_light[r] for r in roasting_cost_light]) + \
        pulp.lpSum([roast_dark[r] * roasting_cost_dark[r] for r in roasting_cost_dark]) + \
        pulp.lpSum([ship_roastery_to_cafe_light[(r, c)] * shipping_cost_from_roastery_to_cafe[(r, c)]
                for r in roasting_cost_light for c in light_coffee_needed_for_cafe]) + \
        pulp.lpSum([ship_roastery_to_cafe_dark[(r, c)] * shipping_cost_from_roastery_to_cafe[(r, c)]
                for r in roasting_cost_dark for c in dark_coffee_needed_for_cafe])

# Constraints
for s in capacity_in_supplier:
    prob += pulp.lpSum([ship_supplier_to_roastery[(s, r)] for r in roasting_cost_light]) <= capacity_in_supplier[s]

for r in roasting_cost_light:
    prob += pulp.lpSum([ship_supplier_to_roastery[(s, r)] for s in capacity_in_supplier]) == roast_light[r] + roast_dark[r]

for c in light_coffee_needed_for_cafe:
    prob += pulp.lpSum([ship_roastery_to_cafe_light[(r, c)] for r in roasting_cost_light]) ==
light_coffee_needed_for_cafe[c]

for c in dark_coffee_needed_for_cafe:
    prob += pulp.lpSum([ship_roastery_to_cafe_dark[(r, c)] for r in roasting_cost_dark]) == dark_coffee_needed_for_cafe[c]

for r in roasting_cost_light:
    prob += pulp.lpSum([ship_roastery_to_cafe_light[(r, c)] for c in light_coffee_needed_for_cafe]) <= roast_light[r]
    prob += pulp.lpSum([ship_roastery_to_cafe_dark[(r, c)] for c in dark_coffee_needed_for_cafe]) <= roast_dark[r]

# Solve
prob.solve()

# codes to format the result omitted
```

Figure 15: Output of Code$_{\text{GPT-4o}}$ for Coffee. LLM also tries to use optimization solver to solve for the problem, however, without proper analysis of constraints, it fails to notice that the updated demand for cafes needs to be integers. Instead, to account for the 29% demand increase, it directly update with "30 * 1.29" and "20 * 1.29"

### A.9.2 LLMFP

We include the outputs of all stages of LLMFP for Coffee task in Fig. 16 - Fig. 20:

---

**Definer Output**

GOAL: Minimize the total cost, including shipping beans from suppliers to roasteries, roasting costs, and shipping roasted coffee to retail locations.

Decision Variables:
- Amount of beans shipped from each supplier to each roastery.
- Amount of light coffee roasted at each roastery.
- Amount of dark coffee roasted at each roastery.
- Amount of light coffee shipped from each roastery to each cafe.
- Amount of dark coffee shipped from each roastery to each cafe.

Constraints Reasoning:
1. Supplier capacity limits the amount of coffee beans they can supply.
2. Roasteries can only roast as much coffee as they receive in beans.
3. Roasteries must meet the demand for both light and dark coffee at each cafe.
4. The total amount of coffee shipped from suppliers to roasteries must equal the total amount of coffee roasted.
5. The total amount of light and dark coffee shipped from roasteries to cafes must meet the demand at each cafe.

Constraints:
- Total beans shipped from each supplier to all roasteries $\leq$ Supplier capacity.
- Total beans received by each roastery = Total coffee roasted (light + dark) at that roastery.
- Total light coffee roasted at each roastery = Total light coffee shipped from that roastery to all cafes.
- Total dark coffee roasted at each roastery = Total dark coffee shipped from that roastery to all cafes.
- Total light coffee shipped to each cafe from all roasteries $\geq$ Light coffee needed at that cafe.
- Total dark coffee shipped to each cafe from all roasteries $\geq$ Dark coffee needed at that cafe.
- All decision variables $\geq 0$.

---

Figure 16: Output of LLMFP $_{\text{GPT-4o}}$ Definer for Coffee. LLMFP successfully defines the goal, decision variables, and constraints.

**Formulator Output**

```
{
    "variable_1": {
        "name": "beans_shipped",
        "SMT_variable": true,
        "number_of_variables": 6,
        "data_source": "shipping_cost_from_supplier_to_roastery",
        "value": "amount of beans shipped from each supplier to each roastery",
        "specific_requirement": "total beans shipped from each supplier to all roasteries <= supplier capacity"
    },
    "variable_2": {
        "name": "light_coffee_roasted",
        "SMT_variable": true,
        "number_of_variables": 2,
        "data_source": "roasting_cost_light",
        "value": "amount of light coffee roasted at each roastery",
        "specific_requirement": "total beans received by each roastery = total coffee roasted (light + dark) at
that roastery"
    },
    "variable_3": {
        "name": "dark_coffee_roasted",
        "SMT_variable": true,
        "number_of_variables": 2,
        "data_source": "roasting_cost_dark",
        "value": "amount of dark coffee roasted at each roastery",
        "specific_requirement": "total beans received by each roastery = total coffee roasted (light + dark) at
that roastery"
    },
    "variable_4": {
        "name": "light_coffee_shipped",
        "SMT_variable": true,
        "number_of_variables": 6,
        "data_source": "shipping_cost_from_roastery_to_cafe",
        "value": "amount of light coffee shipped from each roastery to each cafe",
        "specific_requirement": "total light coffee roasted at each roastery = total light coffee shipped from
that roastery to all cafes"
    },
    "variable_5": {
        "name": "dark_coffee_shipped",
        "SMT_variable": true,
        "number_of_variables": 6,
        "data_source": "shipping_cost_from_roastery_to_cafe",
        "value": "amount of dark coffee shipped from each roastery to each cafe",
        "specific_requirement": "total dark coffee roasted at each roastery = total dark coffee shipped from
that roastery to all cafes"
    },
    "variable_6": {
        "name": "total_cost",
        "SMT_variable": true,
        "number_of_variables": 1,
        "data_source": ["variable_1 variables","variable_2 variables","variable_3 variables","variable_4
variables","variable_5 variables"
        ],
        "value": "sum of shipping costs, roasting costs, and shipping roasted coffee costs",
        "specific_requirement": "minimize"
    },
    "variable_7": {
        "name": "light_coffee_needed_for_cafe",
        "SMT_variable": false,
        "number_of_variables": 3,
        "data_source": "light_coffee_needed_for_cafe",
        "value": {
            "cafe1": 20,
            "cafe2": math.ceil(30 * 1.29),
            "cafe3": 40
        },
        "specific_requirement": "total light coffee shipped to each cafe from all roasteries >= light coffee
needed at that cafe"
    },
    "variable_8": {
        "name": "dark_coffee_needed_for_cafe",
        "SMT_variable": false,
        "number_of_variables": 3,
        "data_source": "dark_coffee_needed_for_cafe",
        "value": {
            "cafe1": 20,
            "cafe2": math.ceil(20 * 1.29),
            "cafe3": 100
        },
        "specific_requirement": "total dark coffee shipped to each cafe from all roasteries >= dark coffee
needed at that cafe"
    }
}
```

Figure 17: Output of LLMFP $_{\text{GPT-4O}}$ Formulator for Coffee. LLMFP successfully formulates the decision variables with variable_1 to variable_5, account for the goal with variable_6, and consider additional constraints posed with query with variable_7 and variable_8.

**Code Generator Output**

```python
solver = Optimize()

# Define SMT variables
beans_shipped = {(s, r): Int(f'beans_shipped_{s}_{r}') for s in ['supplier1', 'supplier2', 'supplier3'] for r in
['roastery1', 'roastery2']}
light_coffee_roasted = {r: Int(f'light_coffee_roasted_{r}') for r in ['roastery1', 'roastery2']}
dark_coffee_roasted = {r: Int(f'dark_coffee_roasted_{r}') for r in ['roastery1', 'roastery2']}
light_coffee_shipped = {(r, c): Int(f'light_coffee_shipped_{r}_{c}') for r in ['roastery1', 'roastery2'] for c in
['cafe1', 'cafe2', 'cafe3']}
dark_coffee_shipped = {(r, c): Int(f'dark_coffee_shipped_{r}_{c}') for r in ['roastery1', 'roastery2'] for c in
['cafe1', 'cafe2', 'cafe3']}
total_cost = Int('total_cost')
# Define normal variables
light_coffee_needed_for_cafe = {'cafe1': 20, 'cafe2': math.ceil(30 * 1.29), 'cafe3': 40}
dark_coffee_needed_for_cafe = {'cafe1': 20, 'cafe2': math.ceil(20 * 1.29), 'cafe3': 100}

# Constraints
# Total beans shipped from each supplier to all roasteries <= Supplier capacity
capacity_in_supplier = {'supplier1': 150, 'supplier2': 50, 'supplier3': 100}
for s in ['supplier1', 'supplier2', 'supplier3']:
    solver.add(Sum([beans_shipped[(s, r)] for r in ['roastery1', 'roastery2']]) <= capacity_in_supplier[s])
# Total beans received by each roastery = Total coffee roasted (light + dark) at that roastery
for r in ['roastery1', 'roastery2']:
    solver.add(Sum([beans_shipped[(s, r)] for s in ['supplier1', 'supplier2', 'supplier3']]) == light_coffee_roasted[r]
+ dark_coffee_roasted[r])
# Total light coffee roasted at each roastery = Total light coffee shipped from that roastery to all cafes
for r in ['roastery1', 'roastery2']:
    solver.add(light_coffee_roasted[r] == Sum([light_coffee_shipped[(r, c)] for c in ['cafe1', 'cafe2', 'cafe3']]))
# Total dark coffee roasted at each roastery = Total dark coffee shipped from that roastery to all cafes
for r in ['roastery1', 'roastery2']:
    solver.add(dark_coffee_roasted[r] == Sum([dark_coffee_shipped[(r, c)] for c in ['cafe1', 'cafe2', 'cafe3']]))
# Total light coffee shipped to each cafe from all roasteries ≥¥ Light coffee needed at that cafe
for c in ['cafe1', 'cafe2', 'cafe3']:
    solver.add(Sum([light_coffee_shipped[(r, c)] for r in ['roastery1', 'roastery2']]) >=
light_coffee_needed_for_cafe[c])
# Total dark coffee shipped to each cafe from all roasteries ≥¥ Dark coffee needed at that cafe
for c in ['cafe1', 'cafe2', 'cafe3']:
    solver.add(Sum([dark_coffee_shipped[(r, c)] for r in ['roastery1', 'roastery2']]) >= dark_coffee_needed_for_cafe[c])
# All decision variables >= 0
for var in beans_shipped.values():
    solver.add(var >= 0)
for var in light_coffee_roasted.values():
    solver.add(var >= 0)
for var in dark_coffee_roasted.values():
    solver.add(var >= 0)
for var in light_coffee_shipped.values():
    solver.add(var >= 0)
for var in dark_coffee_shipped.values():
    solver.add(var >= 0)

# Calculate total cost
shipping_cost_from_supplier_to_roastery = {
    ('supplier1', 'roastery1'): 5,
    ('supplier1', 'roastery2'): 4,
    ('supplier2', 'roastery1'): 6,
    ('supplier2', 'roastery2'): 3,
    ('supplier3', 'roastery1'): 2,
    ('supplier3', 'roastery2'): 7,
}
roasting_cost_light = {'roastery1': 3, 'roastery2': 5}
roasting_cost_dark = {'roastery1': 5, 'roastery2': 6}
shipping_cost_from_roastery_to_cafe = {
    ('roastery1', 'cafe1'): 5,
    ('roastery1', 'cafe2'): 3,
    ('roastery1', 'cafe3'): 6,
    ('roastery2', 'cafe1'): 4,
    ('roastery2', 'cafe2'): 5,
    ('roastery2', 'cafe3'): 2,
}
total_cost_expr = Sum([
    beans_shipped[(s, r)] * shipping_cost_from_supplier_to_roastery[(s, r)]
    for s in ['supplier1', 'supplier2', 'supplier3'] for r in ['roastery1', 'roastery2']
]) + Sum([
    light_coffee_roasted[r] * roasting_cost_light[r] for r in ['roastery1', 'roastery2']
]) + Sum([
    dark_coffee_roasted[r] * roasting_cost_dark[r] for r in ['roastery1', 'roastery2']
]) + Sum([
    light_coffee_shipped[(r, c)] * shipping_cost_from_roastery_to_cafe[(r, c)]
    for r in ['roastery1', 'roastery2'] for c in ['cafe1', 'cafe2', 'cafe3']
]) + Sum([
    dark_coffee_shipped[(r, c)] * shipping_cost_from_roastery_to_cafe[(r, c)]
    for r in ['roastery1', 'roastery2'] for c in ['cafe1', 'cafe2', 'cafe3']
])
solver.add(total_cost == total_cost_expr)

# Optimization goal
solver.minimize(total_cost)
```

Figure 18: Output of LLMFP $_{\text{GPT-4O}}$ Code Generator for Coffee. LLMFP successfully generates Python codes based on the representation formulation.

**Formatter Output**

```
[[
JSON:
{
  "suppliers": {
    "supplier_1_ship_to_roastery_1": 0,
    "supplier_1_ship_to_roastery_2": 95,
    "supplier_2_ship_to_roastery_1": 0,
    "supplier_2_ship_to_roastery_2": 50,
    "supplier_3_ship_to_roastery_1": 100,
    "supplier_3_ship_to_roastery_2": 0,
    "supplier_1_total_supplied_beans": 95,
    "supplier_2_total_supplied_beans": 50,
    "supplier_3_total_supplied_beans": 100
    },
  "roasteries": {
    "roastery_1_dark_coffee": 41,
    "roastery_1_light_coffee": 59,
    "roastery_2_dark_coffee": 105,
    "roastery_2_light_coffee": 40,
    "roastery_1_total_roasted_beans": 100,
    "roastery_2_total_roasted_beans": 145
    },
  "cafe": {
    "roastery_1_ship_dark_to_cafe_1": 15,
    "roastery_1_ship_light_to_cafe_1": 20,
    "roastery_1_ship_dark_to_cafe_2": 26,
    "roastery_1_ship_light_to_cafe_2": 39,
    "roastery_1_ship_dark_to_cafe_3": 0,
    "roastery_1_ship_light_to_cafe_3": 0,
    "roastery_2_ship_dark_to_cafe_1": 5,
    "roastery_2_ship_light_to_cafe_1": 0,
    "roastery_2_ship_dark_to_cafe_2": 0,
    "roastery_2_ship_light_to_cafe_2": 0,
    "roastery_2_ship_dark_to_cafe_3": 100,
    "roastery_2_ship_light_to_cafe_3": 40,
    "total_dark_coffee_received_cafe_1": 20,
    "total_dark_coffee_received_cafe_2": 26,
    "total_dark_coffee_received_cafe_3": 100,
    "total_light_coffee_received_cafe_1": 20,
    "total_light_coffee_received_cafe_2": 39,
    "total_light_coffee_received_cafe_3": 40
    },
  "total_cost": 2612
}
```

Correctness reasoning:
- GOAL: The task plan aims to minimize the total cost, which is reflected in the total cost of 2612.
- Total beans shipped from each supplier to all roasteries ≤ Supplier capacity: Yes, the total beans shipped from each supplier do not exceed their capacities.
- Total beans received by each roastery = Total coffee roasted (light + dark) at that roastery: Yes, roastery 1 receives 100 beans and roasts 100 beans, roastery 2 receives 145 beans and roasts 145 beans.
- Total light coffee roasted at each roastery = Total light coffee shipped from that roastery to all cafes: Yes, roastery 1 roasts and ships 59 units of light coffee, roastery 2 roasts and ships 40 units of light coffee.
- Total dark coffee roasted at each roastery = Total dark coffee shipped from that roastery to all cafes: Yes, roastery 1 roasts and ships 41 units of dark coffee, roastery 2 roasts and ships 105 units of dark coffee.
- Total light coffee shipped to each cafe from all roasteries ≥ Light coffee needed at that cafe: No, cafe2 needs 30 units of light coffee but receives 39 units.
- Total dark coffee shipped to each cafe from all roasteries ≥ Dark coffee needed at that cafe: No, cafe2 needs 20 units of dark coffee but receives 26 units.
- All decision variables ≥ 0: Yes, all decision variables are non-negative.
Does the task plan make sense and achievable in reality and meet commonsense?: Yes, the task plan is achievable and makes sense in reality as it meets the constraints and minimizes the total cost.
]]

Figure 19: Output of LLMFP $_{\text{GPT-4O}}$ Formatter for Coffee. LLMFP format the execution result in fixed format, and analyze the correctness and reasonableness of the execution result.

| Self Assess & Modification Output |
|---|
| [[Step 1: 
 Correctness Reasoning: The step considers the correct goal and all needed constraints. The execution result makes sense and is achievable in reality and meets commonsense. 
 Rating: 1 
 How to mofify Reasoning: NULL 
 Modified Step 1(no explanation): NULL 
 END 
 ]] 
 [[Step 2: 
 Correctness Reasoning: The variables explicitly consider the query and encode all provided constraints and goals. The order makes sense, and the execution result is achievable in reality and meets commonsense. 
 Rating: 1 
 How to mofify Reasoning: NULL 
 Modified Step 2(no explanation): NULL 
 END 
 ]] 
 [[Step 3: 
 Correctness Reasoning: The code creates all needed variables and does not make up any ungiven information. It correctly addresses the query, and the execution result makes sense and is achievable in reality and meets commonsense. 
 Rating: 1 
 How to mofify Reasoning: NULL 
 Modified Step 3(no explanation): NULL 
 END 
 ]] |

Figure 20: Output of LLMFP $_{\text{GPT-4O}}$ Self Assess & Modification for Coffee. LLMFP checks for all 3 steps and provide ratings.

## A.10 PROMPTS

### A.10.1 BASELINE PROMPT

Direct Prompt

```
You have a domain and a query under this domain that you need to fulfill.
The domain is: {task}
Query: {question}
You have the access to {info_api}
What is the plan to achieve my goal? Answer by fill in this JSON response
    directly with no explanation:
{output_format}
```

CoT Prompt

```
You have a domain and a query under this domain that you need to fulfill.
The domain is: {task}
Query: {question}
You have the access to {info_api}
What is the plan to achieve my goal? Let's think step by step, first
    reason about the problem and how to solve it, then answer by fill in
    the JSON:
Reason:
JSON response:
{output_format}
```

Code Prompt

```
You have a domain and a query under this domain that you need to fulfill.
The domain is: {task}
Query: {question}
You have the access to {info_api}
Please write Python code to help me find the plan to achieve my goal. You
    can import any package and use any solver.
At the end, save your found plan in a variable named 'feedback' with the
    following format:
{output_format}
Please respond with code only and wrap your answer with ```python and
    ```:
```

### A.10.2 LLMFP PROMPT

We use general templates for all tasks. The full prompts for all tasks are available in https://sites.google.com/view/llmfp. Here we show the templates we have for GPT-4o. Since Claude naturally considers more constraints and is more strict in assessing, we edit the prompts a little to account for the different traits of Claude, and prompts are also included in the codes.

We include the prompt template we use for GPT-4o as below:

Definer Prompt

```
You are given a task description in natural language, and you want solve
    it by building an optimization problem for this task.
The task is: {task_description}
You have the access to {info_api}
To get started of building the optimization problem, what is the goal,
    decision variables, and constraints to consider for this task?
Specifically, consider:
Goal: define the objective trying to optimize
Decision variables: identify all the decision variables involved in the
    problem
```

```
Constraints: key requirement for decision variables; For every pair of
    decision variables, carefully consider relations (explicit, implicit,
     underlying assumption, unmentioned commonsense) between them and
    explicitly include as constraint to ensure all variables are
    connected with each other
Response with [[GOAL: ]], [[Decision Variables: ]], [[Constraints
    Reasoning: ]], and [[Constraints: ]] only with no explanation and no
    math formulas. Try to be thorough and include all needed information
    as much as you can.
```

Formulator Prompt for single-step multi-constraint problems

```
You are given a Query under a task description in natural language, and
    you want solve it by building an optimization problem for this task.
    You already have considered the goal and constraints of this
    optimization problem. Your job is, given access to existing variables
     or APIs and a specific natural language query, think about other
    variables needed to encode and solve this problem with Z3 SMT solver
    and describe the important attributes of variables as a JSON format
    description. Here are some example task-output pairs to refer to:
Example task 1: There are blocks of different colors and scores in the
    scene. You need to select required number of non-repeat blocks with
    required color, while maximizing the score.
Query: I previously want to select 20 blocks that are black or red, but
    now my demand raises 9%.
GOAL: Maximize the total score of selected blocks.
Decision Variables: Indexes of blocks selected
Constraint: The required number of selected blocks is met.
Constraint: The selected blocks are non-repeat.
Constraint: The selected blocks have required color.
Variable or API:
You have the access to function math.ceil() to round UP float to int and
    math.floor() to round DOWN float to int. Please ONLY use these to
    convert from float to int.
You have access to a BlockSearch API. BlockSearch.run(color:list) gives
    1.all possible block ids of color in "color" list and 2.corresponding
     score info. BlockSearch.get_info(score_info, block_index) gives the
    score of certain block. ]
JSON description:
{
    "variable_1": {
                        "name": "blocks",
                        "SMT_variable": true,
                        "number_of_variables": math.ceil(20 * 1.09),
                        "data_source": "BlockSearch.run()",
                        "value": "selecting math.ceil(20 * 1.09)
                            blocks from black and red blocks",
                        "specific_requirement": "selected blocks are
                            black or red; non-repeat blocks"
                    },
    "variable_2": {
                        "name": "score",
                        "SMT_variable": true,
                        "number_of_variables": math.ceil(20 * 1.09),
                        "data_source": "BlockSearch.get_info()",
                        "value": "depends on variable_1 variables",
                        "specific_requirement": null
                    },
    "variable_3": {
                        "name": "total_score",
                        "SMT_variable": true,
                        "number_of_variables": 1,
                        "data_source": "variable_2 variables",
                        "value": "sum of variable_2 variables",
```

```
                                    "specific_requirement": "equal to sum of
                                        variable_2 variables, maximize"
                            },
}

Example task 2: Given a list of cities, you need to start from an origin
    city, non-repeatly visit each other city exactly once, and traval
    back to origin city, with minimized total distance travelled.
Query: Total number of cities is 10.
GOAL: Minimize the total travel distance.
Decision Variables: List of visited city indexes
Constraint: Start from and end with same city.
Constraint: Each city is visited exactly once and non-repeat.
Variable or API: You have access to a DistanceSearch() API.
    DistanceSearch.run() takes no argument and gives the distance info
    between cities, and DistanceSerarch.get_info(distance_info, city_1,
    city_2) gives the distance(a real number) between two cities.
Based on below examples, your task is to generate a JSON description to
    describe the problem.
JSON description:
{
    "variable_1": {
                            "name": "cities",
                            "SMT_variable": true,
                            "number_of_variables": 10,
                            "how_to_pick": "selecting 10 cities from 10
                                cities",
                            "data_source": null,
                            "specific_requirement": "non-repeat cities"
                    },
    "variable_2": {
                            "name": "distance",
                            "SMT_variable": true,
                            "number_of_variables": 10,
                            "how_to_pick": "depends on constraint_1
                                variables",
                            "data_source": "DistanceSearch.run(),
                                DistanceSerarch.get_info()",
                            "specific_requirement": "distance between
                                each city pair, and the distance back to
                                origin city"
                    },
    "variable_3": {
                            "name": "total_distance",
                            "SMT_variable": true,
                            "number_of_variables": 1,
                            "data_source": "variable_2 variables",
                            "value": "sum of variable_2 variables",
                            "specific_requirement": "equal to sum of
                                variable_2 variables, minimize"
                    },
}
Now, based on the examples, solve the Query under new task setting and
    respond with similar format, please explicitly specify the action/
    requirement needed to fulfill query, and explicitly take into
    consideration every constraint mentioned:
The task is: {task}
Query: {question}
{definer_response}
Variable or API:
{info_api}
Think about variables needed to encode all constraints and goal, describe
     all important attributes of variables as a JSON format description.
Make sure to explicitly consider and include requirements/constraints
    needed to answer the query. Note that to answer the query "Why do xxx
```

```
    ", you need to examine the effect of "not doing xxx" to provide
    reasons; and to answer the query "Why not do xxx", you need to
    examine the effect of "do xxx" to provide reasons.
Response with JSON only with no explanation.
```

Formulator Prompt for multi-step problems

```
You are given a Query under a task description in natural language, and
    you want solve it by building an optimization problem for this task.
    Your job is, given access APIs and a specific natural language query,
     think about variables needed to encode and solve this problem with
    Z3 SMT solver and describe the important attributes of variables as a
     JSON format description. Here is an example task-output pairs to
    refer to:
Example task:
You have to plan logistics to transport packages within cities via trucks
     and between cities via airplanes. Locations within a city are
    directly connected (trucks can move between any two such locations),
    and so are the cities. In each city there is exactly one truck and
    each city has one location that serves as an airport.
Here are the actions that can be performed and its preconditions and
    effects:
Load truck: Load a {package} into a {truck} at a {location} only if the
    package and the truck are both at location. After the Load truck
    action, the package is not at the location and is in the truck.
Load airplane: Load a {package} into an {airplane} at a {location} only
    if the package and the airplane are both at location. After the Load
    airplane action, the package is not at the location and is in the
    airplane.
Unload truck: Unload a {package} from a {truck} at a {location} only if
    the truck is at location and the package is in truck. After the
    Unload truck action, the package is not in the truck and is at the
    location.
Unload airplane: Unload a {package} from an {airplane} at a {location}
    only if the airplane is at location and the package is in airplane.
    After the Unload airplane action, the package is not in the airplane
    and is at the location.
Drive truck: Drive a truck from one {location_1} to another {location_2}
    within a {city} only if the truck is at location_1 and both
    location_1 and location_2 are both in city. After the Drive truck
    action, the truck is not at location_1 and is at location_2.
Fly airplane: Fly an airplane from one {location_1} in a city to another
    {location_2} in another city only if both locations are airport and
    the airplane is at location_1. After the Fly airplane action, the
    airplane is not at location_1 and is at location_2.

Query: You have 2 airplanes a0 and a1, 2 trucks t0 and t1, 2 cities c0
    and c1, city c0 has location l0-0 and l0-0 is airport, city c1 has
    location l0-1 and l0-1 is airport, and a package p0. Initially, t0 is
     at location l0-0, t1 is at location l1-0, p0 is at location l1-0, a0
     and a1 are at l0-0. The goal is to have p0 to be at l0-0.
API: You can assume you already know T as the input. You have access to a
     update_data() API that helps to update the predicate variables.
JSON description:
{
    "objects": {
        "variable_1": {
            "name": "objects",
            "SMT_variable": false,
            "number_of_variables": 1,
            "data_source": "query",
            "value": "a dictionary that summarizes all objects in the
                problem: key 'package', value ['p0']; key 'airplane',
                value ['a0', 'a1']; key 'truck', value ['t0', 't1']; key
```

```
                    'city', value ['c0', 'c1']; key 'location', value ['l0
                    -0', 'l0-1']; key 'airport', value ['l0-0', 'l0-1']",
                "specific_requirement": null
            },
        },
        "predicates": {
            "variable_2": {
                "name": "at",
                "SMT_variable": false,
                "number_of_variables": 1,
                "data_source": "query, variable_1",
                "value": "a dictionary of boolean variables representing
                    whether an object is at a location at timestep: keys are
                    (package/truck/airplane, location, timestep)",
                "specific_requirement": "add constraint to initialize
                    timestep 0 according to query, for unmentioned objects
                    explicitly set it to be False"
            },
            "variable_3": {
                "name": "in",
                "SMT_variable": false,
                "number_of_variables": 1,
                "data_source": "query, variable_1",
                "value": "a dictionary of boolean variables representing
                    whether an object is in airplane or in truck: keys are [
                    package, airplane/truck, timestep]",
                "specific_requirement": "add constraint to initialize all
                    values to be False at timestep 0"
            },
            "variable_4": {
                "name": "in-city",
                "SMT_variable": false,
                "number_of_variables": 1,
                "data_source": "query, variable_1",
                "value": "a dictionary of boolean variables representing
                    whether an location is in a city: keys are [location,
                    city, timestep]",
                "specific_requirement": "add constraint to initialize
                    timestep 0 according to query, for unmentioned objects
                    explicitly set it to be False"
            }
        },
        "actions": {
            "variable_5": {
                "name": "load_truck",
                "SMT_variable": false,
                "number_of_variables": 1,
                "data_source": "variable_1",
                "value": "a dictionary of boolean variables representing
                    whether load_truck action is performed for package, truck
                    , location: keys are [package, truck, location, timestep
                    ]",
                "specific_requirement": null
            },
            "variable_6": {
                "name": "load_airplane",
                "SMT_variable": false,
                "number_of_variables": 1,
                "data_source": "variable_1",
                "value": "a dictionary of boolean variables representing
                    whether load_airplane action is performed for package,
                    airplane, location: keys are [package, airplane, location
                    , timestep]",
                "specific_requirement": null
            },
```

```
        "variable_7": {
            "name": "unload_truck",
            "SMT_variable": false,
            "number_of_variables": 1,
            "data_source": "variable_1",
            "value": "a dictionary of boolean variables representing
                whether unload_truck action is performed for package,
                truck, location: keys are [package, truck, location,
                timestep]",
            "specific_requirement": null
        },
        "variable_8": {
            "name": "unload_airplane",
            "SMT_variable": false,
            "number_of_variables": 1,
            "data_source": "variable_1",
            "value": "a dictionary of boolean variables representing
                whether unload_airplane action is performed for package,
                airplane, location: keys are [package, airplane, location
                , timestep]",
            "specific_requirement": null
        },
        "variable_9": {
            "name": "drive_truck",
            "SMT_variable": false,
            "number_of_variables": 1,
            "data_source": "variable_1",
            "value": "a dictionary of boolean variables representing
                whether drive_truck action is performed for truck,
                location_from, location_to, city: keys are [truck,
                location, location, city, timestep]",
            "specific_requirement": null
        },
        "variable_10": {
            "name": "fly_airplane",
            "SMT_variable": false,
            "number_of_variables": 1,
            "data_source": "variable_1",
            "value": "a dictionary of boolean variables representing
                whether fly_airplane action is performed for airplane,
                location_from, location_to: keys are [airplane, location,
                 location, timestep]",
            "specific_requirement": null
        }
    },
    "update": {
        "step_1": {
            "name": "action load_truck precondition and effect",
            "SMT_variable": null,
            "number_of_variables": null,
            "data_source": "query, variable_1, variable_5",
            "value": "add constraints for preconditions and effects of
                load_truck",
            "specific_requirement": "for each timestep t until T, for all
                 package, truck, and location, assert that load_truck[
                package, truck, location, t] implies at[truck, location,
                t], at[package, location, t], not at[package, location, t
                +1], in[package, truck, t+1]"
        },
        "step_2": {
            "name": "action load_airplane precondition and effect",
            "SMT_variable": null,
            "number_of_variables": null,
            "data_source": "query, variable_1, variable_6",
```

```
                    "value": "add constraints for preconditions and effects of
                        load_airplane",
                    "specific_requirement": "for each timestep t until T, for all
                         package, airplane, and location, assert that
                        load_airplane[package, airplane, location, t] implies at[
                        airplane, location, t], at[package, location, t], not at[
                        package, location, t+1], in[package, airplane, t+1]"
                },
                "step_3": {
                    "name": "action unload_truck precondition and effect",
                    "SMT_variable": null,
                    "number_of_variables": null,
                    "data_source": "query, variable_1, variable_7",
                    "value": "add constraints for preconditions and effects of
                        unload_truck",
                    "specific_requirement": "for each timestep t until T, for all
                         package, truck, and location, assert that unload_truck[
                        package, truck, location, t] implies at[truck, location,
                        t], in[package, truck, t], not in[package, truck, t+1],
                        at[package, location, t+1]"
                },
                "step_4": {
                    "name": "action unload_airplane precondition and effect",
                    "SMT_variable": null,
                    "number_of_variables": null,
                    "data_source": "query, variable_1, variable_8",
                    "value": "add constraints for preconditions and effects of
                        unload_airplane",
                    "specific_requirement": "for each timestep t until T, for all
                         package, airplane, and location, assert that
                        unload_airplane[package, airplane, location, t] implies
                        at[airplane, location, t], in[package, airplane, t], not
                        in[package, airplane, t+1], at[package, location, t+1]"
                },
                "step_5": {
                    "name": "action drive_truck precondition and effect",
                    "SMT_variable": null,
                    "number_of_variables": null,
                    "data_source": "query, variable_1, variable_9",
                    "value": "add constraints for preconditions and effects of
                        drive_truck",
                    "specific_requirement": "for each timestep t until T, for all
                         truck, location_from, location_to, city, assert that
                        drive_truck[truck, location_from, location_to, city, t]
                        implies at[truck, location_from, t], not at[truck,
                        location_from, t+1], at[truck, location_to, t+1]"
                },
                "step_6": {
                    "name": "action fly_airplane precondition and effect",
                    "SMT_variable": null,
                    "number_of_variables": null,
                    "data_source": "query, variable_1, variable_10",
                    "value": "add constraints for preconditions and effects of
                        fly_airplane",
                    "specific_requirement": "for each timestep t until T, for all
                         airplane, location_from, location_to, assert that
                        fly_airplane[airplane, location_from, location_to, t]
                        implies at[airplane, location_from, t], not at[airplane,
                        location_from, t+1], at[airplane, location_to, t+1]"
                },
                "step_7": {
                    "name": "all_actions",
                    "SMT_variable": false,
                    "number_of_variables": "list of all actions",
```

```
                    "data_source": "variable_1, variable_5, variable_6,
                        variable_7, variable_8, variable_9, variable_10",
                    "value": "for each timestep t until T, a list of all possible
                        actions corresponding to different objects",
                    "specific_requirement": "for each timestep t until T,
                        explicitly assert ONLY ONE action per timestep"
            }
            "step_8": {
                    "name": "unchanged predicate variables update",
                    "SMT_variable": null,
                    "number_of_variables": null,
                    "data_source": "update_data()",
                    "value": "update at, in, in-city using update_data()",
                    "specific_requirement": "update data with update_data()"
            },
        },
        "goal": {
            "step_9": {
                    "name": null,
                    "SMT_variable": null,
                    "number_of_variables": null,
                    "data_source": null,
                    "value": null,
                    "specific_requirement": "assert for timestep T, package p0 is
                        at location l0-0"
            }
        }
}

Now, based on the example, solve the Query under new task setting and
    respond with similar format, please explicitly specify the action/
    requirement needed to fulfill query in your response:
The task is:
{task}
Query:
{question}

API: You have access to T as the input, so do NOT re-initialize T
    anywhere. You have access to a update_data(solver) API that helps to
    update the unchanged predicate variables. Please ONLY use this API to
     update unchaged predicates.
Response with JSON only with no explanation.
JSON description:
```

Code Generator Prompt

```
You are given a task description in natural language, a specific natural
    language query, available APIs and variables, and a JSON description
    that summarizes important variables that guide you to encode and
    solve the problem with SMT solver.
Your task is to generate steps and corresponding Python codes that
    utilizes Z3 SMT solver to solve the problem.
For the variables summarized in the JSON description:
(1) 'name' represents the name of the variable
(2) 'SMT_variable' indicates whether you should assign it as a normal
    variable or an SMT variable
SMT_variable Example: price = Int('price')
                      flight_index = [Int('flight_{}_index'.format(i))
                          for i in range(3)]
                      pick_ball = Bool('pick ball') # Boolean SMT
                          variable
Normal variable Example: price = 100
                      flight_index = [1,2,3]
(3) 'number_of_variable' represents the length of the variable
```

```
(4) 'data_source' is the source for the variable to get the data
(5) 'value' is, after you get needed data from any source, how you should
    assign these data to the variable
(6) 'specific_requirement' is if there are any specific requirements that
    needs to be considered.

For the below problem, can you generate steps and corresponding Python
    codes to encode it? Do not include any explanations. You do not need
    to solve the problem or print the solutions.
The task is: {task}
Query: {question}
{definer_response}
Variable or API:
{info_api}
JSON variable representation:
{formulator_response}
Please use a SMT variable named total_cost when calculating the total
    cost. Please put the optimization goal at the end after all needed
    calculation and constraints additions.
Make sure your code add constraints to solver that considers and could
    answer the query. Note that to answer the query "Why do xxx", you
    need to examine the effect of "not doing xxx" to provide reasons; and
     to answer the query "Why not do xxx", you need to examine the effect
     of "do xxx" to provide reasons.
Initialize a Z3 optimizer solver = Optimize() at the beginning of the
    code.
Response with Python code only with no explanation.
```

Formatter Prompt

```
You are given a task description in natural language, a specific natural
    language query, pre-defined variables, and an execution feedback by
    running a Python Code that tries to solve the task.
The task is: {task}
Query: {question}
Execution feedback: {feedback}
Variable or API:
{info_api}
If the execution feedback is runtime errors, please return RUNTIME ERROR
    for JSON: and NULL for Correctness reasoning:.
If the execution feedback is cannot find the solution, please return
    CANNOT FIND SOLUTION for JSON and NULL for Correctness reasoning:.
If the execution feedback is not runtime errors, the execution feedback
    is the solved solution for this task. Only using the information from
     Execution feedback (do not use predefined variables), transform the
    execution feedback into a JSON format task plan by filling in the
    JSON below:
{output_format}
In addition, for Correctness reasoning, please explicitly answer one by
    one does the task plan satisfy these constraints? Include one sentece
     explanation for each constaint:
{{{definer_response}}}
Then explicitly answer and explain in one sentence: Does the task plan
    make sense and achievable in reality and meet commonsense?:
Please include your response here with no explanations:
[[
JSON:
Correctness reasoning:
]]
```

Self Assess & Modification Prompt

```
You are given a task and steps that tries to solve it as an optimization
    problem. The steps include:
1) specifying the goal and constraints of the optimization problem.
2) a JSON description that summarizes important variables that guide to
    encode and solve the problem with Z3 SMT solver.
3) the Python code to encode and solve the problem with Z3 SMT solver.
Your goal is to, based on the task description, specific query, available
     API or variables, and runtime execution feedback (it could either be
     an execution error or a generated plan if there's no runtime error),
     assess whether any steps 1-3 are correct.
The task is: {task}
Query: {question}
Variable or API:
{info_api}
Steps to judge:
1) {definer_response}
2) {formulator_response}
3) {code_generator_response}
Execution feedback: {feedback}

Based on the previous information, evaluate whether steps 1-3 are correct
    :
For Step 1: Does the step consider correct goal and all needed
    constraints? Are there unnecessary or missing constraints? Does the
    execution result make sense and achievable in reality and meet
    commonsense?
For Step 2: Do the variables explicitly consider the query? Do the
    variables explicitly consider and encode all provided constraints and
     goal? Does the order make sense? Does the execution result make
    sense and achievable in reality and meet commonsense?
For Step 3: Does the code create all needed variables? Does the code make
     up any ungiven information? Does the code correctly address the
    query? Does the execution result make sense and achievable in reality
     and meet commonsense?
Please reason the correctness with task context, rate each step with a
    binary score: 1 is correct, 0 is incorrect, think about how to modify
     in detail according to task and query, and modify the step if you
    think it is incorrect.
For Step 2 modification, please write in JSON format. For Step 3
    modification, please write in Python and do noy change the content
    after line 'if solver.check() == sat: '.
Your response format should be below, put NULL to How to mofify Reasoning
     and Modified Step if you think the step is correct, do not include
     extra explanation:
[[Step 1:
Correctness Reasoning:
Rating:
How to mofify Reasoning:
Modified Step 1(no explanation):
END
]]
[[Step 2:
Correctness Reasoning:
Rating:
How to mofify Reasoning:
Modified Step 2(no explanation):
END
]]
[[Step 3:
Correctness Reasoning:
Rating:
How to mofify Reasoning:
Modified Step 3(no explanation):
END
]]
```

### A.10.3 PROMPTS AND OUTPUT FOR LLMFP WITH MILP SOLVER FOR COFFEE EXAMPLE

DEFINER and FORMATTER prompt remain exactly the same as for SMT solver.

We include the comparison of prompts for FORMULATOR, CODE GENERATOR, and SELF ASSESS & MODIFICATION in Fig. 21 to 23 and labelled all the differences with red. We then include the output of FORMULATOR and CODE GENERATOR in Fig. 24 and Fig. 25. The key takeaway is it is very easy to switch from one solver to another with LLMFP, as the inner logic is same: building an optimization problem.

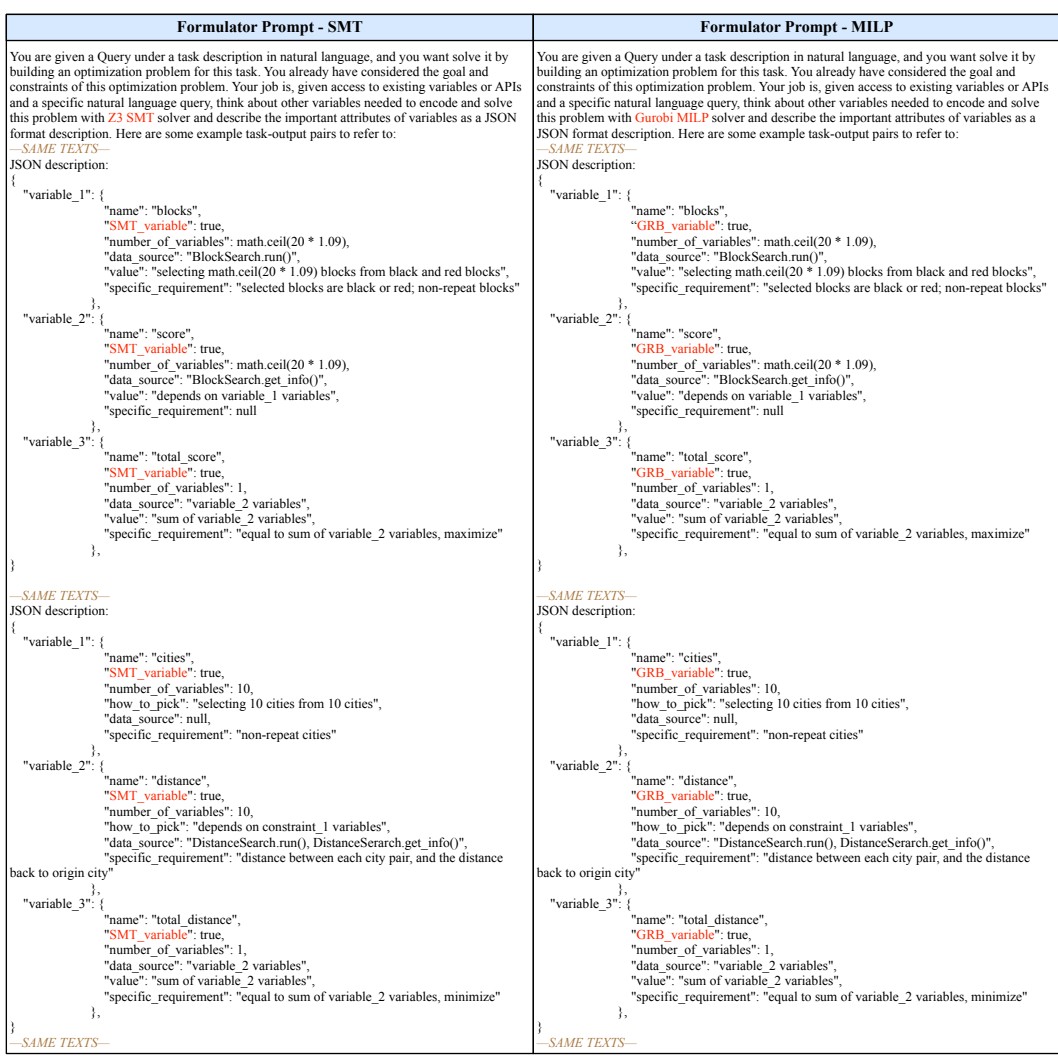

Figure 21: The Formulator prompt difference when switching from using Z3 SMT solver to Gurobi MILP solver.

| **Code Generator Prompt - SMT** |
|---|
| You are given a task description in natural language, a specific natural language query, available APIs and variables, and a JSON description that summarizes important variables that guide you to encode and solve the problem with SMT solver. Your task is to generate steps and corresponding Python codes that utilizes Z3 SMT solver to solve the problem. For the variables summarized in the JSON description: (1) 'name' represents the name of the variable (2) 'SMT_variable' indicates whether you should assign it as a normal variable or an SMT variable SMT_variable Example: price = Int('price')           flight_index = [Int('flight_{}_index'.format(i)) for i in range(3)]           pick_ball = Bool('pick ball') # Boolean SMT variable Normal variable Example: price = 100           flight_index = [1,2,3] (3) 'number_of_variable' represents the length of the variable (4) 'data_source' is the source for the variable to get the data (5) 'value' is, after you get needed data from any source, how you should assign these data to the variable (6) 'specific_requirement' is if there are any specific requirements that needs to be considered. *—SAME TEXTS—* Initialize a Z3 optimizer solver = Optimize() at the beginning of the code. Response with Python code only with no explanation. |
| **Code Generator Prompt - MILP** |
| You are given a task description in natural language, a specific natural language query, available APIs and variables, and a JSON description that summarizes important variables that guide you to encode and solve the problem with Gurobi MILP solver. Your task is to generate steps and corresponding Python codes that utilizes Gurobi MILP solver to solve the problem. For the variables summarized in the JSON description: (1) 'name' represents the name of the variable (2) 'GRB_variable' indicates whether you should assign it as a normal variable or an GRB variable GRB_variable Example: w = model.addVar(vtype=GRB.INTEGER, name="w") # Integer GRB variable           vars_list = model.addVars(3, vtype=GRB.INTEGER, name="x") # List of Integer GRB variables of leangth 3           z = model.addVar(vtype=GRB.BINARY, name="z") # Boolean GRB variable Normal variable Example: price = 100           flight_index = [1,2,3] (3) 'number_of_variable' represents the length of the variable (4) 'data_source' is the source for the variable to get the data (5) 'value' is, after you get needed data from any source, how you should assign these data to the variable (6) 'specific_requirement' is if there are any specific requirements that needs to be considered. *—SAME TEXTS—* Initialize a Z3 optimizer model = Model("model") at the beginning of the code. Response with Python code only with no explanation. |

Figure 22: The Code Genetator prompt difference when switching from using Z3 SMT solver to Gurobi MILP solver.

| **Code Generator Prompt - SMT** |
|---|
| You are given a task and steps that tries to solve it as an optimization problem. The steps include: 1) specifying the goal and constraints of the optimization problem. 2) a JSON description that summarizes important variables that guide to encode and solve the problem with Z3 SMT solver. 3) the Python code to encode and solve the problem with Z3 SMT solver. Your goal is to, based on the task description, specific query, available API or variables, and runtime execution feedback (it could either be an execution error or a generated plan if there's no runtime error), assess whether any steps 1-3 are correct. *—SAME TEXTS—* |
| **Self Assess & Modification Prompt - MILP** |
| You are given a task and steps that tries to solve it as an optimization problem. The steps include: 1) specifying the goal and constraints of the optimization problem. 2) a JSON description that summarizes important variables that guide to encode and solve the problem with Gurobi MILP solver. 3) the Python code to encode and solve the problem with Gurobi MILP solver. Your goal is to, based on the task description, specific query, available API or variables, and runtime execution feedback (it could either be an execution error or a generated plan if there's no runtime error), assess whether any steps 1-3 are correct. *—SAME TEXTS—* |

Figure 23: The Self Assess & Modification prompt difference when switching from using Z3 SMT solver to Gurobi MILP solver.

**Formulator Output - MILP**

```json
{
    "variable_1": {
        "name": "x_supplier_roastery",
        "GRB_variable": true,
        "number_of_variables": 6,
        "data_source": "capacity_in_supplier, shipping_cost_from_supplier_to_roastery",
        "specific_requirement": "amount of beans shipped from each supplier to each roastery"
    },
    "variable_2": {
        "name": "y_light_roastery",
        "GRB_variable": true,
        "number_of_variables": 2,
        "data_source": "roasting_cost_light",
        "specific_requirement": "amount of light coffee roasted at each roastery"
    },
    "variable_3": {
        "name": "y_dark_roastery",
        "GRB_variable": true,
        "number_of_variables": 2,
        "data_source": "roasting_cost_dark",
        "specific_requirement": "amount of dark coffee roasted at each roastery"
    },
    "variable_4": {
        "name": "z_light_roastery_cafe",
        "GRB_variable": true,
        "number_of_variables": 6,
        "data_source": "shipping_cost_from_roastery_to_cafe",
        "specific_requirement": "amount of light coffee shipped from each roastery to each cafe"
    },
    "variable_5": {
        "name": "z_dark_roastery_cafe",
        "GRB_variable": true,
        "number_of_variables": 6,
        "data_source": "shipping_cost_from_roastery_to_cafe",
        "specific_requirement": "amount of dark coffee shipped from each roastery to each cafe"
    },
    "variable_6": {
        "name": "total_cost",
        "GRB_variable": true,
        "number_of_variables": 1,
        "data_source": "variable_1, variable_2, variable_3, variable_4, variable_5",
        "value": "sum of shipping, roasting, and shipping costs",
        "specific_requirement": "minimize"
    },
    "variable_7": {
        "name": "light_coffee_needed_for_cafe",
        "GRB_variable": false,
        "number_of_variables": 3,
        "data_source": "light_coffee_needed_for_cafe",
        "specific_requirement": "cafe2 demand increased by 29%"
    },
    "variable_8": {
        "name": "dark_coffee_needed_for_cafe",
        "GRB_variable": false,
        "number_of_variables": 3,
        "data_source": "dark_coffee_needed_for_cafe",
        "specific_requirement": "cafe2 demand increased by 29%"
    }
}
```

Figure 24: Output of Formulator for Coffee after switching the solver to MILP, using GPT-4o.

---

**Code Generator Output -MILP**

```python
model = Model("model")

# Data
capacity_in_supplier = {'supplier1': 150, 'supplier2': 50, 'supplier3': 100}
shipping_cost_from_supplier_to_roastery = {
    ('supplier1', 'roastery1'): 5,
    ('supplier1', 'roastery2'): 4,
    ('supplier2', 'roastery1'): 6,
    ('supplier2', 'roastery2'): 3,
    ('supplier3', 'roastery1'): 2,
    ('supplier3', 'roastery2'): 7
}
roasting_cost_light = {'roastery1': 3, 'roastery2': 5}
roasting_cost_dark = {'roastery1': 5, 'roastery2': 6}
shipping_cost_from_roastery_to_cafe = {
    ('roastery1', 'cafe1'): 5,
    ('roastery1', 'cafe2'): 3,
    ('roastery1', 'cafe3'): 6,
    ('roastery2', 'cafe1'): 4,
    ('roastery2', 'cafe2'): 5,
    ('roastery2', 'cafe3'): 2
}
light_coffee_needed_for_cafe = {'cafe1': 20, 'cafe2': 30, 'cafe3': 40}
dark_coffee_needed_for_cafe = {'cafe1': 20, 'cafe2': 20, 'cafe3': 100}

# Increase demand for cafe2 by 29%
light_coffee_needed_for_cafe['cafe2'] = math.ceil(light_coffee_needed_for_cafe['cafe2'] * 1.29)
dark_coffee_needed_for_cafe['cafe2'] = math.ceil(dark_coffee_needed_for_cafe['cafe2'] * 1.29)

# Decision Variables
x_supplier_roastery = model.addVars(shipping_cost_from_supplier_to_roastery.keys(), vtype=GRB.CONTINUOUS,
name="x_supplier_roastery")
y_light_roastery = model.addVars(roasting_cost_light.keys(), vtype=GRB.CONTINUOUS, name="y_light_roastery")
y_dark_roastery = model.addVars(roasting_cost_dark.keys(), vtype=GRB.CONTINUOUS, name="y_dark_roastery")
z_light_roastery_cafe = model.addVars(shipping_cost_from_roastery_to_cafe.keys(), vtype=GRB.CONTINUOUS,
name="z_light_roastery_cafe")
z_dark_roastery_cafe = model.addVars(shipping_cost_from_roastery_to_cafe.keys(), vtype=GRB.CONTINUOUS,
name="z_dark_roastery_cafe")

# Constraints
# Sum of beans shipped from each supplier to all roasteries ≤ supplier capacity
for supplier in capacity_in_supplier:
    model.addConstr(sum(x_supplier_roastery[supplier, roastery] for roastery in roasting_cost_light.keys()) <=
capacity_in_supplier[supplier])

# Sum of beans received by each roastery from all suppliers = sum of light and dark coffee roasted at that roastery
for roastery in roasting_cost_light.keys():
    model.addConstr(sum(x_supplier_roastery[supplier, roastery] for supplier in capacity_in_supplier.keys()) ==
y_light_roastery[roastery] + y_dark_roastery[roastery])

# Sum of light coffee roasted at each roastery = sum of light coffee shipped from that roastery to all cafes
for roastery in roasting_cost_light.keys():
    model.addConstr(y_light_roastery[roastery] == sum(z_light_roastery_cafe[roastery, cafe] for cafe in
light_coffee_needed_for_cafe.keys()))

# Sum of dark coffee roasted at each roastery = sum of dark coffee shipped from that roastery to all cafes
for roastery in roasting_cost_dark.keys():
    model.addConstr(y_dark_roastery[roastery] == sum(z_dark_roastery_cafe[roastery, cafe] for cafe in
dark_coffee_needed_for_cafe.keys()))

# Sum of light coffee shipped to each cafe from all roasteries ≥ light coffee needed for that cafe
for cafe in light_coffee_needed_for_cafe:
    model.addConstr(sum(z_light_roastery_cafe[roastery, cafe] for roastery in roasting_cost_light.keys()) >=
light_coffee_needed_for_cafe[cafe])

# Sum of dark coffee shipped to each cafe from all roasteries ≥ dark coffee needed for that cafe
for cafe in dark_coffee_needed_for_cafe:
    model.addConstr(sum(z_dark_roastery_cafe[roastery, cafe] for roastery in roasting_cost_dark.keys()) >=
dark_coffee_needed_for_cafe[cafe])

# Objective: Minimize the total cost
total_cost = model.addVar(vtype=GRB.CONTINUOUS, name="total_cost")
model.addConstr(total_cost ==
    sum(shipping_cost_from_supplier_to_roastery[supplier, roastery] * x_supplier_roastery[supplier, roastery] for
supplier, roastery in shipping_cost_from_supplier_to_roastery.keys()) +
    sum(roasting_cost_light[roastery] * y_light_roastery[roastery] for roastery in roasting_cost_light.keys()) +
    sum(roasting_cost_dark[roastery] * y_dark_roastery[roastery] for roastery in roasting_cost_dark.keys()) +
    sum(shipping_cost_from_roastery_to_cafe[roastery, cafe] * z_light_roastery_cafe[roastery, cafe] for roastery, cafe
in shipping_cost_from_roastery_to_cafe.keys()) +
    sum(shipping_cost_from_roastery_to_cafe[roastery, cafe] * z_dark_roastery_cafe[roastery, cafe] for roastery, cafe in
shipping_cost_from_roastery_to_cafe.keys())
)

model.setObjective(total_cost, GRB.MINIMIZE)

# Optimize model
model.optimize()
```

Figure 25: Output of Code Generator for Coffee after switching the solver to MILP, using GPT-4o.

### A.10.4 Paraphrased Prompts and Performance

To prove the flexibility of our prompt, we also paraphrase our NL task description and re-test the framework with a paraphrased description. The paraphrasing is performed by LLMs.

One example paraphrased description is:

```
In this blocksworld problem, a robot arm can perform four actions: pickup
    , putdown, stack, and unstack. The environment consists of blocks
    that can be stacked, a single-block capacity arm, and a table.
Pickup: The arm can lift a block if it's clear, on the table, and the arm
     is empty. This results in the arm holding the block, which is no
    longer on the table or clear.
Putdown: If the arm is holding a block, it can place it on the table.
    This leaves the arm empty and the block on the table and clear.
Stack: The arm can place a block it's holding onto another clear block.
    This empties the arm, makes the top block clear and on the bottom
    block, while the bottom block becomes unclear.
Unstack: If a clear block is on another block and the arm is empty, it
    can lift the top block. This results in the arm holding the top block
     (no longer clear or on the bottom block), while the bottom block
    becomes clear.
```

With LLM-paraphrased random task descriptions, we test on 50 queries in Blockworld with Claude 3.5 Sonnet and shows LLMFP is still able to correctly generate 46/50 plans, reaching a high optimal rate of 92%, significantly outperforming baselines. This shows our framework is not sensitive to the specific wordings of the task description, as long as they have adequate information. We can add more paraphrasing examples to show the robustness of LLMFP to different user inputs, if the reviewer finds it helpful to show the generalizability of LLMFP.

Table 15: Optimal rate (%) comparison of LLMFP with baselines with paraphrased prompts on Blocksworld with Claude 3.5 Sonnet

| Direct$_{\text{GPT-4o}}$ | CoT$_{\text{GPT-4o}}$ | Code$_{\text{GPT-4o}}$ | Code_SMT$_{\text{GPT-4o}}$ | LLMFP $_{\text{GPT-4o}}$ |
|---|---|---|---|---|
| 32.0 | 46.0 | 0.0 | 0.0 | 92.0 |

