# OpenReview forum: "Planning Anything with Rigor: General-Purpose Zero-Shot Planning with LLM-based Formalized Programming"
_ICLR.cc/2025/Conference — ICLR 2025 Poster_

### Official Review · Reviewer_FxuD · 2024-10-23

**Soundness:** 3
**Presentation:** 3
**Contribution:** 2
**Rating:** 6
**Confidence:** 4

**Summary:**

This paper provides the LLM prompting-based framework for planning tasks. The main contribution lies in its use of prompt and pipeline templates, which can be used across various planning tasks. Here, planning problems from various domains were considered, and planning is treated as an optimization problem. In summary, LLMs are used as an optimizer. They used the formal planner to achieve the goal of planning as already shown in the previous works that LLM still lacks the coherent reasoning needed for planning. The main contribution is an end framework that deploys a zero-shot learning approach for both single and multi-stage planning tasks. Additionally, the author claims that their framework can handle self-critique to assess the problem in planning code to change and achieve the goal. Effectiveness of framework components is supported via the ablations.

**Strengths:**

1. LLMFP introduces a new perspective on using LLMs for formal optimization-based planning, a method that significantly expands the generalizability of planning tasks.
2. Experimental results are solid, with clear evidence of performance gains across diverse tasks and models. The ablation studies reinforce the utility of the framework components, which I really liked.
3. The ability to solve multi-step, multi-constraint problems without task-specific examples or extensive prior efforts is a major step forward in the area of LLM-based planning.

**Weaknesses:**

1. Complexity in FORMULATOR: Some parts, particularly the JSON representation and the code generation steps, could be simplified. While important, the handling of different variable types and constraints might be a bit dense for readers unfamiliar with optimization theory.
2. Regrading Multi-Step Planning Problem: The predicate, object, and update structure are not clear in multi-step planning. Also image shown for this is not utilized in conveying the idea. Overall, Figure. 2 examples are not clear and make things confusing.
3. The author claims that their framework is a "general approach, which does not require task-specific examples or task-specific efforts"; however, in the paper, this statement is not supported in terms of explanations and prompt structure.
4. Some theoretical insights regarding performance would make this work more strong, right now its presented more like a experimental results.

**Questions:**

1. How LLMFP handles generalization across different planning tasks. Please correct me, but it seems that we need a very Elaborate prompt with a high level of detail for each task.
2. In section 3.4 (code generator), readers can Benefit from prior work such as "CAPE: Corrective Actions from Precondition Errors using Large Language Models" and "CoT-TL: Temporal Knowledge Representation of Natural Language Planning Task for Autonomous Agents using Chain-Of-Thought." Or is LLMFP doing differently compared to the above works; if yes, explain or not, and make sure to provide proper background.

---

> ### Author Response · Authors · 2024-11-23
> **Response to Reviewer FxuD 1/3**
>
> We thank the reviewer for the valuable comments and suggestions! We provide some clarifications,  discussions, and experiments regarding the weakness and questions proposed by the reviewer, and experiments suggested by other reviewers (Please see the Revision Summary).  We also updated a revised draft and colored the modifications/additional discussions with blue.
>
> **FxuD-Q1: Complexity in FORMULATOR**
>
> FxuD-A1:
> Thanks for the suggestion. We have rewritten the FORMULATOR section. In particular,
> * We rewrote the first two paragraphs with a clearer way of introducing the JSON representation and field.
> * We cut down the paragraph Single-Step Multi-Constraint Problem to make it less dense for the readers.
>
> Please refer to our updated draft. All the changes are highlighted in blue.
>
> **FxuD-Q2: Unclear presentation in ‘Multi-Step Planning Problem’ section**
>
> FxuD-A2:
> Thanks for the suggestion. We have modified the ‘Multi-Step Planning Problem’ section. In particular,
> * We rewrote the start of the paragraph with a clearer explanation of what different stages mean, and disambiguated them from the fields.
> * We replaced Figure 2 with two clear, to-the-point examples of the JSON representation. Figure 2 can now better assist the explanation.
>
> Please refer to our updated draft. All the changes are highlighted in blue.
>
> **FxuD-Q3: The statement ‘Our approach does not require task-specific examples or task-specific efforts’ is not supported**
>
> FxuD-A3:
> We would like to clarify that the only task-specific information is the user description of the planning problem and the user’s question (i.e., lines 188-196). This is the minimum information needed to define and understand the planning problem.
>
> Other than that, all the prompts for different modules are task-agnostic, i.e. the same prompt works for all planning problems. To emphasize this, we added sentences like ‘which is task-agnostic’ or ‘which is invariant across tasks’. It is worth emphasizing that although our prompts include in-context examples, those examples are from a fixed task that is unrelated to the target task. In our implementation, we used examples from travel salesman, object selection, and logistics problems, which do not belong to any planning task we evaluate. Please refer to the detailed prompts listed in A.10. Those are the prompts that are fixed for all the different planning tasks.

---

> ### Author Response · Authors · 2024-11-23
> **Response to Reviewer FxuD 2/3**
>
> **FxuD-Q4: Theoretical insights regarding performance**
>
> FxuD-A4: Our insight is that LLMs are good at understanding the syntax and semantics of planning problems as optimization problems but are not good at solving optimization problems directly. Specifically,
>
> **Next-token prediction is fundamentally different from deterministic algorithms for optimization.** There’s a growing belief that next token prediction cannot truly model human thought and cannot support human-like capabilities of understanding a problem, imagining, curating, and backtracking plans before executing [1-3]. Specifically, the claim that next token predictions are “ill-suited for planning tasks” is supported by works [4-7], which tested the planning capabilities of LLMs on various planning tasks. These works empirically show that in addition to identifying patterns in language and predicting the next word in a sequence, LLMs still can not truly understand a problem and thus do not have the capability to perform intense calculations to optimize for any objectives. Thus, this is a major reason why baselines are not capable of solving the complex planning problems in our paper. However, since LLMFP teaches LLMs to build the optimization problem step by step and calls the external solver to solve for a plan, this bypasses the need to devise a plan by LLMs themselves.
>
> **LLMs cannot understand and solve an optimization problem. To support this claim,** we conduct an experiment on the Coffee task that, instead of using natural language task descriptions as inputs, we directly map this Coffee task to an optimization problem and use the formal mathematical definition (refer to page 16 for the detailed formal definnition) of this problem as the inputs to LLMs. Thus, LLMs do not need to understand the problem and find the underlying constraints, as a formal definition is given and could be directly solved.
>
> We tested Direct with the most powerful LLM OpenAI o1-preview model on all queries of Coffee, which only achieves an optimal rate of **34.2%**. Compared to its 25.9% optimal rate with natural language task description, this is not a significant improvement, given all goals and constraints are clearly formally specified in the new setting. This is consistent with the conclusion that LLMs still cannot solve optimization problems by themselves, even given a formal representation. **LLMFP enables LLMs to formalize planning problems as optimization problems.**  Since SMT solvers can guarantee to return correct answers given correct input, the high optimal and success rate of LLMFP indicates that LLMFP allows LLM to parse the correct syntax and semantics information of a planning problem from its natural language description to a formal mathematical description. Such translation is also non-trivial when no task-specific examples are provided. As shown in our newly added baseline approach Code_SMT as shown in Table 1 and 2, when we directly ask LLMs to translate and encode the natural language task description in an SMT format, the optimal rate is low, with an average of 2.7% and 62.4% for multi-constraint tasks, and 1.0% and 0.0% for multi-step tasks across two LLMs GPT-4o and Claude 3.5 Sonnet.

---

> > ### Author Response · Authors · 2024-11-23
> > **Response to Reviewer FxuD 3/3**
> >
> > **FxuD-Q5: How LLMFP handles generalization across different planning tasks? It seems that we need a very Elaborate prompt with a high level of detail for each task.**
> >
> > FxuD-A5:
> > As we have clarified in FxuD-Q3, the only task-specific information needed is the user description of the task, the background information, and the user questions. The user description of the task indeed needs to be elaborate and accurate, **because this is the only source of information about the planning task. Otherwise, the planning problem will be ill-defined.** Our method only requires the minimum necessary task-specific information, compared with existing works that require additional human efforts such as providing task-specific examples, human-designed rules or critics, or even action-wise human feedback.
> >
> > Other than that, all the prompts to the LLMs, despite being elaborate and detailed, are **task-agnostic**. The reason why LLMFP can generalize across tasks is that it casts the planning problem into a constrained optimization problem, the processing of which is generic and task-independent.
> >
> > **FxuD-Q6: Prior works for section 3.4**
> >
> > FxuD-A6: We thank the reviewer for referring to two relevant papers in the field. CAPE provided self-corrections to generated plans, which is a good resource to include in section 2.1 (LLMs for Planning). CoT-TL focuses on translating natural language specifications into LTL representations and very briefly mentioned they could generate codes to solve translated representations with Gurobi solver, which is a good resource to include in section 2.2 (LLM+Solver). To provide more background, we included these papers in section 2 of the revised version.
> >
> >
> > However, we believe LLMFP is doing differently compared to the above works for code generation itself. Our understanding is that CAPE focuses on self-corrections with error feedback, which does not include discussions of code generation. Although CoT-TL briefly mentioned that they leveraged LLM to encode the LTL formula as mixed-integer linear constraints for a demo in section IV.D, their focus is the translation process and did not provide detailed discussions or examples describing how they accomplish code generation. Some other works mentioned in Section 2.2 also have the representation -> solver code part, but the code generation process of LLMFP is novel and different to these works for a major reasons: the code generation in LLMFP is completely zero-shot, with no in-context examples or task-specific examples, and completely based on the JSON representation created in Formulator. This JSON representation acts like a plan for code generation, as it includes every variable or step needed to encode this problem into codes and specifies what are the values, sources, specific information, etc., clearly.
> >
> >
> >
> > [1] Bachmann, Gregor, and Vaishnavh Nagarajan. "The pitfalls of next-token prediction." arXiv preprint arXiv:2403.06963 (2024).
> >
> > [2] Bubeck, Sébastien, et al. "Sparks of artificial general intelligence: Early experiments with gpt-4." arXiv preprint arXiv:2303.12712 (2023).
> >
> > [3] LeCun, Yann. "Do large language models need sensory grounding for meaning and understanding." Workshop on Philosophy of Deep Learning, NYU Center for Mind, Brain, and Consciousness and the Columbia Center for Science and Society. 2023.
> >
> > [4] Momennejad, Ida, et al. "Evaluating cognitive maps and planning in large language models with CogEval." Advances in Neural Information Processing Systems 36 (2024).
> >
> > [5] Valmeekam, Karthik, Matthew Marquez, and Subbarao Kambhampati. "Can large language models really improve by self-critiquing their own plans?." arXiv preprint arXiv:2310.08118 (2023).
> >
> > [6] Valmeekam, Karthik, et al. "Planbench: An extensible benchmark for evaluating large language models on planning and reasoning about change." Advances in Neural Information Processing Systems 36 (2024).
> >
> > [7] Valmeekam, Karthik, et al. "On the planning abilities of large language models-a critical investigation." Advances in Neural Information Processing Systems 36 (2023): 75993-76005.

---

> > > ### Comment · Reviewer_FxuD · 2024-11-25
> > > **Response to Authors comment**
> > >
> > > Thank you for providing the detailed corrections in the revised version of the paper and mentioning the theoretical insights about the results. In the final version of the paper, it would be great to see the theoretical insights as well apart from comments provided here. As I really appreciate this and these the important statement not just experimental results.
> > > However, I am still not convinced with your explanation regarding  W3 "general approach, which does not require task-specific examples or task-specific efforts". Upon revisiting the appendix section I observed that for each setup different prompts are created say multiple code generator prompts and seems claim is understating the prompt engineering done to achieve the goal. Don't we need new prompt engineering for each task, each section and each domain? If yes then I would say framework is not very generic as stated. Also this is stated in your response
> > > >"The user description of the task indeed needs to be elaborate and accurate".
> > >
> > > Can you point me to other existing work prompts which require more elaborate prompting compared to your work and differences compared to your prompt design?

---

> > > > ### Author Response · Authors · 2024-11-26
> > > > **Response to Reviewer FxuD - Round 2  1/4**
> > > >
> > > > Thank the reviewer for your reply! We are glad that you appreciate the theoretical insights. We revised the paper to include the analysis and the additional experiment showing LLMs cannot directly solve optimization problems in Appendix A.5.4. Currently, it is a little long to fit in the main text, but we would love to move the analysis to the main texts when we put together the final version.
> > > >
> > > > We believe the reviewer has misunderstandings about LLMFP being a general framework. First, we want to clarify the roles of different components in LLMFP. Then, we use a concrete example to demonstrate that **(a) we do not need new prompt engineering for each section and domain, (b) task-specific prompts for each task have very low engineering efforts, (c) LLMFP has the lowest prompting engineering effort compared to other approaches, and (d) LLMFP has high user prompt robustness and can handle a variety of different prompts from different users.**
> > > >
> > > > **PART (a): Our prompt structure**\
> > > > In LLMFP, we have the following prompt components:
> > > >
> > > > 1. $\color{#E24A33}{\textsf{\textit{LLMFP key components}}}$: These prompts are the key part of LLMFP to understanding how to formulate planning problems as optimization problems. These prompts are **task-agnostic** and are embedded in LLMFP. They are the same files that **do not need to be modified at all** when solving different task domains. These include 5 components, Definer, Formulator, Code Generator, Result Formatter, Self assess & Modification, and corresponding to the prompts on Pages 43-52 in the paper. We have **only one prompt for each component**.\
> > > > We call these prompts ‘$\color{#E24A33}{\textsf{prompt skeleton}}$’ in the clarifications below.
> > > >
> > > > 2. $\color{#2986cc}{\textsf{\textit{Task description input}}}$: This gives a basic setup of the planning problem given by the user, for example, it includes the objects, actions, preconditions and effects of actions. They only need to be accurate since this is all the information LLMFP knows about the planning problem to be solved. In this paper, we considered 9 task domains. Therefore, there are 9 task description prompts.
> > > >
> > > > 3. $\color{#674ea7}{\textsf{\textit{API input}}}$: This gives a list of background information about the tasks as well as information on APIs that the planner can use. This is the same for all multi-step tasks. For multi-constraint tasks, this serves as a supplement to $\color{#2986cc}{\textsf{task descriptions}}$ (2), for example, specifying the exact number for shipping cost of coffee.
> > > >
> > > > 4. $\color{#38761d}{\textsf{\textit{Query input}}}$: This is the question raised by the user that (a) describes the initial and/or goal states, or (b) adds or modifies existing requirements of the tasks of a particular task. Each task domain can have 21-602 queries that we use to evaluate the success rate.
> > > >
> > > > The prompts for 2,3,4 are task domain-specific, and are on Pages 26-33 in the paper. \
> > > > They contain information about the task and user requirements, and are the basics for any planning framework or solver.
> > > >
> > > > For each planning instance, the prompt follows the [$\color{#E24A33}{\textsf{prompt skeleton}}$ (1) + $\color{#2986cc}{\textsf{task descriptions}}$ (2) +$\color{#674ea7}{\textsf{APIs}}$ (3) + $\color{#38761d}{\textsf{query}}$ (4)] pattern. Again $\color{#E24A33}{\textsf{prompt skeleton}}$ (1) is the same everywhere.

---

> > > > > ### Author Response · Authors · 2024-11-26
> > > > > **Response to Reviewer FxuD - Round 2 2/4**
> > > > >
> > > > > **PART (b): Our task-specifc description - an example**\
> > > > > Next, to further clarify the task-specific efforts for prompt design, we use the Blocksworld task as an example.\
> > > > > For multi-step problems, $\color{#2986cc}{\textsf{task descriptions}}$ and $\color{#38761d}{\textsf{queries}}$ are the **only task-specific prompts** needed from users. For the blocksworld problem, the $\color{#2986cc}{\textsf{\textbf{task descriptions}}}$ is:
> > > > >
> > > > >     “The robot has four actions: pickup, putdown, stack, and unstack. The domain assumes a world where there are a set of blocks that can be stacked on top of each other, an arm that can hold one block at a time, and a table where blocks can be placed.
> > > > >     The actions defined in this domain include:
> > > > >     pickup: allows the arm to pick up a block if the block is clear, the block is on_table, and the arm is empty. After the pickup action, the arm will be holding the block thus not empty, and the block will no longer be on_table or clear.
> > > > >     putdown: allows the arm to put down a block if the arm is holding a block. After the putdown action, the arm will be empty thus not holding the block, and the block will be on_table and clear.
> > > > >     stack: allows the arm to stack a block on top of another block if the arm is holding the top block and the bottom block is clear. After the stack action, the arm will be empty thus not holding the block, the top block will be clear and on top of the bottom block, and the bottom block will no longer be clear. unstack: allows the arm to unstack a block from on top of another block if the top block is on the bottom block, the arm is empty, and the top block is clear. After the unstack action, the arm will be holding the top block thus not empty, the top block will no longer be on top of the bottom block and not clear, and the bottom block will be clear.”,
> > > > >
> > > > > **and one example** $\color{#38761d}{\textsf{\textbf{query}}}$ **is:**
> > > > >
> > > > >     You have 4 blocks.
> > > > >     b is on top of c.
> > > > >     c is on top of d.
> > > > >     d is on top of a.
> > > > >     a is on the table.
> > > > >     b is clear.
> > > > >     Your arm is empty.
> > > > >     Your goal is to move the blocks.
> > > > >     a should be on top of c.
> > > > >     d should be on top of a.”
> > > > >
> > > > > Please note that the above task-specific effort is the minimum effort needed to describe the problem, without which the planning problem would become ill-defined. We will show that this does not need heavy description engineering with a paraphrasing experiment.
> > > > >
> > > > > Other than that, all the prompt skeletons for different components, including the prompts for Definer, Formulator, Code Generator, Result Formatter, Self assess & Modification, are task-agnostic. The corresponding prompts are listed in Pages 43-52.

---

> > > > > > ### Author Response · Authors · 2024-11-26
> > > > > > **Response to Reviewer FxuD - Round 2 3/4**
> > > > > >
> > > > > > **PART (c): Comparison of task-specific efforts with other baselines**\
> > > > > > As a comparison, we compare the task-specific part of our prompt with that of another famous paper LLM+P [1], which translates PDDL natural language(NL) descriptions into PDDL problem files with LLMs, and solves them with a PDDL solver. **We need **[$\color{#2986cc}{\textsf{\textbf{NL task descriptions}}}$ + $\color{#38761d}{\textsf{\textbf{query}}}$]**, where LLM+P needs [PDDL task description + 1 task-specific NL-> PDDL translation example + query].**
> > > > > >
> > > > > > * We summarize the task-specific part of their prompts here:
> > > > > >     * To solve a problem, it first takes a task-specific NL->PDDL translation example:
> > > > > >
> > > > > >             ​​An example planning problem is:
> > > > > >             You have 5 blocks.
> > > > > >             b2 is on top of b5.
> > > > > >             b5 is on top of b1.
> > > > > >             b1 is on top of b4.
> > > > > >             b3 is on top of b2.
> > > > > >             b4 is on the table.
> > > > > >             b3 is clear.
> > > > > >             Your arm is empty.
> > > > > >             Your goal is to move the blocks.
> > > > > >             b4 should be on top of b3.
> > > > > >
> > > > > >             The problem PDDL file to this problem is:
> > > > > >             (define (problem BW-rand-5)
> > > > > >             (:domain blocksworld-4ops)
> > > > > >             (:objects b1 b2 b3 b4 b5 )
> > > > > >             (:init
> > > > > >             (arm-empty)
> > > > > >             (on b1 b4)
> > > > > >             (on b2 b5)
> > > > > >             (on b3 b2)
> > > > > >             (on-table b4)
> > > > > >             (on b5 b1)
> > > > > >             (clear b3)
> > > > > >             )
> > > > > >             (:goal
> > > > > >             (and
> > > > > >             (on b4 b3))
> > > > > >             )
> > > > > >             )
> > > > > >
> > > > > >     * Then it provides the NL description of the query it wants to solve and prompts LLMs to translate it to a PDDL problem file.
> > > > > >
> > > > > >             You have 3 blocks.
> > > > > >             b2 is on top of b3.
> > > > > >             b3 is on top of b1.
> > > > > >             b1 is on the table.
> > > > > >             b2 is clear.
> > > > > >             Your arm is empty.
> > > > > >             Your goal is to move the blocks.
> > > > > >             b2 should be on top of b3.
> > > > > >             b3 should be on top of b1.
> > > > > >
> > > > > >     * After translation, it provide this translated problem file and a **required PDDL domain file** into the solver:
> > > > > >
> > > > > >             (define (domain blocksworld-4ops)
> > > > > >             (:requirements :strips)
> > > > > >             (:predicates (clear ?x)
> > > > > >                         (on-table ?x)
> > > > > >                         (arm-empty)
> > > > > >                         (holding ?x)
> > > > > >                         (on ?x ?y))
> > > > > >
> > > > > >             (:action pickup
> > > > > >             :parameters (?ob)
> > > > > >             :precondition (and (clear ?ob) (on-table ?ob) (arm-empty))
> > > > > >             :effect (and (holding ?ob) (not (clear ?ob)) (not (on-table ?ob))
> > > > > >                         (not (arm-empty))))
> > > > > >
> > > > > >             (:action putdown
> > > > > >             :parameters  (?ob)
> > > > > >             :precondition (holding ?ob)
> > > > > >             :effect (and (clear ?ob) (arm-empty) (on-table ?ob)
> > > > > >                         (not (holding ?ob))))
> > > > > >
> > > > > >             (:action stack
> > > > > >             :parameters  (?ob ?underob)
> > > > > >             :precondition (and (clear ?underob) (holding ?ob))
> > > > > >             :effect (and (arm-empty) (clear ?ob) (on ?ob ?underob)
> > > > > >                         (not (clear ?underob)) (not (holding ?ob))))
> > > > > >
> > > > > >             (:action unstack
> > > > > >             :parameters  (?ob ?underob)
> > > > > >             :precondition (and (on ?ob ?underob) (clear ?ob) (arm-empty))
> > > > > >             :effect (and (holding ?ob) (clear ?underob)
> > > > > >                         (not (on ?ob ?underob)) (not (clear ?ob)) (not (arm-empty)))))
> > > > > >
> > > > > > * Our prompts are mostly natural language descriptions of the task, but LLM+P needs very structured prompts including task-specific PDDL problem file translation examples and the PDDL domain files. The files needed in LLM+P need to be strictly correct because they are input to the PDDL solver. One format/syntax error would result in failures in calling the solver. **The need of task-specific example and PDDL formatted files are more strict and hard to obtain from a non-expert.**
> > > > > >
> > > > > > * As a comparison,  NL task description is **more flexible**. We provide experiments below to show that **paraphrased natural language task descriptions will not change the performance of LLMFP**.
> > > > > >
> > > > > > * In addition to LLM+P, other methods that try to solve this problem also require **even more** task-specific in-context examples and existing PDDL domain files: see Figure 2 of [2] (**>500 lines of task-specific examples** for Blocksworld, vs. LLMFP only needs 10 sentences of task description) and Figure 1 of [3]. They all clearly need more prompt engineering effort.

---

> > > > > > > ### Author Response · Authors · 2024-11-26
> > > > > > > **Response to Reviewer FxuD - Round 2 4/4**
> > > > > > >
> > > > > > > **PART (d): The flexibility of our task-specific effort**\
> > > > > > > To prove the flexibility of our prompt, we also **paraphrase our NL task description and re-test the framework with a paraphrased description**. The paraphrasing is performed by LLMs.\
> > > > > > > One example paraphrased description is:
> > > > > > >
> > > > > > >     In this blocksworld problem, a robot arm can perform four actions: pickup, putdown, stack, and unstack. The environment consists of blocks that can be stacked, a single-block capacity arm, and a table.
> > > > > > >     Pickup: The arm can lift a block if it's clear, on the table, and the arm is empty. This results in the arm holding the block, which is no longer on the table or clear.
> > > > > > >     Putdown: If the arm is holding a block, it can place it on the table. This leaves the arm empty and the block on the table and clear.
> > > > > > >     Stack: The arm can place a block it's holding onto another clear block. This empties the arm, makes the top block clear and on the bottom block, while the bottom block becomes unclear.
> > > > > > >     Unstack: If a clear block is on another block and the arm is empty, it can lift the top block. This results in the arm holding the top block (no longer clear or on the bottom block), while the bottom block becomes clear.
> > > > > > >
> > > > > > > With LLM-paraphrased random task descriptions, we test on 50 queries in Blockworld with Claude 3.5 Sonnet and shows LLMFP is still able to correctly generate 46/50 plans, reaching a high optimal rate of **92%, significantly outperforming baselines**. This shows our framework is **not sensitive** to the specific wordings of the task description, as long as they have adequate information. We include the result and analysis in Appendix A.10.4. We can add more paraphrasing examples to show the robustness of LLMFP to different user inputs, if the reviewer finds it helpful to show the generalizability of LLMFP.
> > > > > > >
> > > > > > > Table: Optimal rate (%) comparison of LLMFP with baselines with paraphrased prompts on Blocksworld with Claude 3.5 Sonnet
> > > > > > > | Direct | CoT| Code| Code_SMT | LLMFP |
> > > > > > > |--------------------------|-----------------------|-----------------------|---------------------------|-------------------------|
> > > > > > > | 32.0                      | 46.0                      | 0.0                   | 0.0                       | **92.0**                   |
> > > > > > >
> > > > > > > **PART (e): Clarification on Code Generator prompt**\
> > > > > > > In the comment you mentioned that ‘Upon revisiting the appendix section I observed that for each setup different prompts are created say multiple code generator prompts’ We want to clarify that we only have **one** Code Generator prompt, which is listed on pages 50-51. We are wondering if any part of the paper makes the reviewer feel there are multiple, and we would love to provide any further clarifications to avoid the confusion.
> > > > > > >
> > > > > > > Hope the explanation, examples, and comparisons can make things clearer. We would love to provide further clarification! Thank the reviewer for careful consideration and helpful feedback!
> > > > > > >
> > > > > > > [1] Liu, Bo, et al. "LLM+P: Empowering large language models with optimal planning proficiency." arXiv preprint arXiv:2304.11477 (2023).\
> > > > > > > [2] ISR-LLM: Iterative Self-Refined Large Language Model for Long-Horizon Sequential Task Planning\
> > > > > > > [3] Xie, Yaqi, et al. "Translating natural language to planning goals with large-language models." arXiv preprint arXiv:2302.05128 (2023).

---

> > > > > > > > ### Author Response · Authors · 2024-11-27
> > > > > > > > **Message from authors -- last day of draft revision**
> > > > > > > >
> > > > > > > > Dear reviewer FxuD,
> > > > > > > >
> > > > > > > > We sincerely appreciate your time and efforts in evaluating our paper. Since this is the last day we can revise the draft, we would like to confirm whether our responses have effectively addressed your concerns. To summarize, in our latest response, we clarified in detail why LLMFP is a generic framework and added experiments to show that LLMFP is not sensitive to specific wordings of task descriptions.
> > > > > > > >
> > > > > > > > If there are any additional points you'd like us to discuss or consider, please do not hesitate to let us know. Your insights have been invaluable, and we're grateful for your feedback on our work. We look forward to further discussions with you!
> > > > > > > >
> > > > > > > > Best,\
> > > > > > > > Authors of paper 6029

---

> > > > > > > > ### Comment · Reviewer_FxuD · 2024-11-28
> > > > > > > > **Response to Authors Round 2**
> > > > > > > >
> > > > > > > > Thank you to the authors for providing detailed answers and comparisons regarding LLM+P. I concur with the authors that LLM+P requires more domain-specific prompt engineering compared to this work. However, the current work introduces additional overhead compared to LLM+P. For instance, on the Gripper dataset, LLM+P achieves quite good results in terms of success rate with GPT-4 instead of GPT-4o. I appreciate the robustness of the prompting system with respect to paraphrasing. This work will be a valuable addition to the robotics community. I will maintain my current rating.

---

> > > > > > > > > ### Author Response · Authors · 2024-11-29
> > > > > > > > > **Response to Reviewer FxuD - Round 3**
> > > > > > > > >
> > > > > > > > > Thank the reviewers for the reply! However, we respectfully disagree with the claim *“However, the current work introduces additional overhead compared to LLM+P.  For instance, on the Gripper dataset, LLM+P achieves quite good results in terms of success rate with GPT-4 instead of GPT-4o”*. We would like to explain why this is inaccurate and should not be considered as a source of criticism of our work.
> > > > > > > > > * **LLMFP does not necessarily need stronger models.** LLMFP can also perform well on GPT-4. To show this, we perform an experiment where we tested LLMFP on 10 queries of Gripper and 10 queries with Blocksworld with GPT-4. LLMFP could deliver optimal plans for 7 queries for Gripper and 8 queries for Blocksworld (yielding success rates of 70% and 80% respectively). We are happy to extend this experiment to a larger scale, if you believe this could improve our work.
> > > > > > > > > * **LLM+P is a domain-specific solver, needs domain-specific examples, and can only solve PDDL problems.** In comparison, LLMFP is a general planner and does not need domain-specific examples. It is not equitable to directly compare the performance of LLM+P and LLMFP on one specific task domain.
> > > > > > > > >     * **LLM+P cannot solve non-PDDL problems**, such as the multi-constraint problems in our paper (details provided in response qLBs-A1). LLMFP can solve a broad spectrum of planning problems, including PDDL problems without domain-specific examples.
> > > > > > > > >     * LLM+P requires domain-specific effort **for every task**, but LLMFP is a generic approach.
> > > > > > > > >     * **LLMs only act as a translator in LLM+P, but LLMFP enables LLMs to understand the problem and build an optimization problem by themselves.**
> > > > > > > > > * **GPT-4 is more expensive and slower than GPT-4o [1].** For LLM-based planning frameworks, runtime and cost are important factors. We do not think our contribution should be underestimated by using a cheaper and faster model.
> > > > > > > > >
> > > > > > > > > We hope the answers can address the reviewer’s concern. We would love to have further discussions with the reviewer!
> > > > > > > > >
> > > > > > > > > [1] https://openai.com/api/pricing/

---

> > > > > > > > > > ### Author Response · Authors · 2024-12-01
> > > > > > > > > > **Response to Reviewer FxuD - Round 3 -Followup**
> > > > > > > > > >
> > > > > > > > > > Dear reviewer FxuD,
> > > > > > > > > >
> > > > > > > > > > We would like to follow up on our pending discussion about the ‘overhead’ of our proposed method.
> > > > > > > > > >
> > > > > > > > > > In our last response, we show LLMFP does not necessarily need stronger models by testing it with GPT-4 for two multi-step tasks, Blocksworld and Gripper. We would like to provide some more experiments to support this claim. We performed additional experiments to test it with GPT-4 on three more multi-constraint tasks, Workforce, Task Allocation, and Warehouse. We tested 10 queries from each domain and got an optimal rate of 80%, 70%  and 70% respectively.
> > > > > > > > > >
> > > > > > > > > > We would like to confirm whether our responses have effectively addressed your concerns from the last and follow-up discussions. If there are any additional points you'd like us to discuss or consider, please do not hesitate to let us know. As the discussion period ends on Dec 2, we look forward to timely discussions with you!
> > > > > > > > > >
> > > > > > > > > > Best,\
> > > > > > > > > > Authors of paper 6029

---

### Official Review · Reviewer_dysu · 2024-11-02

**Soundness:** 3
**Presentation:** 3
**Contribution:** 3
**Rating:** 6
**Confidence:** 3

**Summary:**

The paper presents a framework that pairs LLMs with optimization tools to solve planning tasks without using task-specific knowledge. The authors define consecutive stages of reasoning that, generally speaking, consist of understanding, coding, and refining. For each stage, they discuss the prompting, formatting, and other relevant decisions. Through experimental validation, they show that LLMFP outperforms baselines in 9 domains.

**Strengths:**

I like the general idea and the presented approach. One could argue that it is simply a combination of prompt engineering and the incorporation of external tools. However, showing an effective way of doing this can be a significant contribution.

The baselines and ablations are well-chosen for evaluating the performance of LLMFP.

The paper is written very clearly, making it easy to read. The figures are well-chosen (particularly Figure 1), they are helpful in understanding the pipeline. I like the section structure and the focus on key takeaways when discussing experimental results. Most of my questions that arose while reading the text were addressed in later sections.

**Weaknesses:**

The goal stated in the introduction is "Can we build a universal LLM-based planning system that can solve complex planning problems without task-specific efforts?". However, my main concern is whether the tasks used for experiments are indeed complex planning problems. Specifically, the 5 multi-constraint problems resemble simply optimization problems rather than planning problems. Hence it's quite clear that adding an external optimizer to LLM would be much better than just using LLM. On the other hand, the multi-step problems seem to be rather simple and the main difficulty is to understand what we have to do rather than finding a good solution. Hence, I suggest adding at least one multi-step domain with high underlying complexity (e.g. Sokoban). If I missed something and some of your environments are actually NP-hard (or hard in any other reasonable sense), it should be remarked in the paper.

Since the method you propose is clearly subject to a tradeoff between performance and computation time, there should be a discussion of that. What's the wall time of LLMFP compared to the baselines? What's the cost of using SMT compared to querying LLM?

The description of baselines should be extended a bit. Are they prompted vanilla models plus the components described in lines 410-416, or do they also include other components, e.g. formatter? Also, the Code variant uses pure Python, but for a completely fair comparison you should also add variant which is forced to use SMT like LLMFP does. After reading the prompts used, it's also not clear to me whether they are explicitly instructed to provide optimal solutions, which is captured by the metrics. Also, I suggest discussing the failure modes of the baselines (in the Appendix).

**Questions:**

1. What are the most common failure modes of the baselines?

2. Are the baselines prompted vanilla models plus the components described in lines 410-416, or do they also include other components, e.g. formatter?

3. What are the success rates of the tested methods? Do they all achieve 100% and the question is only whether the solution is optimal, or do some methods fail to solve some instances at all?

4. What's the wall time of LLMFP compared to the baselines?

5. Are the methods explicitly instructed to provide optimal solutions?

**Details Of Ethics Concerns:**

I have no concerns.

---

> ### Author Response · Authors · 2024-11-23
> **Response to Reviewer dysu 1/6**
>
> We thank the reviewer for the constructive comments and helpful feedback! We are pleased to see that the reviewer appreciates the idea, approach, and presentation of LLMFP. You brought up some great questions and suggestions which have helped improve our work. We have added more experiments and discussions to address the concerns of the reviewer. We also updated a revised draft and colored the modifications/additional results and discussions with blue.
>
> To summarize:
> * We added experiments on **Sokoban** tasks for 15 queries
>     * LLMFP achieves optimal rates of **80%**, outperforming all baselines
> * We added a baseline **Code_SMT** that is forced to use SMT for code generation for all tasks and LLMs
>     * LLMFP outperforms Code_SMT, which achieves an average of 2.7% and 62.4% for multi-constraint tasks, and 1.0% and 0.0% for multi-step tasks across two LLMs GPT-4o and Claude 3.5 Sonnet
> * We provide results with **success rate** as the metric across all 9 tasks and for both LLMs
>     * LLMFP still outperforms the baselines
> * We added experiments that **explicitly** instruct baselines to output **optimal** solutions for multi-step problems
>     * LLMFP still outperforms the baselines
> * We added wall time comparison and cost comparison
>     * LLMFP runtime and cost are reasonable, comparable to using o1-preview
> * We added complexity analysis and failure cases analysis
>     * All multi-constraint tasks, Blocksworld, and Mystery Blocksworld are proved to be NP-Hard
>
> Detailed response:
>
> **dysu-Q1: The complexity of planning problems**
>
> dysu-A1:
> We thank the reviewer for bringing up the discussion about the complexity of our planning problems. We include a more detailed complexity discussion and put it in Appendix A.1. In short, many of the tasks we experimented on are NP-hard problems. In particular -
> * For 5 multi-constraint problems, they are **all NP-hard problems.** Specifically,
>     * We use the benchmark from [1] for the first 3 problems (Coffee, Workforce, and Facility), in which these 3 problems are built as Mixed-integer linear programming (MILP) problems. Please refer to Appendix A.1 for the formal optimization problem definition example. As MILP is known to be NP-hard, the first 3 problems are NP-hard.
>     * For the Task Allocation problem, since it is equivalent to a multi-agent traveling salesman problem(agent=robots, tasks=cities), it reduces from the classic traveling salesman problem (TSP) and thus is also NP-hard.
>     * For the Warehouse problem, as TSP is a special case when one station can be used to finish one specific task, and there are no extra stations, the Warehouse problem is at least as complex as TSP and, thus, is also NP-hard.
>
> * For multi-step problems, Blocksworld has proved to be an NP-hard problem [2], and the same is true for Mystery Blocksworld, as it is the same problem with obfuscated names. Although there is no existing proof to be NP-hard problems, Movie has 13 predicates and 9 possible actions, and Gripper has 4 types of objects (rooms, objects, robots, grippers), 4 predicates, and 3 possible actions. These show that they are not simple straightforward tasks.
>
> **dysu-Q2: More complex multi-step problem - Sokoban**
>
> dysu-A2:
> As the reviewer suggested, we tested LLMFP on the Sokoban environment. Due to the time limitations, we created a pilot evaluation set containing 15 queries describing the game setup and goals with different map sizes and number of boxes. We have five queries with 5x5 maps and 1 box, five queries with 6x6 maps and 1 box, and five queries with 5x5 maps and 2 boxes. The evaluation results are presented in the following table.
>
> Table 1: Optimal rate (%) comparison of LLMFP with baselines on Sokoban problem
> | Direct_GPT-4o | Direct_o1-preview | CoT_GPT-4o | Code_GPT-4o | Code_SMT_GPT-4o | LLMFP_GPT-4o |
> |--------------------------|----------------------------|-----------------------|-----------------------|---------------------------|-------------------------|
> | 0.0                      | 26.7                      | 0.0                   | 0.0                   | 0.0                       | **80.0**                   |
>
>
> As can be observed, LLMFP achieves a success rate of 80%, outperforming the baselines. The new results, along with other problems, showcase the potential of LLMFP to solve complex problems. We will keep working on expanding our query set and will add the full results to our paper.
>
> We added this result in Appendix A.1 LLMFP Performance on Sokoban of the revised paper, as well as the discussion of failure mode. Please refer to the revision page 16 for detailed discussions.

---

> ### Author Response · Authors · 2024-11-23
> **Response to Reviewer dysu 2/6**
>
> **dysu-Q3: Time and cost comparison of baselines and LLMFP**
>
> dysu-A3:
> In Appendix A.4 we included the time and cost analysis for LLMFP. As the reviewer suggested, we add time and cost comparisons for all methods.
>
> The following tables show the wall time comparison of all methods for GPT-4o on 9 tasks (also in Appendix A.4). From the results, we observe that the time taken for LLMFP, although longer than most of the baselines, is within a reasonable range. Especially for multi-constraint problems, it is shorter than Direct with o1-preview because of the inherent difficulty for LLMs to solve these combinatorial optimization problems.
>
> Table 2: Average wall time (s) per query comparison for 5 multi-constraint problems with GPT-4o.
> | Method                | Coffee | Workforce | Facility | Task Allocation | Warehouse | Average |
> |-----------------------|--------|-----------|----------|-----------------|-----------|---------|
> | Direct_GPT-4o        | 8.8    | 2.2       | 2.1      | 1.8             | 0.9       | 3.2     |
> | Direct_o1-preview     | 104.2  | 63.9      | 77.7     | 70.5            | 63.7      | 76.0    |
> | CoT_GPT-4o           | 16.9   | 12.0       | 6.0      | 9.6             | 7.4       | 10.4     |
> | Code_GPT-4o          | 30.6   | 10.0      | 8.2      | 5.7             | 7.1       | 12.3    |
> | Code_SMT GPT-4o      | 30.0   | 15.3      | 10.3     | 15.0            | 8.3       | 15.8    |
> | LLMFP_GPT-4o         | 87.1   | 55.1      | 29.9     | 62.3            | 28.9      | 52.7    |
>
> Table 3: Average wall time (s) per query comparison for 4 multi-step problems with GPT-4o.
> | Method                | Blocksworld | Mystery Blocksworld | Movie | Gripper | Average |
> |-----------------------|-------------|----------------------|-------|---------|---------|
> | Direct_GPT-4o        | 0.7         | 0.7                  | 0.5   | 8.8     | 2.7     |
> | Direct_o1-preview     | 26.3        | 87.9                 | 25.7  | 23.8    | 40.9    |
> | CoT_GPT-4o           | 2.1         | 4.0                  | 1.0   | 10.2     | 4.3     |
> | Code_GPT-4o          | 19.7        | 8.9                  | 7.3   | 8.2     | 11.0    |
> | Code_SMT GPT-4o      | 9.1        | 8.5                  | 10.6  | 12.9    | 10.3    |
> | LLMFP_GPT-4o         | 43.3        | 48.3                 | 58.6  | 141.6   | 73.0    |
>
> The table below shows the average cost comparison of all methods on the coffee task. We observe that although LLMFP is more costly than most of the baselines, it is cheaper than Direct with o1-preview with better performance. In addition, the average cost per query for all 9 tasks is around 0.1 dollars, indicating LLMFP is not very costly.
>
> Table 4: Average cost ($) per query comparison of LLMFP on the Coffee task.
> | Direct_GPT-4o | Direct_o1-preview | CoT_GPT-4o | Code_GPT-4o | Code_SMT_GPT-4o | LLMFP_GPT-4o |
> |--------------------------|----------------------------|-----------------------|-----------------------|---------------------------|-------------------------|
> | 0.008                      | 0.536                      | 0.013                   | 0.023                   | 0.024                       | 0.139                   |

---

> > ### Author Response · Authors · 2024-11-23
> > **Response to Reviewer dysu 3/6**
> >
> > **dysu-Q4: Baseline descriptions and failure modes discussion**
> >
> > dysu-A4:
> > We thank the reviewer for the valuable suggestion! All baselines are equipped with a result formatter to convert their generated plans, in various forms, to a fixed format for better evaluation. We’ve added more baseline descriptions in the updated draft. We briefly described the major failure cases of baselines as captions of Example baseline outputs in Appendix A.8.1. To make the discussion more structured, we include more detailed failure mode discussions in Appendix A.5.
> >
> > Here we provide a summary and some example failure cases:
> >
> > To summarize, Direct and CoT fail to solve multi-constraint problems because they do not have the capability to directly solve the optimal solution considering various constraints, intensive calculations, and numerous possible solutions, and they fail to solve multi-step problems because they cannot consider preconditions and effects of all actions accurately. For example, given a gripper task with one robot thus two grippers, CoT generates solution as: *["**pick** ball3 robot1 room3 left_gripper", "**pick** ball5 robot1 room3 right_gripper",  "move robot1 room3 room4", "**drop** ball5 robot1 room4 right_gripper", "**pick** ball1 robot1 room4 right_gripper", "**pick** ball4 robot1 room4 left_gripper"...]* which attempts to pick up three things with two grippers. This is due to the misconsideration of action preconditions.
> >
> > While Code and Code_SMT fail to solve the problems often due to the failure to encode the correct problem setup, query, and constraint/actions or failure to write codes with correct logic or syntax. For example, Code_SMT often fails to distinguish the difference between And and Implies, and Code sometimes ignores the task description “the finish time counts the time when the last robot stops working” for task_allocation task.

---

> > > ### Author Response · Authors · 2024-11-23
> > > **Response to Reviewer dysu 4/6**
> > >
> > > **dysu-Q5: Code variant that is forced to use SMT**
> > >
> > > dysu-A5: We thank the reviewer for bringing this up! We added another baseline called Code_SMT and evaluated it on all tasks and both LLMs. We include the main result in Table 1 and Table 2, and more discussions in the revision.
> > >
> > > For multi-constraint problems, although Code_SMT_GPT-4o performs poorly, Code_SMT_Claude 3.5 Sonnet is able to improve 18.2% compared to Code_Claude 3.5 Sonnet. This showcases the strong coding capability of Claude 3.5 Sonnet, especially the capability to understand and utilize the SMT solver. At the same time, it showcases the instability of different LLMs in reaching strong performance, motivating the need for frameworks like LLMFP that could overcome the existing limitations of LLMs.
> > >
> > > Both LLMs perform poorly for multi-step problems. As mentioned in dysu-Q3, we discussed the failure cases in Appendix A.5.
> > >
> > > Table 5: Optimal rate (%) comparison of LLMFP with baselines on 5 multi-constraint problems.
> > > | Method                          | Coffee | Workforce | Facility | Task Allocation | Warehouse | Average |
> > > |---------------------------------|--------|-----------|----------|-----------------|-----------|---------|
> > > | Direct_GPT-4o      | 0.8    | 2.6       | 0.0      | 0.0             | 0.0       | 0.7     |
> > > | Direct_o1-preview  | 25.9   | 47.6      | 4.8      | 4.0             | 66.0      | 29.7    |
> > > | CoT_GPT-4o         | 0.0    | 5.6       | 0.0      | 0.0             | 16.0      | 4.3     |
> > > | Code_GPT-4o        | 17.7   | 75.8      | 53.9     | 0.0             | 8.0      | 31.1    |
> > > | Code_SMT_GPT-4o    | 0.0    | 10.8      | 0.6      | 0.0             | 2.0       | 2.7     |
> > > | LLMFP_GPT-4o       | **64.7**   | **92.2**      | **70.7**     | **96.0**            | **72.0**      | **79.1**    |
> > > ||
> > > | Direct_Claude 3.5 Sonnet | 0.0    | 0.0       | 0.0      | 0.0             | 0.0       | 0.0     |
> > > | CoT_Claude 3.5 Sonnet    | 7.1    | 0.0       | 0.0      | 0.0             | 14.0      | 4.2     |
> > > | Code_Claude 3.5 Sonnet   | 59.8   | 71.9      | 47.3     | 0.0            | 42.0      | 44.2    |
> > > | Code_SMT_Claude 3.5 Sonnet | 75.6   | 36.8      | **49.7**     | 86.0            | 64.0      | 62.4    |
> > > | LLMFP_Claude 3.5 Sonnet  | **80.5**   | **88.7**      | 48.2     | **96.0**            | **90.0**      | **80.7**    |
> > >
> > > Table 6: Optimal rate (%) comparison of LLMFP with baselines on 4 multi-step problems.
> > > | Method                          | Blocksworld | Mystery Blocksworld | Movie  | Gripper | Average |
> > > |---------------------------------|-------------|----------------------|--------|---------|---------|
> > > | Direct_GPT-4o      | 41.5        | 0.8                  | 85.7   | 0.0     | 32.0    |
> > > | Direct_o1-preview  | 88.4        | 31.9                 | **100.0**  | 52.0    | 68.1    |
> > > | CoT_GPT-4o         | 39.9        | 2.7                  | 81.0   | 0.0     | 30.9    |
> > > | Code_GPT-4o        | 0.0         | 0.3                  | 0.0    | 0.0     | 0.1     |
> > > | Code_SMT_GPT-4o    | 0.0         | 0.0                  | 0.0    | 4.0     | 1.0     |
> > > | LLMFP_GPT-4o       | **96.2**        | **77.7**                 | **100.0**  | **76.0**    | **87.5**    |
> > > ||
> > > | Direct_Claude 3.5 Sonnet | 43.2   | 0.5                  | **100.0**  | 12.0    | 38.9    |
> > > | CoT_Claude 3.5 Sonnet    | 52.8   | 2.8                  | **100.0**  | 28.0    | 45.9    |
> > > | Code_Claude 3.5 Sonnet   | 0.0    | 0.0                  | 0.0    | 0.0     | 0.0     |
> > > | Code_SMT_Claude 3.5 Sonnet | 0.0   | 0.0                  | 0.0    | 0.0     | 0.0     |
> > > | LLMFP_Claude 3.5 Sonnet  | **93.0**   | **98.0**                 | **100.0**  | **76.0**    | **91.8**    |

---

> > > > ### Author Response · Authors · 2024-11-23
> > > > **Response to Reviewer dysu 5/6**
> > > >
> > > > **dysu-Q6: Are the methods explicitly instructed to provide optimal solutions?**
> > > >
> > > > dysu-A6:
> > > > For all methods, including LLMFP and baselines, we describe the goal of each multi-constraint task in the task description. For example, for the Coffee task, the task description *“...**The company's objective is to minimize the total cost**, including shipping beans, roasting, and shipping roasted coffee, while ensuring that all coffee produced meets or exceeds the demand at each retail location”* implicitly shows the goal is to find the plan that minimizes the total cost. LLMFP uses an SMT solver, which can **optimize the objective and guarantees to find the optimal solution if the formulation and generated codes are correct**. However, since all representations and codes are generated by LLMs, they are not 100% correct, so the success rates are not 100%.
> > > >
> > > >
> > > > The methods are not explicitly instructed to provide optimal solutions for multi-step problems. However, since SMT solver guarantees to find the solution if there exists one, it can rigorously show the solution does not exist for smaller timesteps and increase timestep, thus can always find the optimal solution if the formulation and generated codes are correct (similarly, the success rates are not 100%). This is an advantage of incorporating a complete and sound solver like SMT in our framework.
> > > >
> > > >
> > > > However, to better understand the capabilities of baselines, we modify the baseline prompts to explicitly instruct them to find the optimal solution and re-evaluate them as Direct_Opt, CoT_Opt, Code_Opt, and Code_SMT_Opt on the 4 multi-step problems. We include the optimal rates in the table below and also in Appendix A.7. Compared with Table2, we could observe that some baselines achieve better performance (from average 0.1% to 16.4% for Code_Opt_GPT-4o, and from average 30.9% to 36.7% for CoT_Opt_GPT-4o), while some achieve slightly worse performance(average 68.1% to 67.0% for Direct_Opt_o1-preview). However, **despite the changes due to the explicit instruction to find the optimal plan, LLMFP still could largely outperform all baselines.**
> > > >
> > > > Table 7: Optimal rate (%) comparison of LLMFP with baselines that explicitly instructed to generate optimal plans on 4 multi-step problems.
> > > > | Method                      | Blocksworld | Mystery Blocksworld | Movie  | Gripper | Average |
> > > > |-----------------------------|-------------|----------------------|--------|---------|---------|
> > > > | Direct_Opt GPT-4o          | 35.2        | 0.8                  | **100.0**  | 0.0     | 34.0    |
> > > > | Direct_Opt o1-preview       | 80.9        | 39.0                 | **100.0**  | 48.0    | 67.0    |
> > > > | CoT_Opt GPT-4o             | 33.4        | 2.3                  | 95.2   | 16.0    | 36.7    |
> > > > | Code_Opt GPT-4o            | 0.0         | 3.8                  | 61.9   | 0.0     | 16.4    |
> > > > | Code_SMT_Opt GPT-4o        | 0.0         | 0.0                  | 0.0    | 0.0     | 0.0     |
> > > > | LLMFP_GPT-4o               | **96.2**        | **77.7**                 | **100.0**  | **76.0**    | **87.5**    |
> > > > ||
> > > > | Direct_Opt Claude 3.5 Sonnet| 40.9        | 1.5                  | **100.0**  | 20.0    | 40.6    |
> > > > | CoT_Opt Claude 3.5 Sonnet   | 52.5        | 4.5                  | **100.0**  | 20.0    | 44.2    |
> > > > | Code_Opt Claude 3.5 Sonnet  | 0.0         | 0.0                  | 0.0    | 0.0     | 0.0     |
> > > > | Code_SMT_Opt Claude 3.5 Sonnet| 0.0       | 0.0                  | 0.0    | 0.0     | 0.0     |
> > > > | LLMFP_Claude 3.5 Sonnet     | **93.0**        | **98.0**                 | **100.0**  | **76.0**    | **91.8**    |

---

> ### Author Response · Authors · 2024-11-23
> **Response to Reviewer dysu 6/6**
>
> **dysu-Q7: Success rates of tested methods? Do they all achieve 100%?**
>
> dysu-A7:
> We thank the reviewer for the suggestion to include success rates as another metric for evaluation. We include the success rates of all methods in the tables below and also in Appendix A.3.
>
> Note that although the optimization goal is described in the task description for multi-constraint problems, we exclude the optimization goal when calculating the success rate and only evaluate whether the plan fulfills the task setup and the query. This would largely decrease the difficulty of multi-constraint problems for baseline methods. For example, even assigning all tasks to one robot is considered a success for the task allocation task. Thus, the success rates of all baselines for multi-constraint problems are significantly higher than the optimal rates. However, the success rates of LLMFP almost remain the same as optimal rates since the SMT solver guarantees to output the optimal result with the correct encoding. **Even when compared in such an unfair way, the performance of LLMFP still outperforms other baselines, with an average of 86.4%, 18.1% higher than the best baseline.**
>
> While for the multi-step problems, considering all initial conditions, predicate and action definitions, and goals are the same, developing a reasonable and correct plan is not significantly easier than developing an optimal plan with the least number of steps. Thus, the success rates of baselines are improved, but not significantly, compared to the optimal rates.
>
>
> Table 8: Success rate (%) comparison of LLMFP with baselines on 5 multi-constraint problems.
>
> | Method                      | Coffee | Workforce | Facility | Task Allocation | Warehouse | Average |
> |-----------------------------|--------|-----------|----------|------------------|-----------|---------|
> | Direct_GPT-4o               | 5.6    | 54.5      | 31.7     | **100.0**           | 42.0      | 46.8    |
> | Direct_o1-preview     | 26.3   | **92.6**  | 41.5     | 94.0            | 86.0      | 68.1    |
> | CoT_GPT-4o                 | 17.7   | 72.3      | 31.7     | **100.0**            | 82.0      | 60.7    |
> | Code_GPT-4o                | 18.8   | 76.2      | 64.6 | 92.0            | 90.0      | 68.3    |
> | Code_SMT-GPT-4o            | 0.0    | 10.8      | 1.2     | 0.0            | 34.0      | 9.2     |
> | LLMFP_GPT-4o           | **64.7** | 92.2     | **79.3**     | **100.0**       | **96.0**  | **86.4** |
> ||
> | Direct_Claude 3.5 (Sonnet)   | 5.3    | **91.3**  | 36.0     | **100.0**       | 76.0      | 61.7    |
> | CoT_Claude 3.5 (Sonnet)     | 10.9    | 60.6      | 1.2     | **100.0**            | 96.0      | 53.7    |
> | Code_Claude 3.5 (Sonnet)    | 61.3   | 89.2      | 59.1     | **100.0**       | 60.0      | 73.9    |
> | Code_SMT-Claude 3.5 (Sonnet)| 77.1   | 39.0      | 59.1     | 90.0            | 74.0      | 67.8    |
> | LLMFP_Claude 3.5 (Sonnet) | **80.5** | 88.7     | **61.6**     | **100.0**       | **92.0**  | **84.6** |
>
> ---
>
> Table 9: Success rate (%) comparison of LLMFP with baselines on 4 multi-step problems.
> | Method                      | Blocksworld | Mystery Blocksworld | Movie | Gripper | Average |
> |-----------------------------|-------------|----------------------|-------|---------|---------|
> | Direct_GPT-4o               | 56.1        | 1.0                | 90.5  | 16.0    | 40.9    |
> | Direct_o1-preview     | 90.9    | 37.9                | 100.0 | **76.0**    | 76.2    |
> | CoT_GPT-4o                 | 62.0        | 3.0                 | **95.2** | 10.0    | 42.5    |
> | Code_GPT-4o                | 0.0         | 0.3                 | 0.0   | 0.0     | 0.1     |
> | Code_SMT-GPT-4o            | 0.2         | 0.0                 | 0.0   | 4.0     | 1.0     |
> | LLMFP_GPT-4o           | **96.2**    | **77.7**                | **100.0** | **76.0**    | **87.5** |
> ||
> | Direct_Claude 3.5 Sonnet   | 54.5        | 0.5                | **100.0** | 56.0    | 52.7    |
> | CoT_Claude 3.5 Sonnet     | 76.1        | 3.2                 | **100.0** | 72.0    | 62.8    |
> | Code_Claude 3.5 Sonnet    | 0.0        | 0.0            | 0.0 | 0.0    | 0.0    |
> | Code_SMT-Claude 3.5 Sonnet| 0.0         | 0.0                 | 4.0   | 0.0     | 1.0     |
> | LLMFP_Claude 3.5 Sonnet | **93.4**    | **98.0**            | **100.0** | **76.0**    | **91.8** |
>
> [1] Li, Beibin, et al. "Large language models for supply chain optimization." arXiv preprint arXiv:2307.03875 (2023).
>
> [2] Gupta, Naresh, and Dana S. Nau. "On the complexity of blocks-world planning." Artificial intelligence 56.2-3 (1992): 223-254.

---

> > ### Comment · Reviewer_dysu · 2024-11-23
> >
> > Thank you for your extensive responses and extensive additional experiments, I deeply appreciate that. I have a few follow-up questions.
> >
> > > For 5 multi-constraint problems, they are all NP-hard problems. [...] these 3 problems are built as Mixed-integer linear programming (MILP) problems. [...] As MILP is known to be NP-hard, the first 3 problems are NP-hard.
> >
> > I disagree. The fact that a problem is an instance of MILP doesn't mean it's NP-hard. Note that e.g. simple path finding in a graph can also be framed as MILP. And it seems that e.g. the Coffee problem can be easily framed as a max-flow min-cost network (even for arbitrary number of suppliers, roasters, retailers, and coffee roast colours) and hence can be solved in polynomial time. It's hard to find a precise definition of other tasks to check.
> >
> > > we tested LLMFP on the Sokoban environment.
> >
> > Thank you for that experiment. While <=6x6 instances with <=2 boxes is not the top complexity one could expect, I acknowledge that it (a) is indeed a non-trivial widely-used test (provided that those instance were not chosen to be super-simple), and (b) is enough to show the superiority of LLMFP over baselines. It would be interesting to push it even further and see how far it can go, but that's just for your consideration, you don't have to show me additional experiments. Could you instead share how long does it take to solve those instances?
> >
> > > In Appendix A.4 we included the time and cost analysis for LLMFP. As the reviewer suggested, we add time and cost comparisons for all methods.
> >
> > Thank you for adding that. Please clarify what exactly means "Average wall time (s) per **query**" ? What exactly is a query? Is it a single problem instance? Are the queries evenly distributed among the methods? What's the average time required to solve a single instance for LLMFP (that's the most informative metric for me)?

---

> > > ### Author Response · Authors · 2024-11-24
> > >
> > > ### Q: Coffee problem can be easily framed as a max-flow min-cost network?
> > > A: Thank the reviewer for pointing this out! We agree that the fact that the problems can be formulated as MILP instances do not mean they are NP-Hard. We made some mistakes on the complexity analysis for the two instances of MILP problems: Coffee and Workforce (only the base setup). We have corrected them and revised the paper (Appendix A.1 Complexity Analysis). The new analysis is shown below. We would appreciate any corrections from your expertise:
> > > * The coffee problem can be framed as a max-flow problem, which can be solved in polynomial time. Specifically, some algorithms can solve the max-flow problem with $O(VE)$ or $O(V^2E)$
> > > * The workforce problem, with no additional constraint, can also be framed as a max-flow problem. However, different types of constraints are added by users to form different instances. Some types of queries can increase the complexity. For example, "What if Gu and Bob cannot work on the same day?". **Adding constraints to introduce conflicting workers turns the problem to be as hard as a maximum independent set problem (also NP-Hard)**, where we add an edge between the conflicted workers, and finding the maximum independent set.
> > > * The facility problem **is a NP-Hard** problem because it is encoded as Capacitated Facility Location Problem(CFLP). The formal definition of CFLP is as below:
> > >
> > >     $\min \sum_{i=1}^n \sum_{j=1}^m c_{ij} d_j y_{ij} + \sum_{i=1}^n f_i x_i$
> > >
> > >     $\text{s.t.} \quad \sum_{i=1}^n y_{ij} = 1 \quad \text{for all } j = 1, \ldots, m$
> > >
> > >     $\sum_{j=1}^m d_j y_{ij} \leq u_i x_i \quad \text{for all } i = 1, \ldots, n$
> > >
> > >     $y_{ij} \geq 0 \quad \text{for all } i = 1, \ldots, n \text{ and } j = 1, \ldots, m$
> > >
> > >     $x_i \in \{0, 1\} \quad \text{for all } i = 1, \ldots, n$
> > >
> > >     where $x_i = 1$ if facility $i$ is open, and $x_i = 0$ otherwise. $y_{ij}$ for $i = 1, \ldots, n$ and $j = 1, \ldots, m$, which represents the fraction of the demand $d_j$ filled by facility $i$.
> > >
> > > Despite the mistake, **at least 5 out of 9 planning problem categories (Facility, Task Allocation, Warehouse, Blocksworld, Mystery Blocksworld) we experimented on are proved to be NP-Hard**. Some instances of the rest 4 planning problems can also be NP-Hard because users queries can introduce additional constraints. This shows the ability of LLMFP to solve diverse planning problems including NP-Hard problems, and its capability to understand and generalize to different natural language constraint additions/modifications. We believe that the correction on Coffee and Workforce does not detract from the core contributions and significance of our work. We sincerely appreciate the reviewer for catching this and helping strengthen our revised draft.

---

> > > > ### Author Response · Authors · 2024-11-24
> > > >
> > > > ### Q: what exactly means "Average wall time (s) per query" ?
> > > > A: **A query is a single instance**. The task description only gives a basic setup of the problem. The query is the questions raised by the user that (a) describes the detailed initial and/or goal states, or (b) adds or modifies existing requirements of the tasks. Each query, together with the basic task description, together describe a single unique instance. For Coffee, the dataset we use has 267 queries. For blocksworld, it has 602 queries. The optimal rate metric we use is also defined as (# optimal solution / # all queries).
> > > >
> > > > * An example query for Coffee is: How would cafe cafe2 be affected if their demand were to increase by 29%?
> > > >
> > > > * An example query for Blocksworld is:
> > > >
> > > >     You have 4 blocks.
> > > >
> > > >     a is on top of c.
> > > >
> > > >     c is on top of b.
> > > >
> > > >     d is on top of a.
> > > >
> > > >     b is on the table.
> > > >
> > > >     d is clear.
> > > >
> > > >     Your arm is empty.
> > > >
> > > >     Your goal is to move the blocks.
> > > >
> > > >     a should be on top of d.
> > > >
> > > >     d should be on top of b.
> > > >
> > > >
> > > > ### Q: Could you instead share how long does it take to solve those instances (on the Sokoban environment)?
> > > >
> > > > A: As we use interactive deepening for multi-step problems, the solver starts from a small timestep to check satisfaction. If the solver finds a satisfiable solution for a given timestep, it is guaranteed to be the optimal solution; if the solver finds no solution given a timestep, it adds one more step and repeats the process.
> > > >
> > > > In the sokoban problem since the plan length for some instances is long, especially with 2 boxes or on a 6x6 map, the average runtime per instance is **1358.3 seconds** (we set timestep to start from 5). However, for a length-23 plan, when timestep=23, the runtime to solve for the solution is only **240.7** seconds.
> > > >
> > > > As we discussed in [Appendix A.1 LLMFP Performance on Sokoban], “although LLMFP is demonstrated to be capable of correctly encoding and solving the Sokoban problem, it is true that there are many more variables in the Sokoban problem than in other tasks because the problem is represented with a map with a large number of different positions. This slows down the speed of the SMT solver. To mitigate this problem, some potential solutions include 1) introducing methods to estimate the lower and upper bounds of step numbers needed and start from there, 2) developing heuristics to prioritize some possible options first, and 3) developing methods that put attention on a part of the map and ignore the unnecessary positions in the map. We would love to extend our work to explore these directions to make our framework more efficient.”
> > > >
> > > > We believe that the major contribution of our paper is to **allow the LLMs to be able to correctly encode the problem**. For multi-step problems like this, correctly encoding the initial conditions, goals, and actions with codes is the most difficult part. For larger maps, the difficulty of encoding using LLMFP is only slightly higher even if there are more object descriptions in the prompt. Although the solving runtime increases exponentially for larger maps, this is an inherent problem for the SMT solver. This can be potentially solved by supporting different solvers in LLMFP, which we will explore as future work.

---

> > > > > ### Author Response · Authors · 2024-11-27
> > > > > **Message from authors -- last day of draft revision**
> > > > >
> > > > > Dear reviewer dysu,
> > > > >
> > > > > We sincerely appreciate your time and efforts in evaluating our paper. Since this is the last day we can revise the draft, we would like to confirm whether our responses have effectively addressed your concerns. To summarize, in our latest response, we updated the complexity analysis of tasks and answered further questions regarding query and Sokoban runtime.
> > > > >
> > > > > If there are any additional points you'd like us to discuss or consider, please do not hesitate to let us know. Your insights have been invaluable, and we're grateful for your feedback on our work. We look forward to further discussions with you!
> > > > >
> > > > > Best,\
> > > > > Authors of paper 6029

---

> > > > > > ### Comment · Reviewer_dysu · 2024-11-28
> > > > > >
> > > > > > Thank you once again for your extensive responses. My questions have been resolved. I will maintain my rating and recommend accepting the paper.

---

> > > > > > > ### Author Response · Authors · 2024-12-01
> > > > > > >
> > > > > > > Dear reviewer dysu,
> > > > > > >
> > > > > > > Thank you for your reply! We are glad to hear that our response addresses your concerns and that you recommend accepting the paper. Since your score is [borderline accept], would you please let us know what concerns are holding you back from further raising the score to [accept]? We are more than happy to address them!
> > > > > > >
> > > > > > > Best,\
> > > > > > > Authors of paper 6029

---

> > > > > > > > ### Comment · Reviewer_dysu · 2024-12-02
> > > > > > > >
> > > > > > > > If a score of 7 were available, I would have selected it. However, given the current scale, I must choose between 6 and 8. The rebuttal addressed my primary concerns, and I view this paper as a good contribution. That said, LLMs are not my area of expertise, and my familiarity with the latest advancements in this domain is limited to fully assess the paper's novelty and impact. Therefore, while I support accepting this paper, I'm not confident enough to raise my rating to 8.

---

### Official Review · Reviewer_8M6X · 2024-11-03

**Soundness:** 2
**Presentation:** 2
**Contribution:** 1
**Rating:** 1
**Confidence:** 4

**Summary:**

LLMFP is proposed which leverages LLMs to tackle complex planning problems by formulating them as optimization tasks.

**Strengths:**

LLMFP's ability to handle a wide variety of planning problems without task-specific examples is a significant strength.

**Weaknesses:**

1. The baselines for comparison do not seem to be a fair comparison to LLMFP. See questions.
2. The related work does not cover relevant set of papers that should have been used a baseline to compare this work. Mentioning a few of them below -
[1] Webb, T., Mondal, S. S., Wang, C., Krabach, B., & Momennejad, I. (2023). A Prefrontal Cortex-inspired Architecture for Planning in Large Language Models. arXiv preprint arXiv:2310.00194.
[2] Fabiano, F., Pallagani, V., Ganapini, M. B., Horesh, L., Loreggia, A., Murugesan, K., ... & Srivastava, B. (2023, December). Plan-SOFAI: A Neuro-Symbolic Planning Architecture. In Neuro-Symbolic Learning and Reasoning in the era of Large Language Models.
[3] Katz, M., Kokel, H., Srinivas, K., & Sohrabi, S. (2024). Thought of Search: Planning with Language Models Through The Lens of Efficiency. In The First Workshop on System-2 Reasoning at Scale, NeurIPS'24.

**Questions:**

1. What is the definition of a planning problem in this paper?
2. Why are the baselines only LLMs when the proposed approach is a framework/architecture? LLM-PFC [1] approaches planning problems similarly and there are other baselines to consider like Plan-SOFAI [2].
3. LLMs when used with API's are found to hallucinate new API functions or overuse a specific API call. Is such behavior observed here?
4. When it is a planning problem, why not directly use a symbolic planner and why is this architecture beneficial?

---

> ### Author Response · Authors · 2024-11-23
> **Response to Reviewer 8M6X 1/2**
>
> We thank the reviewer for comments and suggestions! We provide some clarifications and discussions regarding the weakness and questions proposed by the reviewer, and experiments suggested by other reviewers (Please see the Revision Summary). We also updated a revised draft and colored the modifications/additional discussions with blue.
>
> **8M6X-Q1: Baselines of LLMFP**
>
> 8M6X-A1:
> The baselines for comparisons in our paper are **fair comparisons** in that they have the **same inputs** and are all **zero-shot methods with no task-specific examples**. To the best of our knowledge, LLMFP is the first general-purpose planning framework that requires no task specific examples or external critics. We were not able to locate other frameworks/architectures that provide such a fair comparison to LLMFP at the time of submitting the paper.
>
> Regarding the three papers -
>
> * We have added the three papers mentioned by the reviewer to the **Related Works** section as they are relevant LLMs for Planning papers.
> * However, these papers are **not comparable** to our framework LLMFP. This is because we focus on the setting where we do not have task-specific examples, external humans/verifiers/critics, whereas [1-3] require either domain-specific examples or feedback from real humans. Specifically, [1] requires task-specific in-context examples, [2] requires a Memory that is filled with 25 solved problems, and [3] is interactive and user-guided, which needs a real human person capable of validating the code.
> * We noticed that [1] recently submitted another version to ArXiv after the ICLR submission deadline, which added a new experiment that investigated a zero-shot version of their method, MAP, on one multi-step task, Mystery Blocksworld, and showed that MAP could solve **8.2%** of the problems. Meanwhile, LLMFP could achieve **77.7%** and **98.0%** optimal rates for GPT-4o and Claude 3.5 Sonnet. The comparison of the performance results further proves the effectiveness of our paper, which **significantly outperformed the zero-shot MAP.**
>
> **8M6X-Q2: Are LLMs hallucinate or overuse APIs?**
>
> 8M6X-A2:
> We do observe some hallucinations and misuse of the APIs. For example,
> * We provide a Max(variable_list) function to LLMs for the task allocation task. Although the input is a list of variables, LLMs sometimes input multiple variables separately and use it like Max(variable_1, variable_2,...). This phenomenon would result in runtime errors and can be corrected during later iterations.
> * As we discussed in Appendix A.4.5, LLMFP Failure Case Analysis, the major failure case for Warehouse is: Code Generator overwrites the provided API get_distance and provides 1 as the output. Thus, the distance between each station is mistakenly set to be 1. Although we do not provide a fix for this issue in our paper, it is easy to locate and correct this type of issue by adding a checker to check for repeated initialization of functions.
>
> Despite these hallucinations and misuse of APIs, they are sporadic and LLMFP still significantly outperforms all baselines.

---

> > ### Author Response · Authors · 2024-11-23
> > **Response to Reviewer 8M6X 2/2**
> >
> > **8M6X-Q3: Definition of planning problems**
> >
> > 8M6X-A3: We extend the classical planning problem and define our planning problem as a tuple $P= \langle \mathcal{S}, \mathcal{A}, \mathcal{C}, \mathcal{T}, \mathcal{I}, \mathcal{G}, f\rangle$, where $\mathcal{S}$ is a finite set of states, $\mathcal{A}$ is the set of actions, $\mathcal{C}$ is a set of constraints, $\mathcal{T}: \mathcal{S} \times \mathcal{A} \rightarrow \mathcal{S}$ is the transition function, $\mathcal{I} \subseteq \mathcal{S}$ is the initial state, $\mathcal{G} \subseteq \mathcal{S}$ is the set of goal states, and $f: \mathcal{S} \rightarrow \mathbb{R}$ is the cost function.
> >
> > For the coffee supply chain example:
> > * States $\mathcal{S}$: all possible states of raw/roasted/shipped coffee
> > * Actions $\mathcal{A}$: source coffee bean, roast beans into dark or light coffee, ship roasted coffee…
> > * $\mathcal{T}$: how state changes after actions
> > * Initial state $\mathcal{I}$:  all coffee beans are still in suppliers,
> > * Goal state $\mathcal{G}$: coffee are roasted and shipped to cafes, fulfilling cafe demands
> > * Constraints $\mathcal{C}$: explicit and implicit constraints, such as shipped coffee beans can not exceed supplier capacity, shipped coffee from roastery can not exceed received coffee beans, etc…
> > * f: calculate total cost
> >
> > Then the planning problem involves delivering a plan $\pi$ accomplishing the task specified considering all constraints at the cheapest cost.
> >
> > **8M6X-Q4: Why not use a symbolic planner**
> >
> > 8M6X-A4:
> > * All symbolic planners have some learning curve. To use symbolic planners, the end-users have to be the experts who understand, interpret, and program the problem to be utilized by the symbolic planners. We imagine that this means the end-users would require at least a bachelor’s degree in CS or relevant majors.
> > * To use our framework, the end-users are **anyone who speaks natural languages**. LLMs act as the interface with the end users who can use their daily spoken language and can **generalize** to different user **queries** and even different **tasks** with **no task-specific efforts**.
> >
> > Like other LLM tool-using frameworks [4,5], the purpose of LLM tool-using is to allow end-users to solve complex problems without becoming experts in using those tools. For example, it often requires at least 2 years for a graduate student to understand SMT solvers. Now, our framework enables common end-users to develop plans efficiently and rigorously without knowing what an SMT solver is.
> >
> >
> > [1] Webb, T., Mondal, S. S., Wang, C., Krabach, B., & Momennejad, I. A Prefrontal Cortex-inspired Architecture for Planning in Large Language Models. arXiv preprint arXiv:2310.00194.
> >
> > [2] Fabiano, F., Pallagani, V., Ganapini, M. B., Horesh, L., Loreggia, A., Murugesan, K., ... & Srivastava, B. Plan-SOFAI: A Neuro-Symbolic Planning Architecture. In Neuro-Symbolic Learning and Reasoning in the era of Large Language Models, 2023.
> >
> > [3] Katz, M., Kokel, H., Srinivas, K., & Sohrabi, S. Thought of Search: Planning with Language Models Through The Lens of Efficiency. In The First Workshop on System-2 Reasoning at Scale, 2024
> >
> > [4] Liang, Jacky, et al. "Code as policies: Language model programs for embodied control." 2023 IEEE International Conference on Robotics and Automation (ICRA). IEEE, 2023.
> >
> > [5] Liu, Bo, et al. "Llm+ p: Empowering large language models with optimal planning proficiency." arXiv preprint arXiv:2304.11477 (2023).

---

> > > ### Author Response · Authors · 2024-11-27
> > > **Message from authors -- last day of draft revision**
> > >
> > > Dear reviewer 8M6X,
> > >
> > > We sincerely appreciate your time and efforts in evaluating our paper. Since this is the last day we can revise the draft, we would like to confirm whether our responses have effectively addressed your concerns. To summarize, in our response,
> > > * We added the citations for the three mentioned papers to the Related Work section
> > > * We discussed why they are not comparable to our method (although a work updated its draft with one zero-shot experiment after the ICLR submission deadline, LLMFP significantly outperforms it on this task)
> > > * We answered the questions about planning problem definition, LLM hallucination, and reasons for not directly using a symbolic solver
> > >
> > > We also included other important updates and additional experiments we have done in the Revision Summary. If there are any additional points you'd like us to discuss or consider, please do not hesitate to let us know. Your insights have been invaluable, and we're grateful for your feedback on our work. We look forward to further discussions with you!
> > >
> > > Best,\
> > > Authors of paper 6029

---

### Official Review · Reviewer_qLBs · 2024-11-05

**Soundness:** 3
**Presentation:** 3
**Contribution:** 3
**Rating:** 6
**Confidence:** 3

**Summary:**

This paper addresses the problem of solving planning problems that are given in natural language. The proposed algorithm they propose – LLMFP -  is a workflow of multiple LLMs, including an LLM to extract variables and constraints from the text, an LLM to formulate the extracted variables and constraints as an SMT problem in a specific format,  an LLM to convert this format to code that can be run by an SMT solver, and an LLM to verify and correct mistakes by the other LLMs. This LLM workflow is evaluated against the other LLM-based methods to solve planning problems, including one that is similar to the LLMFP but creates PDDL instead of an SMT problem. The authors also examine how results can be better by adding some task-specific expertise. The results over a set of benchmark problems show that LLMFP is, in general, much better than the baselines.

**Strengths:**

Strength
- The paper is in general clear (even if it is sometimes hand-wavey)
- The problem is interesting and the related work seems to cover all bases
- The results are impressive, and much better than the baselines.
- The proposed workflow makes sense and works well.

**Weaknesses:**

- I’m not sure if the novelty of the proposed work over the PDDL-based approach is sufficiently novel for a top conference.
- The appendix is huge (~40 pages!). This seems to me not reasonable, as the main paper should be self contained.
- The presentation is too much hand-wavy. It would be great to try to capture more of it in a more formal manner

**Questions:**

1.  As the authors noted, LLMs have been used to translate natural language to planning problems. Similarly, the mapping from planning to SMT is well known in the planning literature. So, is the novelty is limited to combining the two ideas together??
2. In page 3, just above the first paragraph, you seem to say that encoding to PDDL requires more human effort than encoding to SMT. Can you elaborate why?
3. How do you encode the length of the plan? When compiling planning to SAT or SMT, this is an issue because the solver (SAT/SMT) requires to set an upper bound while in PDDL it does not have too.

---

> ### Author Response · Authors · 2024-11-23
> **Response to Reviewer qLBs 1/4**
>
> We thank the reviewer for the constructive comments and helpful feedback! You brought up some great questions and suggestions, which have helped improve our work. **We have added additional experiments to show that the PDDL-based approach cannot solve some planning problems we consider in this paper**, and experiments suggested by other reviewers (Please see the Revision Summary). We address the reviewer's concerns below. We also updated a revised draft and colored the modifications/additional results and discussions with blue.
>
> **qLBs-Q1: Novelty of the proposed work over the PDDL-based approach**
>
> qLBs-A1:
> Our framework is **the first of its kind to perform zero-shot planning from natural language description across various domains**. We argue that the existing approaches, including the PDDL-based approach, are not sufficient to solve the problem we consider in this paper. Specifically,
> * **Existing PDDL-based approaches rely on task-specific efforts.** Our LLMFP framework does not need any task-specific examples or human inputs. However, current PDDL-based approaches depend either on task-specific in-context examples[1,3,4], existing PDDL domain files[1,5], or human corrections[2]. These all limit their cross-task generalization capabilities.
> * **PDDL-based approaches cannot handle general planning problems.** Even if the task-specific efforts in the existing PDDL-based approaches can be omitted in the future version of LLMs, the fact that PDDL planners need domain and problem files limits their applicability to non-PDDL problems that we consider in this paper. For example, the multi-constraint problems, such as the Coffee task in our paper, cannot be solved using PDDL-based planners in a zero-shot way. Specifically, the Coffee task’s goal is to develop plans that satisfy retail location demands considering supplier capacities with minimized total cost. The definition of actions, as needed in the PDDL domain file, is not clear and straightforward. We slightly modified the prompts from [1], a work that uses LLMs to generate PDDL domain files, to generate actions and their preconditions and effects for the Coffee task, given only the description of the natural language task (same setting as LLMFP). The method in [1] cannot generate reasonable action definitions. One representative failure mode is:
>     * Action 1: Source Beans from Supplier
>     * Action: This action enables the company to source a unit of coffee beans from a supplier.
>     * Parameters:
>         1. `?s - supplier`: the supplier from which to source beans
>         2. `?f - facility`: the facility to which the beans are shipped
>     * Preconditions:
>         (and
>             (supplier-has-capacity ?s)
>             (facility-can-receive ?f))
>     * Effects:
>         (and
>             (not (supplier-has-capacity ?s))
>             (facility-has-beans ?f))
>         * Failure reason: This action sources only one unit of coffee beans, but the effects mean the supplier would not have any capacity after the action, which does not take the number of coffee beans capacity of suppliers into consideration.
>
>     Other than this, actions are also not defined comprehensively. For example, from the output, we observe that only “Ship Dark Coffee to Retail Location” is defined, but shipping light coffee is not considered.
>
>     These results show the generation of PDDL domain files is not straightforward for non-PDDL problems, which limits PDDL-based approaches’ applicability. Even for PDDL problems, no existing PDDL-based approach can plan PDDL problems in a zero-shot way without PDDL domain files or human corrections. However, LLMFP is able to handle diverse sets of problems, including multi-constraint problems and multi-step problems without task-specific examples, as shown in Table 1 and 2  in the paper.
> * LLMFP can use any solvers. Although we use SMT as the solver in this work, LLMFP can be adapted to any planner or solver by updating the requirements and representation format in the prompts. We already included an example in Appendix A.7.3 in the original paper, where prompts can be easily modified to support using MILP. We would love to extend our work to support more solvers or even multiple solvers at the same time. In fact, we believe that each solver has its own specialized types of problems. Our future research plan is to equip LLMs with various solvers and allow LLMs to select preferred solvers or planners automatically.

---

> > ### Author Response · Authors · 2024-11-23
> > **Response to Reviewer qLBs 2/4**
> >
> > **qLBs-Q2: Appendix length**
> >
> > qLBs-A2:
> > The main reason why our appendix is long is that our framework is primarily a prompt-based framework with 5 components, and we tested LLMFP on 9 different tasks. In an effort to promote reproducibility and transparency, we put all prompts, task inputs, and task outputs in the appendix, all of them are lengthy. In addition, our framework has 5 components and we tested it on 9 different tasks, which makes our logs even longer. We believe that our main paper is self-contained since it includes all important descriptions, results, and analyses. We hope our effort to ensure transparency and reproducibility could be understood and appreciated.
> >
> > **qLBs-Q3: Hand-wavy presentation**
> >
> > qLBs-A3: We updated our draft and added some formal definitions and descriptions to make the presentation more rigorous and structured. Specifically,
> > * We have rewritten section 3.1 with a formal definition of the optimization problem
> > * We have modified sections 3.2 and 3.3 with more mathematical formulation to better illustrate our method
> > * We have also adjusted the description of 3.3 to make it more accessible to readers
> >
> > Please refer to our updated draft. All the changes are highlighted in blue.

---

> ### Author Response · Authors · 2024-11-23
> **Response to Reviewer qLBs 3/4**
>
> **qLBs-Q4: LLMs have been used to translate natural language to planning problems. Similarly, the mapping from planning to SMT is well known in the planning literature. So, is the novelty limited to combining the two ideas together?**
>
> qLBs-A4:
> We argue that LLMFP is more than translating natural language to planning problems. Specifically,
> * **Simply translating from natural language to planning problems and using SMT solvers does not work.** We added a new baseline approach Code_SMT, where we directly ask LLMs to translate and encode the natural language task description as a planning problem in SMT format and use SMT solvers to solve the problem. Code_SMT achieves an average of 2.7% and 62.4% for multi-constraint tasks, and 1.0% and 0.0% for multi-step tasks across two LLMs GPT-4o and Claude 3.5 Sonnet.
> * **Existing works using LLMs as translators need task-specific examples.** As we mentioned in the paper, [1,4,6] leverages LLMs as translators to convert problems into fixed formats(like PDDL or JSON) and input them to external planners. They accomplish this by giving LLMs example input-output pairs under the same contexts and leveraging LLMs as pure translators.
> * **LLMFP is a framework that enables LLM agents to understand and formalize planning problems from natural language descriptions.** When given a planning problem in natural language description, a planning researcher or engineer will formalize the natural language description as a formal optimization problem, and then use appropriate solvers to solve it. LLMFP framework enables LLM agents to act as expert planning engineers. **Without** any task-specific examples, LLMFP starts from understanding and analyzing the problem to generate valid goals, decision variables, and (explicit & implicit) constraints. Then, it summarizes all needed variables and necessary related information, again with no task-specific reference. This process is **non-trivial** and is the key to enabling LLMs to solve problems across **various inherently different tasks** in a **zero-shot setting**.
>
> Table 1: Optimal rate (%) comparison of LLMFP with baselines on 5 multi-constraint problems.
> | Method                          | Coffee | Workforce | Facility | Task Allocation | Warehouse | Average |
> |---------------------------------|--------|-----------|----------|-----------------|-----------|---------|
> | Direct_GPT-4o      | 0.8    | 2.6       | 0.0      | 0.0             | 0.0       | 0.7     |
> | Direct_o1-preview  | 25.9   | 47.6      | 4.8      | 4.0             | 66.0      | 29.7    |
> | CoT_GPT-4o         | 0.0    | 5.6       | 0.0      | 0.0             | 16.0      | 4.3     |
> | Code_GPT-4o        | 17.7   | 75.8      | 53.9     | 0.0             | 8.0      | 31.1    |
> | Code-SMT_GPT-4o    | 0.0    | 10.8      | 0.6      | 0.0             | 2.0       | 2.7     |
> | LLMFP_GPT-4o       | **64.7**   | **92.2**      | **70.7**     | **96.0**            | **72.0**      | **79.1**    |
> ||
> | Direct_Claude 3.5 Sonnet | 0.0    | 0.0       | 0.0      | 0.0             | 0.0       | 0.0     |
> | CoT_Claude 3.5 Sonnet    | 7.1    | 0.0       | 0.0      | 0.0             | 14.0      | 4.2     |
> | Code_Claude 3.5 Sonnet   | 59.8   | 71.9      | 47.3     | 0.0            | 42.0      | 44.2    |
> | Code-SMT_Claude 3.5 Sonnet | 75.6   | 36.8      | **49.7**     | 86.0            | 64.0      | 62.4    |
> | LLMFP_Claude 3.5 Sonnet  | **80.5**   | **88.7**      | 48.2     | **96.0**            | **90.0**      | **80.7**    |
>
> Table 2: Optimal rate (%) comparison of LLMFP with baselines on 4 multi-step problems.
> | Method                          | Blocksworld | Mystery Blocksworld | Movie  | Gripper | Average |
> |---------------------------------|-------------|----------------------|--------|---------|---------|
> | Direct_GPT-4o      | 41.5        | 0.8                  | 85.7   | 0.0     | 32.0    |
> | Direct_o1-preview  | 88.4        | 31.9                 | **100.0**  | 52.0    | 68.1    |
> | CoT_GPT-4o         | 39.9        | 2.7                  | 81.0   | 0.0     | 30.9    |
> | Code_GPT-4o        | 0.0         | 0.3                  | 0.0    | 0.0     | 0.1     |
> | Code-SMT_GPT-4o    | 0.0         | 0.0                  | 0.0    | 4.0     | 1.0     |
> | LLMFP_GPT-4o       | **96.2**        | **77.7**                 | **100.0**  | **76.0**    | **87.5**    |
> ||
> | Direct_Claude 3.5 Sonnet | 43.2   | 0.5                  | **100.0**  | 12.0    | 38.9    |
> | CoT_Claude 3.5 Sonnet    | 52.8   | 2.8                  | **100.0**  | 28.0    | 45.9    |
> | Code_Claude 3.5 Sonnet   | 0.0    | 0.0                  | 0.0    | 0.0     | 0.0     |
> | Code-SMT_Claude 3.5 Sonnet | 0.0   | 0.0                  | 0.0    | 0.0     | 0.0     |
> | LLMFP_Claude 3.5 Sonnet  | **93.0**   | **98.0**                 | **100.0**  | **76.0**    | **91.8**    |

---

> ### Author Response · Authors · 2024-11-23
> **Response to Reviewer qLBs 4/4**
>
> **qLBs-Q5: Why does Encoding to PDDL requires more human efforts than LLMFP**
>
> qLBs-A5: We would like to clarify that we do not mean encoding to PDDL inherently requires more human effort than encoding to SMT. Instead, existing methods using LLM+PDDL need human efforts [1-5]. Specifically, [1,3,4] needs task-specific in-context examples, [2] needs human corrections from experts, and [1,5] needs an existing PDDL domain file. We listed a comprehensive set of papers in the related work session and discussed their major differences from LLMFP.
>
> **qLBs-Q6: How do you encode the length of the plan?**
>
> qLBs-A6:
> * For multi-constraint problems such as Coffee, Workforce, Facility, Task_allocation, and Warehouse, since they are inherently combinatorial optimization problems, it is encoded as a single-step problem.
> * For multi-step problems such as Blocksworld, Mystery Blocksworld, Movie, and Gripper, we do not give LLMFP a fixed horizon. Instead, we use interactive deepening. The solver starts from timestep=1 to check satisfaction. If the solver finds a satisfiable solution for a given timestep, it is guaranteed to be the optimal solution; if the solver finds no solution given a timestep, it adds one more step and repeats the process. This process is repeated until reaching a predefined limit set by us.
>
> The requirement predefined limit is a shortcoming of the SMT solver. In the future, we would love to extend our work to explore methods that could help mitigate the runtime issue for large-scale problems. Some possible directions are: 1) introducing methods to estimate the lower and upper bounds of step numbers needed, 2) developing heuristics to prioritize some possible options first, and 3) developing methods that put attention on a part of the map and ignore the unnecessary positions in the map. In addition, one advantage of the LLMFP is that it can work with different solvers. Thus, we will also explore other solver options for long-horizon tasks.
>
> [1] Liu, Bo, et al. "Llm+ p: Empowering large language models with optimal planning proficiency." arXiv preprint arXiv:2304.11477 (2023).
>
> [2] Guan, Lin, et al. "Leveraging pre-trained large language models to construct and utilize world models for model-based task planning." Advances in Neural Information Processing Systems 36 (2023): 79081-79094.
>
> [3] ISR-LLM: Iterative Self-Refined Large Language Model for Long-Horizon Sequential Task Planning.
>
> [4] Xie, Yaqi, et al. "Translating natural language to planning goals with large-language models." arXiv preprint arXiv:2302.05128 (2023).
>
> [5] Silver, Tom, et al. "Generalized planning in pddl domains with pretrained large language models." Proceedings of the AAAI Conference on Artificial Intelligence. Vol. 38. No. 18. 2024.
>
> [6] Gundawar, Atharva, et al. "Robust Planning with LLM-Modulo Framework: Case Study in Travel Planning." arXiv preprint arXiv:2405.20625 (2024).

---

> > ### Author Response · Authors · 2024-11-27
> > **Message from authors -- last day of draft revision**
> >
> > Dear reviewer qLBs,
> >
> > We sincerely appreciate your time and efforts in evaluating our paper. Since this is the last day we can revise the draft, we would like to confirm whether our responses have effectively addressed your concerns. To summarize, in our response,
> > * We explained and demonstrated with experiments the novelty of LLMFP over PDDL-based approaches and LLM translation + SMT
> > * We updated the paper to make the presentation more rigorous and structured
> > * We answered the questions about the effort comparison of LLMFP and PDDL-based approaches and LLMFP encoding length
> >
> > We also included other important updates and additional experiments we have done in the Revision Summary. If there are any additional points you'd like us to discuss or consider, please do not hesitate to let us know. Your insights have been invaluable, and we're grateful for your feedback on our work. We look forward to further discussions with you!
> >
> > Best,\
> > Authors of paper 6029

---

### Author Response · Authors · 2024-11-23
**Revision Summary: additional experiments, discussions, and draft revisions**

We thank all reviewers for their thoughtful comments and suggestions! To help reviewers to better access the updates we have made, we include this summary of revisions as below:

To summarize the additional experiments and discussions we added:
* We added experiments on a complex **Sokoban** tasks for 15 queries
    * LLMFP achieves optimal rates of **80%**, outperforming all baselines.
    * Appendix A.1
* We added a baseline **Code_SMT** that is forced to use SMT for code generation for all tasks and LLMs
    * LLMFP outperforms Code_SMT, which achieves an average of 2.7% and 62.4% for multi-constraint tasks, and 1.0% and 0.0% for multi-step tasks across two LLMs GPT-4o and Claude 3.5 Sonnet
    * Table 1, Table 2, Section 4.2
* We provide results with **success rate** as the metric across all 9 tasks and for both LLMs
    * LLMFP still outperforms the baselines
    * Appendix A.3
* We added experiments that **explicitly** instruct baselines to output optimal solutions for multi-step problems
    * LLMFP still outperforms the baselines
    * Appendix A.7
* We added experiment to test **PDDL-based** approaches on non-PDDL problems
    * PDDL-based approaches cannot handle non-PDDL planning problems (multi-constraint problems)
    * qLBs-A1
* We added theoretical insights of why LLMFP outperforms baselines. We added experiment to directly include **formal mathematical definition** of Coffee as task description in prompt, and test Direct with o1-preview
    * LLMs cannot understand and solve an optimization problem. o1-preview only achieves 34.2%
    * Appendix A.5.4
* We added **wall time** comparison and **cost** comparison
    * LMFP runtime and cost are reasonable, comparable to using o1-preview
    Appendix A.4
* We added **complexity** analysis and **failure cases** analysis
    * All multi-constraint tasks, Blocksworld, and Mystery Blocksworld are proved to be NP-Hard
    * Appendix A.1, Appendix A.5
* We added experiments to **paraphrase** task descriptions
    * LLMFP is not sensitive to specific wordings of task descriptions
    * Appendix A.10.4

In addition to above additions, we revised the following parts of the main paper to make the presentation clearer and colored the modifications/additional results and discussions with blue:
* Section 2 Related Works: we add more citations
* Section 3.1 Overview, 3.2 Definer,  3.3 Formulator: we revised the presentation
* Figure 2: we replaced Figure 2 with two to-the-point examples of the JSON representation
* Section 4.2: we added discussions of new baseline Code_SMT

---

### Meta-Review · Area_Chair_G5k4 · 2024-12-21

**Metareview:**

The paper presents LLM-Based Formalized Programming (LLMFP), a framework for incorporating LLM's to solve natural language planning tasks. The framework uses an LLM iteratively with external planning tools to create a viable solution. The LLM is used to extract variables and constraints from text, construct and parse an instance of a SMT formula from the text, and catch errors in the process via another LLM. The experiments demonstrate improved performance compared to a direct application of an off-the-shelf LLM.

**Additional Comments On Reviewer Discussion:**

The authors provided detailed responses to the concerns raised by the reviewers. Most reviewers concur that the bar for acceptance has been reached. Unfortunately, reviewer 8M6X did not engage with the authors or other reviewers during the rebuttal and discussion periods despite recommending rejection. Furthermore, the review does not provide sufficient grounds for providing this paper the lowest possible rating of 1. To be fair to the authors I am not taking 8M6X's rejection recommendation into consideration for the final decision. The other reviewers have unanimously recommended acceptance. As it stands, especially accounting for the changes already incorporated in the paper after the rebuttal period, the paper makes a significant enough contribution to warrant acceptance.

---

### Decision · Program_Chairs · 2025-01-22

Accept (Poster)